# How reinforcement learning after next-token prediction facilitates learning

**Nikolaos Tsilivis**
New York University
nt2231@nyu.edu

**Eran Malach***
Harvard University

**Karen Ullrich**[†]
FAIR, Meta

**Julia Kempe**[†]
New York University & FAIR, Meta

## Abstract

Recent advances in reasoning domains with neural networks have primarily been enabled by a training recipe that optimizes Large Language Models, previously trained to predict the next-token in a sequence, with reinforcement learning algorithms. We introduce a framework to study the success of this paradigm, and we theoretically expose the optimization mechanisms by which reinforcement learning improves over next-token prediction in this setting. We study learning from mixture distributions of short and long "chain-of-thought" sequences encoding a single task. In particular, when the task consists of predicting the parity of $d$ bits and long sequences are rare, we show how reinforcement learning after next-token prediction enables autoregressive transformers to generalize, whereas mere next-token prediction requires extreme statistical or computational resources to do so. We further explain how reinforcement learning leverages increased test-time computation, manifested in longer responses, to facilitate this learning process. In a simplified setting, we theoretically prove that autoregressive linear models following this training recipe can efficiently learn to predict the parity of $d$ bits as long as the proportion of long demonstrations in the data mix is not exponentially small in the input dimension $d$. Finally, we demonstrate these same phenomena in other settings, including the post-training of Llama-series models on mixture variations of common mathematical reasoning benchmarks.

## 1 Introduction

The application of reinforcement learning techniques to large neural networks has led to many machine learning advances and successes (Silver et al., 2016; 2017; Berner et al., 2019; Bakhtin et al., 2023). In the realm of Large Language Models (LLMs), a recent paradigm that leverages reinforcement learning consists of two main ingredients: training a large model, such as an autoregressive transformer (Vaswani et al., 2017) on diverse types of data (sequences) from the web with a next-token prediction objective (Radford et al., 2019), followed by a fine-tuning stage with a reinforcement learning algorithm that seeks to improve model generations with respect to a reward function[1]. The latter part is, simply put, a *guess-and-check* procedure (Recht, 2025): the model generates one or more guesses per prompt, a reward function evaluates them (usually for correctness), and training proceeds on (prompts, guesses) with a next-token prediction loss that weights each sequence according to its reward. When this procedure is applied on a frontier model using a mathematical, logical or reasoning reward, it leads to rapid increase in the model's capabilities in related domains and is often accompanied by a significant length increase in model response. Prominent examples of such "reasoning" language models include OpenAI's o1 model (Jaech et al., 2024) and Deepseek's R1 model (DeepSeek-AI et al., 2025). The prolonged LLM generation that precedes an answer has

---

[*]Currently at Apple.

[†]Joint senior authors.

[1]There is sometimes an intermediate stage of supervised fine-tuning that aims to hone specific model skills, but we treat it as part of the first stage of next-token prediction optimization in our discussion.

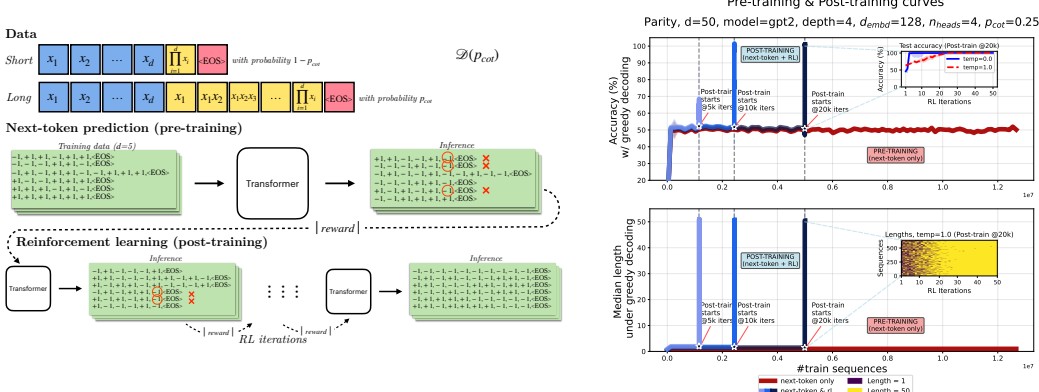

Figure 1: *Left*: **An illustration of our main learning setting:** a mixture of long and short sequences encoding the parity of $d$ bits, along with a representation of pre-training and post-training for $d$=5. *Right*: **Advantage of next-token prediction followed by reinforcement learning over mere next-token prediction** in predicting the parity of $d$=50 bits with a transformer trained from scratch. The red line corresponds to pre-training using next-token prediction, while blue lines correspond to the same pre-training runs, each followed by GRPO (with a final token accuracy reward) from a different checkpoint. Lines correspond to median across 3 seeds. *Top*: Test accuracy under greedy decoding. *Bottom*: Median length of model response under greedy decoding. The inset figures zoom-in on the curves during post-training. Accuracy plateaus around 50% (random guess) during pre-training and then immediately grows during post-training. Also, length increases during post-training.

been dubbed the "thinking process" of the model, as it often resembles the "chain-of-thought" (Nye et al., 2021; Wei et al., 2022) of a human expert solving said task.

In this paper, we study how autoregressive models succeed in solving challenging prediction tasks by following this training recipe (next-token prediction followed by reinforcement learning [2]). To provide a theoretical account, we make a central simplification: we assume that pre-training data already contain correct and elaborate but perhaps *rare* demonstrations for a task of interest. We argue this is a natural and productive way to view internet-scale data, even for the most challenging tasks. Based on this modeling assumption, we study and explain:

1. Why in certain cases it is difficult for a model to generalize during pre-training.
2. How reinforcement learning (RL) leads to a rapid improvement in terms of samples, and
3. What optimization pressures cause increase in the length of the response.

Specifically, we study learning from a distribution that contains both short and long, chain-of-thought-like, sequences encoding a single task. The first part of the paper (Sections 2, 3, 4) is devoted to the task of predicting the *parity* of $d$ bits, while later on we consider more variations of this learning setup. Predicting the parity, or XOR, of objects has been an influential learning problem in the study of artificial neural networks (Minsky & Papert, 1969; Shalev-Shwartz et al., 2017; Daniely & Malach, 2020; Abbe et al., 2023; Barak et al., 2022; Glasgow, 2024), statistical learning more broadly (Kearns, 1998) and biological learning in animals (Howard et al., 2022). Through extensive empirical simulations on transformers trained from scratch (Section 3), we show that if the task complexity is high (large input dimension $d$) and long demonstrations are scarce, then mere next-token prediction results in a model that fails to predict the parity on unseen examples. On the other hand, when switching to an RL algorithm with a correctness reward, model performance improves rapidly and the length of the generated sequences increases. The results hold for a large array of model choices, problem and optimization hyperparameters, and reinforcement learning algorithms. Figure 1 (Left: Illustration, Right: Simulations) summarizes the landscape.

We demonstrate that this learning behaviour is primarily driven by two factors: (a) during pre-training, the model learns much faster from long demonstrations than from short ones and remains

---

[2]In the argot of LLMs, these two stages are often called pre-training and post-training, respectively.

calibrated in how often it generates long versus short responses, and (b) during post-training, when we reward successful generations and train with a reward-weighted loss, we tend to amplify the presence of long responses in the training batches, as the model has a much higher probability of being correct when the response is long rather than short. This is the mechanism that leads to length increase which in turn enables generalization. We must emphasize that the transformer's inability to learn during pre-training in this setup is not due to insufficient model size or depth, but rather due to insufficient number of samples; an estimation, not an approximation, limitation.

In Section 4, we theoretically capture most of the previously observed phenomena in a simplified setting. Let $p_{\text{cot}}$ be the proportion of long data in the parity mixture distribution $\mathcal{D}(p_{\text{cot}})$. We analyze pre-training (next-token prediction) with Stochastic Gradient Descent on $\mathcal{D}(p_{\text{cot}})$ for an autoregressive architecture that consists of a series of linear models (Malach, 2024), while for post-training, we consider the STaR algorithm (Zelikman et al., 2022) (which is a REINFORCE (Williams, 1992)-type algorithm) with a reward that verifies correctness of the whole chain-of-thought. In this setting, we prove that the model: (a) fails to generalize under greedy decoding during the course of pre-training if long demonstrations are rare (i.e., if $p_{\text{cot}} < 1/3$, which matches the empirical threshold observed with transformers) (Theorem 1), (b) learns fast from long demonstrations and remains length-calibrated (Theorem 1), and (c) succeeds in generalizing after the completion of both pre- and post-training in $O(\text{poly}(\text{d}))$ SGD iterations, provided that long demonstrations are not exponentially rare in the data distribution, while this is accompanied with length increase during RL (Theorem 2). The analysis captures the immediate progress during RL seen in practice: as we show, only $O\left(\log \frac{1-p_{\text{cot}}}{p_{\text{cot}}}\right)$ RL rounds suffice to obtain a generalizing model. To the best of our knowledge, this suite of results provides the first theoretical separation between next-token prediction and next-token prediction combined with RL in the autoregressive setting, as well as the first optimization result demonstrating length increase during RL in LLMs.

Finally, in Section 5, we experimentally probe the same behavior in other settings, including GPT2 (Radford et al., 2019) models trained from scratch on the task of number multiplication, and pre-trained Llama 3-8B models (Dubey et al., 2024) that are fine-tuned (first to predict the next-token and then with RL) on a short/long mixture version of common reasoning benchmarks (GSM8K (Cobbe et al., 2021) and MATH (Hendrycks et al., 2021) datasets).

## 2 SETUP: MIXTURE OF LONG AND SHORT SEQUENCES ENCODING PARITY

In this section, we present our main learning problem and the models and algorithms we consider.

**Data** We study the task of predicting the parity of $d$ bits given access to a source of sequences which either consist of: (i) input bits and their parity, or (ii) input bits, intermediate computations and the final parity – see Figure 1 for a demonstration. Let $\mathcal{X} = \{\pm 1\}^d$ be the input space of $d$ bits and $\mathcal{Y} = \{\pm 1, \texttt{<EOS>}\}^*$ be the output space of sequences, where $\texttt{<EOS>}$ is a special symbol denoting the end of a string. Let $\mathcal{D}(p_{\text{cot}})$ be a distribution over $\mathcal{X} \times \mathcal{Y}$, parameterized by $p_{\text{cot}} \in [0,1]$, such that[3]: $x_1, \ldots, x_d \sim \text{Rad}(1/2)$ and $(y_1, \ldots, y_{d+1}) = Z(x_1, x_1 x_2, \ldots, \prod_{i=1}^{d} x_i, \texttt{<EOS>}) + (1 - Z)\left(\prod_{i=1}^{d} x_i, \texttt{<EOS>}\right)$ where $Z \sim \text{Ber}(p_{\text{cot}})$. We will primarily be focused on the case where the mixture weight $p_{\text{cot}}$ is small, i.e., where the distribution contains more short responses than long.

Predicting the parity of uniform random bits is a difficult learning problem for neural networks, as it is believed to require exponentially many samples/iterations (Shalev-Shwartz et al., 2017; Daniely & Malach, 2020; Shoshani & Shamir, 2025). On the other hand, when intermediate computations are available in the form of an elaborate chain of thought, the parity function, as any other binary computable function, can be learned efficiently (Joshi et al., 2025; Malach, 2024; Wies et al., 2023).

**Models** In our experiments, we train decoder-only transformers (Vaswani et al., 2017), a standard approach for language and sequence modeling. We experiment with the GPT-2 (Radford et al., 2019) and Mistral (Jiang et al., 2023) variants of the transformer architecture. See Appendix C.1 for further details on the specifics of the architectures.

---

[3]We denote Rademacher random variables (uniform $\pm 1$) as $\text{Rad}(1/2)$ and Bernoulli random variables with bias $p$ as $\text{Ber}(p)$. Our notation is described in Appendix E.

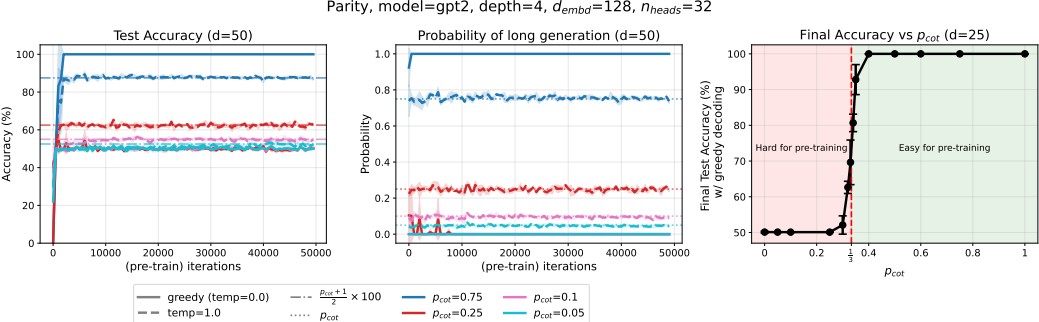

Figure 2: **Pre-training of transformers on mixture of long and short sequences encoding the parity of $d$ bits.** *Left*: Test accuracy during the course of pre-training. *Center*: The probability that a model's generation length is equal to the maximum length present in the training distribution (which equals $d$). Solid lines correspond to greedy decoding, while dashed lines correspond to sampling with temperature 1. Each color denotes a training distribution with a separate mixture coefficient $p_{\text{cot}}$. Figure shows average and 1 standard deviation across 3 seeds. *Right*: Test accuracy with greedy decoding at the end of pre-training (50k iterations) versus mixture coefficient $p_{\text{cot}}$. The red dashed line corresponds to the critical threshold. Each bullet is the median of 3 runs.

**Pre-training** In the first stage of training (pre-training), we train a transformer to estimate the probability of the next token on sequences drawn from $\mathcal{D}(p_{\text{cot}})$ (Radford et al., 2019). We consider online learning, sampling a fresh batch of data at each iteration.

**Post-training** After several steps of pre-training, we switch training algorithm and further optimize the model with reinforcement learning methods. We consider three reinforcement learning algorithms: *STaR* (Zelikman et al., 2022), *REINFORCE* (Williams, 1992) and *GRPO* (Shao et al., 2024). At a high level, all three algorithms seek to maximize the reward of model generations. We defer their description to Appendix A due to space constraints. We focus on two types of rewards:
*End-to-end correctness*: The reward function assesses whether the last token is equal to the correct answer: $r_{\text{e2e}}(\mathbf{x}, y) = \mathbb{1}\left\{ y[-1] = \prod_{i=1}^{d} x_i \right\}$, where $y[-1] \in \{\pm 1\}$ denotes the token appearing right before the `<EOS>` token in sequence $y$.
*Chain-of-thought correctness*: The reward function assesses whether the whole sequence is valid: $r_{\text{cot}}(\mathbf{x}, y) = \mathbb{1}\left\{ y = \left( x_1, x_1 x_2, \ldots, \prod_{i=1}^{d} x_i, \texttt{<EOS>} \right) \vee y = \left( \prod_{i=1}^{d} x_i, \texttt{<EOS>} \right) \right\}$. For all post-train methods, we can optionally set the sampling temperature $\tau_{\text{RL}}$ to a value different than 1.

## 3 EXPERIMENTS ON PARITY

We present our experiments with autoregressive transformers trained on the mixture distribution $\mathcal{D}(p_{\text{cot}})$ of long and short sequences encoding the parity of $d$ bits. Throughout training, we evaluate (test) accuracy in the prediction task, by measuring whether the token generated before `<EOS>` equals the parity of the input. We also measure the length of the generated sequence.

### 3.1 PRE-TRAINING VS (PRE- & POST-TRAINING)

First, we fix the input dimension $d$ to a large value (e.g. $d$=50) and consider pre-training (next-token prediction) on data sampled from $\mathcal{D}(p_{\text{cot}})$ for various values of mixture weight $p_{\text{cot}}$. We present results in Figure 2 for a GPT2 architecture. As we can see, performance under greedy decoding is determined by the value of $p_{\text{cot}}$: if the percentage of long sequences in the data is large enough (e.g. $p_{\text{cot}}$=0.75), then the model quickly learns to predict the parity of the input correctly by generating long responses and continues to generalize perfectly until the end of training. On the other hand, when the long sequences are under-represented in the source distribution, the model – under greedy decoding – generates short responses and performs on par with random guessing even after training on several millions of sequences (256 sequences per iteration for 50,000 iterations). In fact, the critical threshold is around $p_{\text{cot}}$=1/3 – see Figure 2 (*right*).

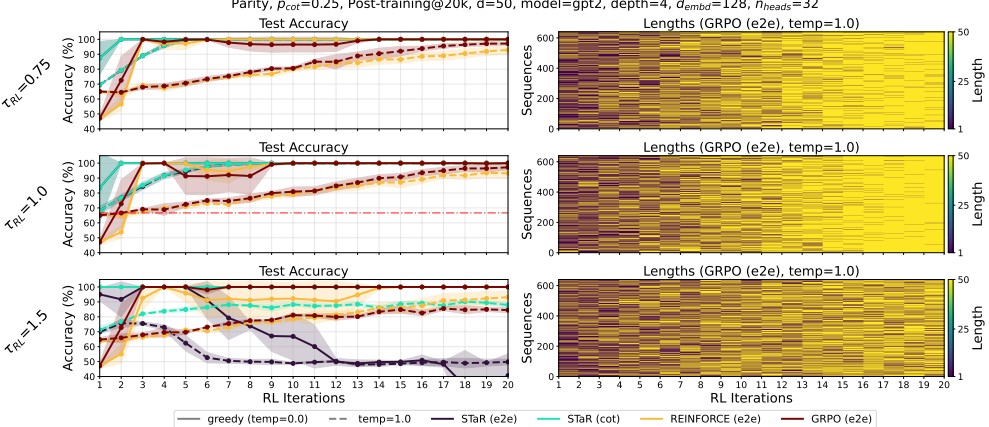

Figure 3: **Post-training of transformers on mixture of long and short sequences encoding the parity of $d$ bits with various RL methods and generation temperatures ($\tau_{\mathrm{RL}}$).** *Left*: Test accuracy during the course of post-training with greedy decoding (solid lines) and sampling with temperature 1 (dashed lines) for various RL methods. Figure shows average and 1 standard deviation across 3 seeds. *Right*: Length of generated response (sampled with temperature 1) during the course of a post-training run (GRPO with end-to-end reward) for 640 test inputs, after 20k pre-training iterations. **Note:** The sample size $n$ of each RL iteration differs amongst the RL algorithms: $n$=64 for GRPO, REINFORCE and $n$=3,200 sequences for STaR.

For the distributions with small $p_{\mathrm{cot}}$ where the models did not manage to generalize during pre-training, we consider switching to post-training: at some iteration, we stop training with the next-token objective of equation 5 and resume model training with a reinforcement learning algorithm. We show results in Figure 3 for $p_{\mathrm{cot}}$=0.25, where we plot post-training accuracy as a function of RL iterations for all post-train algorithms described in Sections 2, A and for a few different values of sampling temperature $\tau_{\mathrm{RL}}$. Progress during RL is immediate: after training on only a handful of sequences, model performance under greedy decoding grows from random guessing to 100% accuracy. This improvement is accompanied with concurrent increase in the length of the model's generations, which grows from 1 (min train length) to 50 (max train length). The previous observations are robust with respect to changes on the reinforcement learning algorithm and the hyperparameters of post-training (e.g. sampling temperature), with the notable exception of STaR with high sampling temperature $\tau_{\mathrm{RL}}$ and end-to-end reward, where the lack of negative rewards and the sparsity of the reward function $r_{\mathrm{e2e}}$ cause unstable post-training.

To further emphasize the difference between the two paradigms, we show aggregated results (pre- & post-training combined) in Figure 1 for $d$=50, $p_{\mathrm{cot}}$=0.25 and post-training with GRPO starting from various checkpoints. Observe that if post-training starts too early, then it might not lead to a generalizing model despite the length increase. However, GRPO, when started from later checkpoint, improves performance rapidly, and the number of post-training samples required for generalization is a miniscule fraction of the overall number of samples. On the other hand, mere next-token prediction does not result in a generalizing model (and the median greedy response remains short) even with access to $\sim 10^7$ training samples. This demonstrates the enormous sample-complexity gap between the two approaches for this task. Note that when the input dimension $d$ is smaller, pre-training eventually does lead to a well-generalizing model (if we optimize for sufficiently long) and post-training does not seem strictly necessary (see Figure 11 for $d \in \{11, 12\}$). However, for larger values of $d$, the amount of SGD iterations required is unreasonably large. In other words, the transformers' inability to succeed during pre-training **does not stem from insufficient model capacity** (small number of parameters or lack of depth[4]), but it is rather because next-token prediction on this distribution requires many **samples** to lead to a good predictor. In fact, as the complexity of the task increases, so does the advantage of reinforcement learning after next-token prediction

---

[4]A shallow transformer can approximate the parity function for any input dimension $d$; uniform attention for aggregating the bit values and a 2-layer MLP for computing their XOR – see Lemma 6 in (Liu et al., 2023).

over mere next-token prediction (Figure 12). Appendix C.1 contains more experimental results for various model and task hyperparameters.

## 3.2 DIFFERENT SPEEDS OF LEARNING & GROWTH OF TEST-TIME COMPUTE

We now describe what makes reinforcement learning so effective and why post-training leads to length increase in our simple setting. First, notice the curves in Figure 2 that correspond to sampling with temperature 1 during pre-training. Even when $p_{\text{cot}} < 1/3$, the accuracy of the models at the end of pre-training is greater than 50% – Figure 2 (Left). In fact, it is equal to $\frac{p_{\text{cot}}+1}{2} \times 100\%$ for all cases of $p_{\text{cot}}$. Furthermore, notice in the length plot of the same figure (Figure 2 (Center)) that the models are *calibrated* with respect to length, i.e., they generate long responses with probability $p_{\text{cot}}$. These two simple observations directly suggest that the models learn the two parts of the mixture "in parallel" during pre-training: after a few training iterations, when prompted with $d$ bits $x_1, \ldots, x_d$, the models learn to generate long and correct responses containing the parity $\prod_{i=1}^{d} x_i$ with probability $p_{\text{cot}}$ and short responses with probability $1 - p_{\text{cot}}$. As mentioned before, learning from the short sequences requires learning the parity of $d$ bits in a single prediction step, which belongs to a class of computationally difficult problems (Kearns, 1998) and is believed to be hard for standard neural networks to learn in practice (Shalev-Shwartz et al., 2017; Shoshani & Shamir, 2025). As a result, we do not expect the model to learn using any reasonable amount of samples, and hence it resorts to short, random guessing with probability $1 - p_{\text{cot}}$. Learning from long demonstrations, on the other hand, can be performed efficiently (Malach, 2024). This asymmetric learning difficulty leads to accuracy equal to $p_{\text{cot}} \times 100\% + (1 - p_{\text{cot}}) \times 50\% = \frac{p_{\text{cot}}+1}{2} \times 100\%$.

The learning process during pre-training sets the stage for what follows in post-training. It is perhaps simpler to understand the RL dynamics in the case of the STaR algorithm with the chain-of-thought correctness reward $r_{\text{cot}}$ (cyan lines in Figure 3). The objective of equation 6 implements next-token prediction solely on model-generated sequences which contain a correct chain of thought and a correct final answer. If, at the end of pre-training, the probability of a long, correct generation is $p_{\text{cot}}$ and the probability of a short, correct one is $(1 - p_{\text{cot}})/2$ (as suggested by Figure 2), then the model fits a data mixture which contains both long and short, correct sequences with proportions equal to $p_1 := \frac{p_{\text{cot}}}{p_{\text{cot}}+(1-p_{\text{cot}})/2}$ and $\frac{(1-p_{\text{cot}})/2}{p_{\text{cot}}+(1-p_{\text{cot}})/2}$, respectively. If, further, the model succeeds in fitting them, then the conditional distribution of model generations at the end of the first round of STaR has the same weights, provided the model remains length-calibrated and did not manage to learn from the short sequences. Continuing inductively, at the start of the $n$'th round, we effectively sample from distribution $\mathcal{D}(p_n)$, where $p_0 = p_{\text{cot}}$ and $p_n = \frac{2p_{n-1}}{1+p_{n-1}}$ converges to 1 exponentially fast. Once the effective coefficient $p_n$ becomes larger than $\frac{1}{3}$ (which happens when accuracy with temp. 1 exceeds $\frac{1/3+1}{2} = \frac{2}{3}$ – red dashed line in Figure 3), the model starts generating long, correct responses with greedy decoding, thus generalizing perfectly at the task. This is the mechanism that causes length increase in model response, which in turns allows the model to learn to generalize.

In Appendix C.2, we consider a variation of our setting, where the data include a third type of sequence consisting of a partial cot. We demonstrate how this affects length distributions at the end of RL, and the effectiveness of length penalties during RL as a function of prior pre-train iterations.

## 4 THEORY

Next, we theoretically capture the main empirical findings of the previous section and prove the identified mechanisms. We seek a theoretical model that, when $p_{\text{cot}}$ is small, fails to generalize during pre-training, while being capable of learning the long component of the distribution. As we show, the analysis of a *linear autoregressive model* (Malach, 2024) satisfies the above desiderata.

**Architecture** We study an autoregressive model that consists of a series of $d+1$ linear predictors. We reduce next-token prediction on $(\mathbf{x}, y) \sim \mathcal{D}(p_{\text{cot}})$ to a series of binary classification problems. At a high-level, the feature embedding used in each one of them involves (at most) second-degree monomials of the input, which is a canonical embedding choice. In particular, for the first output position, we consider a linear hypothesis class $\mathcal{H}_1 = \{\mathbf{x} \mapsto \langle \mathbf{w}_1, \mathbf{x} \rangle : \mathbf{x} \in \{\pm 1\}^d, \mathbf{w}_1 \in \mathbb{R}^d\}$. For the second output position, we need to learn a mapping from $\{\pm 1\}^{d+1}$ to $\{\pm 1, \texttt{<EOS>}\}$, as all three characters are valid second output tokens under distribution $\mathcal{D}(p_{\text{cot}})$. We build such a mapping in

two stages: first, a linear model decides whether the halting symbol needs to be outputted and, if not, we then make a decision between $\pm 1$ with a different linear model. This makes the analysis of the output of SGD simpler than directly considering a single multiclass model. That is, we define $\mathcal{H}_{2a} = \left\{ \mathbf{x} \mapsto \langle \mathbf{w}_{2a}, \phi_2(\mathbf{x}) \rangle + b_{2a} : \mathbf{x} \in \{\pm 1\}^{d+1}, \mathbf{w}_{2a} \in \mathbb{R}^{2d+1}, b \in \mathbb{R} \right\}$, where $\phi_2$ is a feature map that augments the input with all second-degree monomials involving the last input bit. For the second piece of the second position and for the rest of the positions, we proceed in a similar fashion[5]: we define $\mathcal{H}_l = \left\{ \mathbf{x} \mapsto \langle \mathbf{w}_l, \phi_l(\mathbf{x}) \rangle : \mathbf{x} \in \{\pm 1\}^{d+l-1}, \mathbf{w}_l \in \mathbb{R}^{2d+l-1} \right\}$. The embedding function is defined as $\phi_l \colon \mathbf{x} \mapsto [x_1, \ldots, x_{d+l-1}, x_{d+l-1} x_1, \ldots, x_{d+l-1} x_d]^T$ for $2 \leq l \leq d$. Our final model is a function $h$ that belongs to the product of these classes $\mathcal{H}_{\mathrm{AR}}^{\mathrm{Lin}} = \mathcal{H}_1 \times \mathcal{H}_{2a} \times \mathcal{H}_2 \ldots \times \mathcal{H}_d$. For all $d+1$ models, the output space is $\{\pm 1\}$ and we map the label to a character in $\{\pm 1, \texttt{<EOS>}, \epsilon\}$ accordingly. Note that the token at position $d+1$ is a constant function of the input and hence we omit it from our analysis. Furthermore, we define $\texttt{GREEDY}$ to be the operation of greedy decoding from the model logits (takes the sign of each model's output). By composing each individual component of $h$ with the sigmoid function, we also define the probability measure induced by $h$, namely $\pi_h(\cdot \mid \mathbf{x}) \in \Delta(\mathcal{Y})$. This corresponds to sampling from the model with temperature 1. Despite their apparent simplicity, linear autoregressive models can be efficient universal learners (Malach, 2024; Joshi et al., 2025). Our architecture with our embedding choices, in particular, is capable of identifying and learning any sparse parity function, hence we argue the model is not particularly tailored to the full parity function. Appendix E contains background and formal definitions.

Our first result characterizes the model during pre-training. We consider minimization of the (regularized) next-token prediction objective (logistic loss evaluated at all output positions) with SGD.

**Theorem 1.** *(Pre-training, Informal) Let $d \geq 2$, $p_{\mathrm{cot}} \in (0, 3/4)$. Consider distribution $\mathcal{D}(p_{\mathrm{cot}})$, as defined in Section 2, and linear hypothesis classes $\mathcal{H}_1, \mathcal{H}_{2a}, \mathcal{H}_2, \ldots, \mathcal{H}_d$ as defined earlier. Consider running Stochastic Gradient Descent (SGD) for minimizing the next-token prediction objective over $\mathcal{H}_{\mathrm{AR}}^{\mathrm{Lin}} = \mathcal{H}_1 \times \mathcal{H}_{2a} \times \mathcal{H}_2 \ldots \times \mathcal{H}_d$ with an additional $\ell_2$ regularization term. Then, for any $0 < \varepsilon < (d+1) \min \left\{ \frac{1}{2}, \ln \frac{1+p_{\mathrm{cot}}}{1-p_{\mathrm{cot}}}, \left| \ln \frac{1-p_{\mathrm{cot}}}{2p_{\mathrm{cot}}} \right| \right\}$ and after $O(\mathrm{poly}(d, 1/\varepsilon, 1/p_{\mathrm{cot}}))$ iterations, with high probability over sampling from $\mathcal{D}(p_{\mathrm{cot}})$, SGD returns model $h_{\mathrm{pre}} \in \mathcal{H}_{\mathrm{AR}}^{\mathrm{Lin}}$ for which the following hold:*

1. ***Length calibration:*** *When prompted with input $\mathbf{x} \in \{\pm 1\}^d$, model $h_{\mathrm{pre}}$ generates a long, correct sequence with probability approximately equal to $p_{\mathrm{cot}}$ and a short, correct one with probability approximately equal to $\frac{1-p_{\mathrm{cot}}}{2}$. That is, for any $\mathbf{x} \in \{\pm 1\}^d$,*

$$
\left| \pi_{h_{\mathrm{pre}}} \left( \left( x_1, x_1 x_2, \ldots, \prod_{i=1}^d x_i, \texttt{<EOS>} \right) \middle| \mathbf{x} \right) - p_{\mathrm{cot}} \right| \lesssim \varepsilon,
$$
$$
\left| \pi_{h_{\mathrm{pre}}} \left( \left( \prod_{i=1}^d x_i, \texttt{<EOS>} \right) \middle| \mathbf{x} \right) - \frac{1-p_{\mathrm{cot}}}{2} \right| \lesssim \varepsilon.
$$

(1)

2. ***Failure of greedy decoding:*** *For any $\mathbf{x} \in \{\pm 1\}^d$, it holds:*

$$
\texttt{GREEDY}(h_{\mathrm{pre}}(\mathbf{x})) = \begin{cases} (x_1, \texttt{<EOS>}), & \text{if } p_{\mathrm{cot}} < 1/3, \\ \left( x_1, x_1 x_2, \ldots, \prod_{i=1}^d x_i, \texttt{<EOS>} \right), & \text{otherwise.} \end{cases}
$$

(2)

*That is, test accuracy under greedy decoding is on par with random chance if $p_{\mathrm{cot}} < 1/3$, and perfect otherwise.*

Theorem 1 proves the main phenomena identified in the pre-training of transformers in Section 3, i.e., length calibration and failure of greedy decoding, in our linear autoregressive setting. Its formal statement (Theorem 4) and proof can be found in Appendix E. The proof proceeds by analyzing the output of SGD on each binary classification problem independently. Precisely, a standard convex learning result (Shalev-Shwartz & Ben-David, 2014) implies that SGD's output has generalization error comparable to the best-in-class. Then, strong convexity (afforded by the regularization term) and a case-to-case analysis establishes calibration and performance under greedy for each model.

---

[5] For notational brevity in the proofs, we use subscript 2 to denote the second piece of the second position.

*Remark* 1. The model's length calibration at the end of pre-training arises from the use of the logistic loss, which is a proper loss, in the next-token prediction objective.

*Remark* 2. The critical threshold $p_{\text{cot}}=1/3$, which is the same as the empirical critical threshold observed in transformers in Section 3, arises from the model's greedy decision at the first and second token. Indeed, in absence of a parity feature $\prod_{i=1}^{d} x_i$, it holds: $\mathbb{P}\left[y_1 = x_1\right] = p_{\text{cot}} + \frac{1-p_{\text{cot}}}{2} = \frac{1+p_{\text{cot}}}{2} > \frac{1-p_{\text{cot}}}{2} = \mathbb{P}\left[y_1 = -x_1\right]$. Thus, a calibrated model's "hard" decision for the first token will be $x_1$ for any $p_{\text{cot}}$. For the second step, we calculate the conditional probabilities : $\mathbb{P}\left[y_2 = x_1 x_2 | y_1 = x_1\right] = \frac{\mathbb{P}[y_2 = x_1 x_2, y_1 = x_1]}{\mathbb{P}[y_1 = x_1]} = \frac{2p_{\text{cot}}}{1+p_{\text{cot}}}$, while $\mathbb{P}\left[y_2 = \texttt{<EOS>} | y_1 = x_1\right] = 1 - \frac{2p_{\text{cot}}}{1+p_{\text{cot}}} = \frac{1-p_{\text{cot}}}{1+p_{\text{cot}}}$. That is, as long as $p_{\text{cot}} < 1/3$, the prediction for the second token after $x_1$ will be $\texttt{<EOS>}$. That is, greedy decoding will generate a short sequence $(x_1, \texttt{<EOS>})$. See also Appendix E.2. Theorem 1 asserts that SGD after sufficient many iterations returns a model that approximately matches these conditional probabilities.

Next, we study reinforcement learning with (regularized) STaR objective and with chain of thought reward $r_{\text{cot}}$ (defined in Section 2). Theorem 2 below is the main result of the paper: it shows that pre-training followed by post-training can be an efficient recipe for learning the parity from $\mathcal{D}(p_{\text{cot}})$, and an indispensable element of this success is the increase of model response during RL.

**Theorem 2.** *(Post-training, Informal) Let $h_{\text{pre}}$ be the output of SGD after pre-training under the conditions of Theorem 1 for $p_{\text{cot}} < 1/3$ and $\varepsilon \le c_0 \frac{p_{\text{cot}}}{1-p_{\text{cot}}}$ for a sufficiently small constant $c_0 > 0$. Consider post-training of $h_{\text{pre}}$ with the STaR algorithm using reward $r_{\text{cot}}$. Suppose each STaR round uses $O(\text{poly}(d, 1/\varepsilon, 1/p_{\text{cot}}))$ SGD iterations, and let $h^{(n)}$ denote the model after $n$ rounds. Then, there exists an integer $n^\star = O\left(\log \frac{1-p_{\text{cot}}}{p_{\text{cot}}}\right)$ such that with high probability over sampling from $\mathcal{D}(p_{\text{cot}})$, and the model-sampled outputs used by STaR in rounds $1, \ldots, n^\star - 1$, the following hold:*

1. ***Length increases:*** *The probability of a long generation increases:*

$$\left| \pi_{h^{(n)}}\left( \left(x_1, x_1 x_2, \ldots, \prod_{i=1}^{d} x_i, \texttt{<EOS>}\right) \middle| \mathbf{x} \right) - q_n \right| \lesssim \varepsilon, \qquad \forall \mathbf{x} \in \{\pm 1\}^d \qquad (3)$$

*where $|q_n - p_n| \lesssim 2^n \varepsilon$ and $p_{n+1} = \frac{2p_n}{1+p_n}$ for all $n \le n^\star$ with $p_0 = p_{\text{cot}}$.*

2. ***Perfect generalization:*** *After $n^\star$ RL rounds, the model under greedy decoding is only generating long responses and generalizes perfectly:*

$$\textit{GREEDY}\left(h^{(n^\star)}(\mathbf{x})\right) = \left(x_1, x_1 x_2, \ldots, \prod_{i=1}^{d} x_i, \texttt{<EOS>}\right), \qquad \forall \mathbf{x} \in \{\pm 1\}^d. \qquad (4)$$

The proof, which appears in Appendix E.4, proceeds in two steps. We first observe that the STaR objective for the $n$'th round is equivalent to next-token prediction with respect to a re-weighted version $\mathcal{D}(q_n)$ of the original distribution. Then, we bound the deviation of sequence $q_n$ from a "noise-less" sequence $p_n = \frac{2p_{n-1}}{1+p_{n-1}}$ that describes the evolution of the proportion of long data in the data mix.

*Remark* 3. Consider the case where long demonstrations are very rare, that is $p_{\text{cot}} \to 0$ as $d \to \infty$, which is perhaps the most interesting case. Theorem 2 shows that as long as $p_{\text{cot}} = \Omega(d^{-\kappa})$ for some constant $\kappa \in \mathbb{N}$, then we can learn the parity using $O(\text{poly}(d))$ SGD steps. This should be contrasted with the hardness of learning parities (Kearns, 1998; Shoshani & Shamir, 2025).

## 5 EXPERIMENTS WITH MATHEMATICAL REASONING BENCHMARKS

Next, we consider the same training recipe (next-token prediction followed by RL) in less idealized settings (including large pre-trained models) and for mixture distributions encoding tasks computationally deeper than parity, such as 5-digit multiplication and mathematical reasoning benchmarks.

**Numbers Multiplication** First, we study the task of multiplying n-digit numbers. Following our previous setup, we construct datasets that contain the n digits of the multiplier and multiplicand,

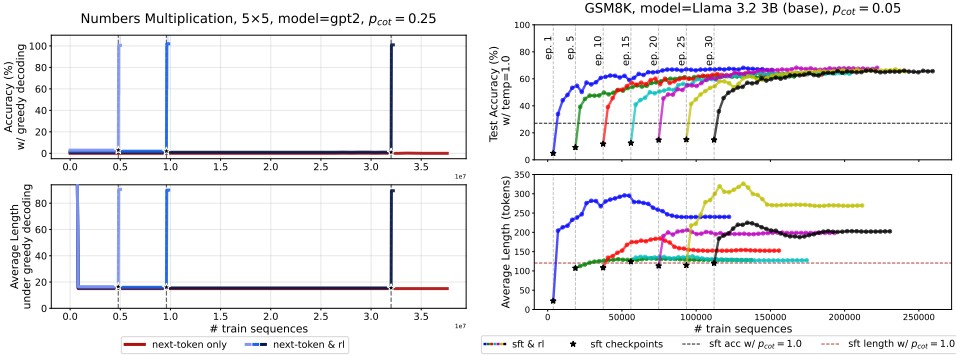

Figure 4: **Advantage of next-token prediction followed by reinforcement learning over mere next-token prediction** in (*Left*) multiplying two 5-digit numbers with a GPT2 model trained from scratch, and in (*Right*) solving grade-school math problems with Llama 3.2 3B (base). *Left*: Red line: pre-training on a 5x5 dataset with $p_{\text{cot}}$=0.25 using next-token prediction. Blue lines: the same pre-training runs, each followed by GRPO from a different checkpoint. Top: median test accuracy under greedy decoding (3 seeds). Bottom: median average response length (median over seeds, averaged over sequences). *Right*: ⋆ markers: supervised fine-tuning checkpoints on GSM8k with $p_{\text{cot}}$=0.05 (epochs 1–30). Colored curves: GRPO post-training from these checkpoints. Training sequences are reused across epochs; during post-training, multiple generations per input are counted.

together with the execution trace of the grade-school multiplication algorithm and the final answer, with probability $p_{\text{cot}}$, and just question, answer with probability $1-p_{\text{cot}}$ (Figure 22). We train GPT2 models from scratch on these datasets. Aggregated results for n=5, $p_{\text{cot}}$=0.25 are shown in Figure 4 (left). Full description and results (n=4, 7, $p_{\text{cot}} \in \{0.1, 0.25, 0.5\}$), along with detailed figures, are deferred to Appendix C.4. In summary, we observe similar learning dynamics as in the previous sections, except that here accuracy can even increase from ∼0% to 100% when switching to RL.

**Grade School & High School Mathematics (GSM8K & MATH datasets)** Finally, we experiment with pre-trained LLMs and mathematical reasoning benchmarks. We train Llama pre-trained models (Dubey et al., 2024) on variations of two standard mathematical benchmarks (GSM8k (Cobbe et al., 2021) and MATH (Hendrycks et al., 2021)). We format the data by surrounding the chain of thought of each sample with special "thinking" tokens. As previously, the cot is replaced by the empty string with probability $p_{\text{cot}}$. We first perform supervised fine-tuning (SFT) on these mixture datasets, and then switch to RL with GRPO and a reward that assesses both correctness of the response and consistency with the specified data format. For experimental details, see Appendices B.3, B.4.

We plot results for Llama 3.2 3B (base) on GSM8k, $p_{\text{cot}}$=0.05, in Figure 4 (right). We show test accuracy and average response length for the SFT checkpoints (stars) and the post-training curves starting from these checkpoints. As a baseline, we also plot the best SFT model trained with full chain of thought data; the $p_{\text{cot}}$=1.0 runs can be found in Figure 27. Early in SFT, accuracy is low and the average length of the model response is much smaller than the baseline. As training with the next-token objective continues, accuracy improves and length increases, but accuracy does not reach the baseline and appears to saturate at a lower value. Once we switch to RL with GRPO, the model rapidly reaches baseline accuracy while increasing its average response length. Notice how few samples RL requires to reach the baseline, even when starting from the first epoch checkpoint. This sample count should be contrasted with the number of SFT updates needed to reach comparable accuracy. Furthermore, in Figures 30 to 32, we provide examples of model completions during RL for this first checkpoint. We observe that early on, the model respects the SFT format but primarily generates short responses (Figure 30). As RL progresses, the model learns to produce longer responses with greater probability (Figure 31), eventually reaching a point where it consistently generates correct elaborate responses that mimic the chain of thoughts in the long-form part of the training distribution (Figure 32). These training patterns are consistent with the mechanisms observed in the parity setting of the previous sections. In Figure 27, we repeat this experiment for various $p_{\text{cot}}$ values. Beyond the point where accuracy exceeds the SFT baseline, the model continues to improve and its response length grows even further. This phase of post-training differs from

the situations observed in the previous sections and is likely due to the model leveraging pre-training data. Indeed, in Figure 33, we show that model generations at the end of RL are qualitatively different from the train sequences. Based on these observations, we hypothesize that the initial steep phase of RL reflects in-distribution learning (where the model "mines" SFT data), while the later phase reflects out-of-distribution gains. MATH experiments with Llama 3.2 3B, 3.1 8B (instruct) are deferred to Appendix C.5.

# 6 DISCUSSION

In this work, we introduced a theoretical framework to study the success of reinforcement learning applied after next-token prediction in Large Language Models. We demonstrated and proved that when the data contain rare elaborate sequences encoding a challenging target function, RL can help the model to learn by effectively up-sampling the presence of the long demonstrations in the data mix. Future work can address the limitations of our setting, by understanding better, for example, how noisy chains of thought affect the conclusions, as well as considering separate pre-train and post-train target functions.

**Learning with chain of thought data**    Recently, a few theoretical works have attempted to capture the success of autoregressive modeling with LLMs (Wies et al., 2023; Malach, 2024; Joshi et al., 2025). In particular, Joshi et al. (2025); Malach (2024) showed that next-token prediction can lead to computationally efficient learning of any efficiently computable binary function, which stands in sharp contrast to standard supervised learning where only a limited class of functions can be learned efficiently. However, these results rely on the assumption that a learner has access to a dataset with perfect chain of thought data. While this has turned out to be a very productive assumption for the theoretical study of autoregressive learning, it nonetheless constitutes a strong assumption. Our work takes a step forward in relaxing this requirement; it instead assumes that the dataset contains a small but nonzero (possibly polynomially small in the context length) fraction of chain of thought data. This is arguably a more natural assumption for modeling the presence of elaborate "good" demonstrations in the vast ocean of internet text. As we showed in Section 4, this is enough to guarantee efficient learning of the parity function and, interestingly, the guarantee is achieved through a variation of the popular post-training recipe used in state-of-the-art LLMs. We believe our proof strategy can be modified to show that pre-training followed by post-training can lead to efficient learning for other "hard" functions. We leave such a study for future work.

**Length increase as a learning phenomenon**    The common wisdom in the literature has been that the chain of thought during RL grows to enable the approximation of complex algorithmic tasks. Indeed, some computational problems require large computational depth to be executed with reasonable resources (Hastad, 1987) and the standard decoder-only transformer architecture with context length $n$ appears limited in what it can represent exactly with constant depth. On the other hand, a transformer augmented with $O(\text{poly}(n))$ chain of thought can simulate any function described by a circuit of polynomial size (Feng et al., 2023; Li et al., 2024; Merrill & Sabharwal, 2023), as the additional chain of thought provides computational depth to the model. As such, it seems theoretically satisfying that when chain of thought grows during RL, the model's performance improves on complex tasks. One shortcoming of this perspective is that it does not aim to explain why or how optimization pressures lead to length increase, which was the main focus of our work. On a more abstract level, the approximation advantage of long responses suggested by prior work is indeed fundamental, as one cannot hope to learn a task without the capacity to represent it[6]. Nevertheless, the learning advantage we captured in this paper can be more prevalent as it even appears in cases where representation is not an issue (such as the parity task we considered). Based on the above, we suggest interpreting the growth of the length during reinforcement learning of autoregressive models as a bona fide learning phenomenon.

---

[6]The actual situation is a bit more nuanced, as exact representation is not a strict requirement for learning through approximation (which is the goal in e.g. PAC learning). See Kamath et al. (2020); Malach & Shalev-Shwartz (2022) for some alternative definitions of approximation which can be fruitful for understanding learning.

ACKNOWLEDGEMENTS

JK thanks the Simons Foundation for support through the Collaborative Grant "The Physics of Learning and Neural Computation". NT and JK acknowledge support by the NSF through NRT Award 1922658.

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

**Reproducibility Statement**    To allow independent reproduction of our experimental results, we provide extensive experimental details in the main paper and in the appendix: (i) description of synthetic datasets, model and algorithms hyperparameters (Section 2 and Appendix B.1,B.2,B.3,B.4) (ii) number of runs per result, with mean/median ± std at each figure; (iii) compute details and estimates of GPU hours needed for reproduction, (iv) reference on any open-source code used. Our theoretical assumptions are mentioned in several places. Appendix E contains theoretical details, which were not possible to cover in the main text due to space constraints.

# A    MODELS AND ALGORITHMS

Herein, we describe the main models and training algorithms we consider in the parity experiments.

## A.1    ARCHITECTURE

A transformer is a sequence-to-sequence neural network, which in its simplest form consists of layers of self-attention followed by a position-wise feed-forward network. The term "autoregressive" indicates that the prediction of the model for each new token is conditioned only on tokens that appeared earlier in the sequence. To obtain a distribution over the next token, it is common to compose the output of the transformer with a soft-max layer across the vocabulary index, which can be potentially parameterized by a temperature value.

## A.2    PRE-TRAINING

The next-token prediction objective consists of the log loss of the model token distribution evaluated at output positions $1$ to $d$. We omit the first $d$ input positions of the sequence from the loss objective, as input bits $x_1, \ldots, x_d$ are distributed uniformly at random. If we denote the predicted sequence-level distribution by $\pi_\vartheta(y \mid \mathbf{x}) = \prod_{j=1}^{|y|} \pi_{\vartheta,j}(y_j \mid \mathbf{x}, y_{<j}) \in (0, 1)$, which is induced by the prediction of the transformer at each position composed with the soft-max, then training corresponds to:

$$\min_{\vartheta} \mathbb{E}_{(\mathbf{x},y)\sim\mathcal{D}(p_{cot})} \left[ \sum_{j=1}^{|y|} -\ln \pi_{\vartheta,j}\left(y_j \mid \mathbf{x}, y_{<j}\right) \right] \tag{5}$$

## A.3    POST-TRAINING

We consider three main reinforcement learning algorithms: *Self-Taught Reasoner* (STaR) (Zelikman et al., 2022), *REward Increment=Nonnegative Factor times Offset Reinforcement times Characteristic Eligibility* (REINFORCE) (Williams, 1992) and *Group Relative Policy Optimization* (GRPO) (Shao et al., 2024). Let $r : \mathcal{X} \times \mathcal{Y} \mapsto \{0, 1\}$ be a binary reward function. We consider outcome-based rewards, which assign one reward value per sequence.

**STaR**    In STaR (Algorithm 1), we first query the pre-trained model to generate responses for inputs. We then filter out those with 0 reward and fine-tune the model on the rest using the next-token prediction objective. We repeat this process for several rounds, each time sampling from the model returned by the previous round. The algorithm over one round corresponds to:

$$\min_{\vartheta} \mathbb{E}_{\substack{\mathbf{x}\sim\mathrm{Rad}(1/2)^{\otimes d}, \\ y\sim\pi_{\vartheta_{\mathrm{prev}}}(\cdot|\mathbf{x})}} \left[ \sum_{j=1}^{|y|} -\ln \pi_{\vartheta,j}\left(y_j \mid \mathbf{x}, y_{<j}\right) \middle| r(\mathbf{x}, y) = 1 \right], \tag{6}$$

where $\vartheta_{prev} \in \mathbb{R}^p$ corresponds to parameters returned in the previous round of the algorithm.

The pseudocode of STaR can be found in Algorithm 1.

**REINFORCE**    REINFORCE is a standard policy gradient algorithm. It seeks to maximize the expected reward of a sequence generated by the model:

$$\max_{\vartheta} \mathbb{E}_{\substack{\mathbf{x}\sim\mathrm{Rad}(1/2)^{\otimes d}, \\ y\sim\pi_{\vartheta}(\cdot|\mathbf{x})}} \left[ r(\mathbf{x}, y) \right]. \tag{7}$$

---

**Algorithm 1** Self-Taught Reasoner (STaR) Algorithm

---

**Require:** Pre-trained model parameters $\vartheta_0$, RL rounds $n$, Fine-tuning epochs per round $E$, Input distribution $\mathcal{D}_{\mathbf{x}} = \mathrm{Rad}(1/2)^{\otimes d}$, Reward function $r(\mathbf{x}, y)$, Number of samples to generate per round $N$.
1: Set $\vartheta = \vartheta_0$
2: **for** $r = 1$ to $n$ **do**
3:      Set $\mathcal{S} = \emptyset$
4:      **for** $i = 1$ to $N$ **do**
5:          Sample $\mathbf{x} \sim \mathcal{D}_{\mathbf{x}}$.
6:          Sample $y \sim \pi_{\vartheta}(\cdot \mid \mathbf{x})$.
7:          **if** $r(\mathbf{x}, y) = 1$ **then**
8:             Set $\mathcal{S} = \mathcal{S} \cup \{(\mathbf{x}, y)\}$.
9:          **end if**
10:      **end for**
11:      **for** epoch = 1 to $E$ **do**
12:          **for** each $(\mathbf{x}, y) \in \mathcal{S}$ **do**
13:             Update $\vartheta$ by taking a gradient step on the next-token prediction loss for $(\mathbf{x}, y)$.
14:          **end for**
15:      **end for**
16: **end for**
17: **return** Final model parameters $\vartheta$.

---

The gradient of the objective function can be written as:

$$\mathbb{E}_{\substack{\mathbf{x}\sim\mathrm{Rad}(1/2)^{\otimes d}, \\ y\sim\pi_{\vartheta}(\cdot|\mathbf{x})}} \left[\nabla_{\vartheta} \ln \pi_{\vartheta}(y|\mathbf{x}) r(\mathbf{x}, y)\right], \tag{8}$$

which implies that the algorithm can be recast as:

$$\min_{\vartheta} \mathbb{E}_{\substack{\mathbf{x}\sim\mathrm{Rad}(1/2)^{\otimes d}, \\ y\sim\pi_{\vartheta}(\cdot|\mathbf{x})}} \left[-\ln \pi_{\vartheta}(y \mid \mathbf{x}) r(\mathbf{x}, y)\right], \tag{9}$$

where the random variable $y$ is treated as a constant and not as a function of $\vartheta$. This is the surrogate objective that is optimized in practice. Observe the similarities with the STaR objective of equation 6. Furthermore, a widely adopted option is to center the rewards in order to reduce the variance of the updates during the execution of a stochastic optimization algorithm. This leads to the final version of the REINFORCE method:

$$\min_{\vartheta} \mathbb{E}_{\substack{\mathbf{x}\sim\mathrm{Rad}(1/2)^{\otimes d}, \\ y\sim\pi_{\vartheta}(\cdot|\mathbf{x})}} \left[\sum_{j=1}^{|y|} -\ln \pi_{\vartheta,j}(y_j \mid \mathbf{x}, y_{<j}) A(\mathbf{x}, y)\right], \tag{10}$$

where $A(\mathbf{x}, y) = r(\mathbf{x}, y) - \mathbb{E}_{\substack{\mathbf{x}\sim\mathrm{Rad}(1/2)^{\otimes d}, \\ y\sim\pi_{\vartheta}(\cdot|\mathbf{x})}} [r(\mathbf{x}, y)]$ is often called the advantage function.

**GRPO** Group Relative Policy Optimization, or GRPO for short, is a policy gradient algorithm, motivated by the class of Proximal Policy Optimization (PPO) algorithms (Schulman et al., 2017), that is widely used for post-training large language models with reinforcement learning. Its objective amounts to:

$$\mathbb{E}_{\substack{\mathbf{x}\sim\mathrm{Rad}(1/2)^{\otimes d}, \\ y^{(1)},\ldots,y^{(N)}\sim\pi_{\mathrm{old}}(\cdot|\mathbf{x})}} \left[\frac{1}{N}\sum_{i=1}^{N}\frac{1}{|y^{(i)}|}\sum_{j=1}^{|y^{(i)}|} \min\left\{r_{i,j}A_i, \ \mathrm{clip}(r_{i,j}, 1-\epsilon_{\mathrm{clip}}, 1+\epsilon_{\mathrm{clip}})A_i\right\}\right],$$

where $A_i = \frac{r(\mathbf{x}, y^{(i)}) - \mathrm{mean}(r(\mathbf{x}, y^{(1)}),\ldots,r(\mathbf{x}, y^{(N)}))}{\mathrm{std}(r(\mathbf{x}, y^{(1)}),\ldots,r(\mathbf{x}, y^{(N)}))}$ is a normalized reward, $\epsilon_{\mathrm{clip}}$ is a hyperparameter and $\pi_{\mathrm{old}}$ is a predicted distribution that corresponds to an earlier model checkpoint during the execution of the algorithm. We defined $r_{i,j} := \frac{\pi_{\vartheta,j}\left(y_j^{(i)}|\mathbf{x}, y_{<j}^{(i)}\right)}{\pi_{\mathrm{old},j}\left(y_j^{(i)}|\mathbf{x}, y_{<j}^{(i)}\right)}$. The term "Group" indicates sampling of a group of $N$ responses per input.

# B EXPERIMENTAL DETAILS

The experiments are implemented using PyTorch (Paszke et al., 2019).

## B.1 PARITY

We use the Transformers library (Wolf et al., 2020) for our parity experiments.

**Architecture** The transformer is initialized with context length $d'=2d$, to be able to process the whole sequence, where recall $d$ is the number of input bits. The two architectures we consider primarily differ in the type of positional encodings: GPT2 uses absolute positional encodings of the input sequence, while Mistral uses relative positional encodings (Su et al., 2024). We use the `GPT2Config`, `MistralConfig` config classes to define the models. We consider several model hyperparameters: depth (number of transformer blocks) $L \in \{2, 4, 8\}$, embedding dimension $d_{\text{emdb}} \in \{128, 256\}$, number of heads $n_{\text{heads}} \in \{4, 32\}$. By the default convention, each head has dimension $d_{\text{emdb}}/n_{\text{heads}}$.

**Pre-training** We consider distributions with

$$p_{\text{cot}} \in \{0, 0.05, 0.1, 0.25, 0.3, 0.32, 0.33, 0.34, 0.35, 0.4, 0.5, 0.6, 0.75, 1.0\}.$$

We use the Adam (Kingma & Ba, 2015) optimizer with learning rate $\eta_{\text{pre}}=10^{-3}$ and a batch size $B_{\text{pre}}=256$. Test statistics are estimated using 2,560 new samples.

In Figure 2 (right), each bullet corresponds to the final test accuracy with greedy decoding over 3 random seeds. The final accuracy of each run is defined as the average accuracy from the last 5 checkpoints of the pre-trained run (that is, average accuracy at iterations 47,500 to 49,500).

**Post-training** We continue post-training from the last checkpoint of pre-training. For STaR, we generate $3,200$ samples per RL iteration and train on the correct ones for 3 epochs with batch size 64. For REINFORCE and GRPO, we use batch size 64. We found it necessary to use a larger sample size per RL iteration for STaR. The GRPO experiments use $N = 4$ rollouts per input and clip parameter $\epsilon_{\text{clip}} = 0.2$. No KL penalty is applied (as outlined in Section A). We use the Adam (Kingma & Ba, 2015) optimizer with learning rate $\eta_{\text{pre}}=10^{-4}$. Test statistics are estimated using 640 new samples.

## B.2 NUMBER MULTIPLICATION

**Architecture** We use the GPT2 (Radford et al., 2019) architecture which contains 124,439,808 parameters and the default GPT2 tokenizer that includes a vocabulary of size 50,257 and utilizes Byte Pair Encoding (Sennrich et al., 2016).

**Pre-training** We consider datasets with $p_{\text{cot}} \in \{0.1, 0.25, 0.5, 1\}$. The dataset and the pre-training code is based on the public repository of (Deng et al., 2024). We train with the next-token prediction objective applied to the chain of thought (if it exists) and the answer tokens of the sequence. We use the AdamW (Loshchilov & Hutter, 2019) optimizer with learning rate $\eta_{\text{pre}}=5 \cdot 10^{-5}$, batch size $B_{\text{pre}}=32$, and maximum gradient norm equal to 1.0. We train for 15 epochs for the $4 \times 4$ task and 50 epochs for $5 \times 5$ and $7 \times 7$. All 3 training datasets consist of 808,000 unique samples. We train each model (3 random seeds per task) in a single GPU (NVIDIA A100-SXM4-80GB). For $4 \times 4$, each run takes approximately 3-5 hours to complete (depending on the value of $p_{\text{cot}}$). For $5 \times 5$ and $7 \times 7$, the completion time ranges in 21-38 hours.

**Post-training** We train with GRPO with an end-to-end reward function (that is, 1 if the tokens generated after #### correspond to the correct answer[7] and 0 otherwise). We use the AdamW (Loshchilov & Hutter, 2019) optimizer with learning rate $\eta_{\text{post}}=3 \cdot 10^{-6}$ (and Hugging-face's Transformers' (Wolf et al., 2020) default rest of hyperparameters), batch size $B_{\text{post}}=16$, group size (for GRPO) equal to 4, 2 steps of gradient accumulation (which makes the effective batch size equal to 32), no KL penalty, $\epsilon_{\text{clip}} = 0.2$. We use generation temperature $\tau_{\text{RL}} = 1.0$. We use the

---

[7]That is, "complete number match".

TRL library for efficient post-training (von Werra et al., 2020). We found it necessary to implement a wrapper around the model forward's function to force the model to be in evaluation mode during the generation of the rollouts for RL (that is, to turn off any sources of non-determinism in the model, such as dropout layers). Otherwise, under TRL's default implementation, the model was not able to generate correct responses, even during the very first RL iteration. We perform post-training for each random seed in a single GPU (NVIDIA A100-SXM4-80GB) which concludes in about 3.5 hours.

Test accuracy is estimated using a set of 1000 unseen pairs of numbers.

Estimate of total GPU hours needed to reproduce the pre-training and post-training results: 900

## B.3 GSM8K

Our code is based on the torchtune library (torchtune maintainers & contributors, 2024).

**Architecture**   We use the Llama 3.2 3B base model for our GSM8K experiments.

We split each dataset into equal parts, perform supervised fine-tuning (SFT) on the first half with the next-token prediction objective applied to the answer portion of the sequence, and then switch to RL on the second half.

**Supervised fine-tuning**   We train with the next-token prediction objective applied to the chain of thought and answer tokens of the sequence. We structure the data with preamble prompt (not calculating loss): `"A conversation between User and Assistant. The user asks a question, and the Assistant solves it. The assistant first thinks about the reasoning process in the mind and then provides the user with the answer. The reasoning process and answer are enclosed within <think></think> and <answer></answer> tags, respectively, i.e., <think>reasoning process here</think> <answer>answer here</answer>. User: {question} Assistant: "` and structure the following sequence as: `<think>cot</think> <answer>answer</answer>`, where `cot` is everything that exists before characters `####` in the original GSM8k dataset. As before, we drop the `cot` from each sample with probability $p_{\text{cot}}$.

We use the AdamW (Loshchilov & Hutter, 2019) optimizer with learning rate $\eta_{\text{sft}}=10^{-5}$, batch size $B_{\text{sft}}=256$, and bf16 precision. We train for 30 epochs. We consider datasets with $p_{\text{cot}} \in \{0.1, 0.25, 0.5, 1\}$. We use PyTorch for distributed training. Each run is performed in 8 GPUs (NVIDIA A100-SXM4-80GB) and takes 1.5 hour to complete.

**Post-training**   We train with GRPO with a reward function that has stepwise structure: 5 points for the presence of the answer tag, 5 points for the presence of the thinking tag, 20 points if it contains the correct answer in part of the response, and 100 points for a correct answer at the appropriate place. We found it important using the `math-verify` module of Huggingface for simplifying mathematical expressions and accurately rewarding generations. We use the AdamW (Loshchilov & Hutter, 2019) optimizer with learning rate $\eta_{\text{post}}=10^{-5}$, cosine learning rate scheduler, batch size $B_{\text{post}}=1$, group size (for GRPO) equal to 32, no KL penalty, $\epsilon_{\text{clip}} = 0.2$. We use generation temperature $\tau_{\text{RL}} = 1.0$. We perform post-training in 8 GPUs (NVIDIA A100-SXM4-80GB) and each run concludes in 31-35hours and 20hours for $p_{\text{cot}} = 0$.

Test statistics are estimated on the test set of GSM8k using 4 generations per question and taking the average.

Estimate of total GPU hours needed to reproduce the pre-training and post-training results: 11k

## B.4 MATH

Our code is based on the torchtune library (torchtune maintainers & contributors, 2024).

**Architecture**   We use the Llama 3.2 3B, and 3.1 8B instruct models.

We split each dataset into equal parts[8], perform supervised fine-tuning (SFT) on the first half with the next-token prediction objective applied to the answer portion of the sequence, and then switch to RL on the second half.

**Supervised fine-tuning**  We train with the next-token prediction objective applied to the chain of thought and answer tokens of the sequence. We structure the data with preamble prompt (not calculating loss):

```
<|begin of text|><|start header id|>system<|end header id|>
Cutting Knowledge Date: "
"December 2023
Today Date: 26 Jul 2024
A conversation between User and Assistant. The user "
"asks a question, and the Assistant solves it. "
"The assistant first thinks about the reasoning "
"process in the mind and then provides the user with the answer. "
"The reasoning process and "
"answer are enclosed within <think></think> "
"and <answer></answer> tags, respectively, i.e., "
"<think>reasoning process here</think> <answer>answer "
"here</answer>. Inside the answer tag, put only "
"the answer and no additional commentary. <|eot id|><|start "
"header id |>user<|end header id|> {question} "
"<|eot id|><|start header id|>assistant<|end header id|>
```

and structure the sequence as: `<think>cot</think> <answer>answer</answer>`, where `cot` is everything that exists in the original MATH sequence before the boxed answer. We drop the `cot` with probability $p_{cot}$.

We use the AdamW (Loshchilov & Hutter, 2019) optimizer with learning rate $\eta_{sft}=10^{-5}$, batch size $B_{sft}=24$ for the 3.2 3B model and $B_{sft}=16$ for the 3.1 8B model, and bf16 precision. We train for 20 epochs. We consider datasets with $p_{cot} \in \{0.1, 0.25, 0.5, 1\}$. We use PyTorch for distributed training. Each run is performed in 8 GPUs (NVIDIA A100-SXM4-80GB) and takes 1.5 hour to complete.

**Post-training**  We train with GRPO with a reward function that has stepwise structure: 5 points for the presence of the answer tag, 5 points for the presence of the thinking tag, 20 points if it contains the correct answer in part of the response, and 100 points for a correct answer at the appropriate place. We found it important using the `math-verify` module of Huggingface for simplifying mathematical expressions and accurately rewarding generations. We use the AdamW (Loshchilov & Hutter, 2019) optimizer with learning rate $\eta_{post}=10^{-5}$ (and Huggingface's default rest of hyperparameters), batch size $B_{post}=1$, group size (for GRPO) equal to 20, 2 steps of gradient accumulation (which makes the effective batch size equal to 2), no KL penalty, $\epsilon_{clip} = 0.2$. We use generation temperature $\tau_{RL} = 1.0$. We perform post-training in 8 GPUs (NVIDIA A100-SXM4-80GB) for 300 steps which concludes in about 14-29 hours.

## C  ADDITIONAL EXPERIMENTAL RESULTS

### C.1  MORE EXPERIMENTS ON MAIN PARITY SETTING

**Model hyperparameters ablation**  We present aggregated results for pre-training and post-training on $\mathcal{D}(p_{cot}), p_{cot}=0.25$ for GPT2 and Mistral architectures of varying embedding dimension and number of heads in Figure 5 (depth $L=2$), Figure 6 (depth $L=4$) and Figure 7 (depth $L=8$). Post-training uses GRPO with end-to-end reward function $r_{e2e}$ and sampling temperature $\tau_{RL}=1.0$ for all configurations. Notice that the Mistral architecture exhibits more unstable learning (pre-training) than GPT2. We observe that pre-training alone does not induce a model capable of consistent generalization in any of the cases. We plot number of training sequences on the x-axis (as opposed to,

---

[8]We split the datasets, since they are of small size and we perform multi-epoch training during SFT.

say, SGD iterations to allow a simpler comparison with post-training). For the number of post-train samples, we take the multiplicity of GRPO samples into consideration.

**Comparison of RL algorithms for more values of $p_{\text{cot}}$**  In Figure 3, we presented test accuracy and response length during post-training for all RL algorithms (STaR, REINFORCE, GRPO) and for a few different values of sampling temperature used for generating training sequences for $p_{\text{cot}}$=0.25. We show now additional results for distributions where data are even more rare ($p_{\text{cot}}$=0.1 and $p_{\text{cot}}$=0.05). In Figure 8, we observe that all RL algorithms induce generalization for most values of sampling temperature (post-training starts after 20k pre-training iterations) when $p_{\text{cot}}$=0.1. On the other hand, when $p_{\text{cot}}$ is even smaller ($p_{\text{cot}}$=0.05), post-training after 20k iterations does not reliably lead to well-generalizing models for any of the RL algorithms (Figure 9). If, instead, we consider more pre-training iterations (50k) before the start of post-training, then we observe greater probability of success with all RL algorithms for reasonable ($\approx$1) sampling temperatures (Figure 10). This illustrates, amongst others, that the learning behaviors of STaR, REINFORCE and GRPO are similar in our simple setting.

**Smaller input dimension**  In Figure 11, we demonstrate that pre-training sometimes leads to a generalizing model if we wait long enough (example seeds shown for $d$=11 and $d$=12). However, as $d$ increases, this waiting time becomes prohibitive.

**Sample complexity gap**  We design an experiment to try estimating the sample complexity gap between the two paradigms (next-token only vs next-token followed by RL). We consider a GPT2 architecture of depth 4, number of heads 32 and embedding dimension equal to 128. We fix $p_{\text{cot}}$=0.25 and we vary the input dimension $d \in \{6, 8, 10, 11, 12\}$. We train for at most 100,000 iterations (of batch size 256) and we consider switching to post-training (GRPO with e2e reward) at several checkpoints. We consider a checkpoint to be successful if its test accuracy is greater than 95%. In Figure 12, we show the gap between the number of samples required for any pre-trained checkpoint to generalize vs the number of samples required by any pre- and post-trained model (combined) for several input dimensions $d$. For the number of post-train samples, we take the multiplicity of GRPO samples into consideration. This gap seems to be growing exponentially fast. Some limitations of this simple experiment: the pre-trained checkpoints from which we consider starting post-training are at iterations $\{$1000, 1500, 2000, 2500, 3000, 5000, 10000, 15000, 20000, 30000, 50000, 100000$\}$. Also, for $d$=12, only one of three seeds ended up generalizing during pre-training (so by definition we could only calculate the gap for that seed and we discarded the rest).

### C.2  PARTIAL CHAIN OF THOUGHT

The purpose of this subsection is to relax our main setting, and consider a more complex variation which might better capture "real-world" data. In particular, we consider a setting, where the data include a third type of sequence consisting of a medium-sized chain of thought, in addition to short and long sequences.

We consider the distribution $\mathcal{D}_3(p_{\text{cot}}, p_{\text{odd}})$ over $\{\pm 1\}^d \times \{\pm 1, \texttt{<EOS>}\}^*$, parameterized by $p_{\text{cot}}, p_{\text{odd}} \in (0, 1)$, such that: $x_1, \ldots, x_d \sim \text{Rad}(1/2)$ and

$$(y_1, \ldots, y_{d+1}) = \begin{cases} \left(x_1, x_1 x_2, \ldots, \prod_{i=1}^d x_i, \texttt{<EOS>}\right), \text{ w.p. } p_{\text{cot}}, \\ \left(x_1, x_1 x_2 x_3, x_1 x_2 x_3 x_4 x_5, \ldots, \prod_{i=1}^d x_i, \texttt{<EOS>}\right), \text{ w.p. } p_{\text{odd}}, \\ \left(\prod_{i=1}^d x_i, \texttt{<EOS>}\right), \text{ w.p. } 1 - p_{\text{cot}} - p_{\text{odd}}. \end{cases} \quad (11)$$

The medium-sized sequence consists of the same random variables as before in input positions 1 to $d$, while the chain of thought skips the intermediate computations that involve partial parities ending at an even position (with the exception of the final parity, if $d$ is even). This results in sequences that contain a *partial* chain of thought with omitted reasoning steps. The purpose of this sequence is to provide the model with some samples of intermediate complexity between the two extremes of $\mathcal{D}(p_{\text{cot}})$. Indeed, learning to predict each term of the chain of thought after the first position requires a *leap* (Abbe et al., 2023) of order 2, as opposed to either leap 1 or $d$ for short and long sequences, respectively. For instance, predicting $x_1 x_2 x_3$ from the history of the sequence $(x_1, \ldots, x_d, x_1)$ is a

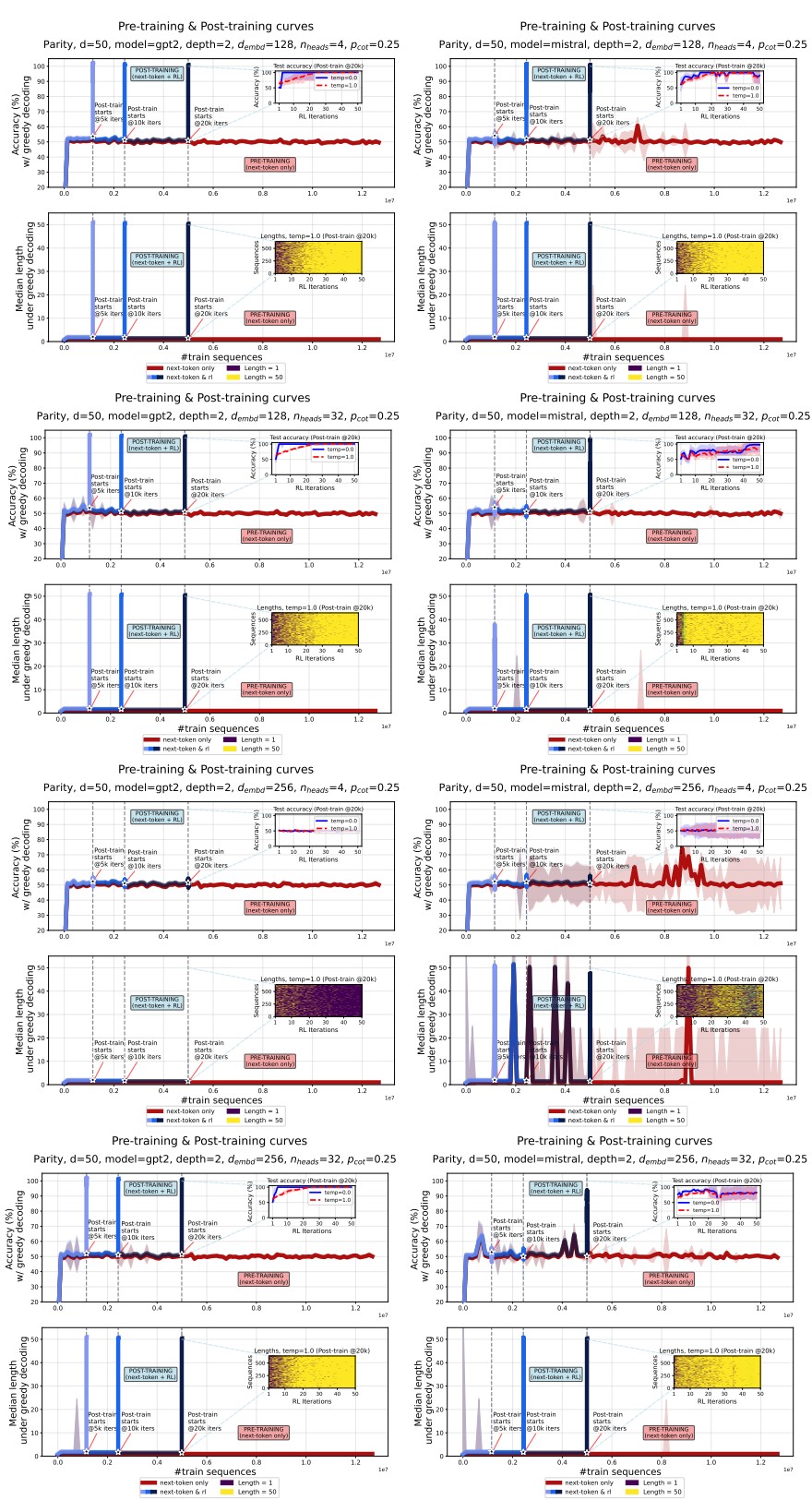

Figure 5: **Pre-training and post-training curves combined on the parity task for GPT2 and Mistral architectures.** We vary the embedding dimension and the number of heads. Depth $L$=2.

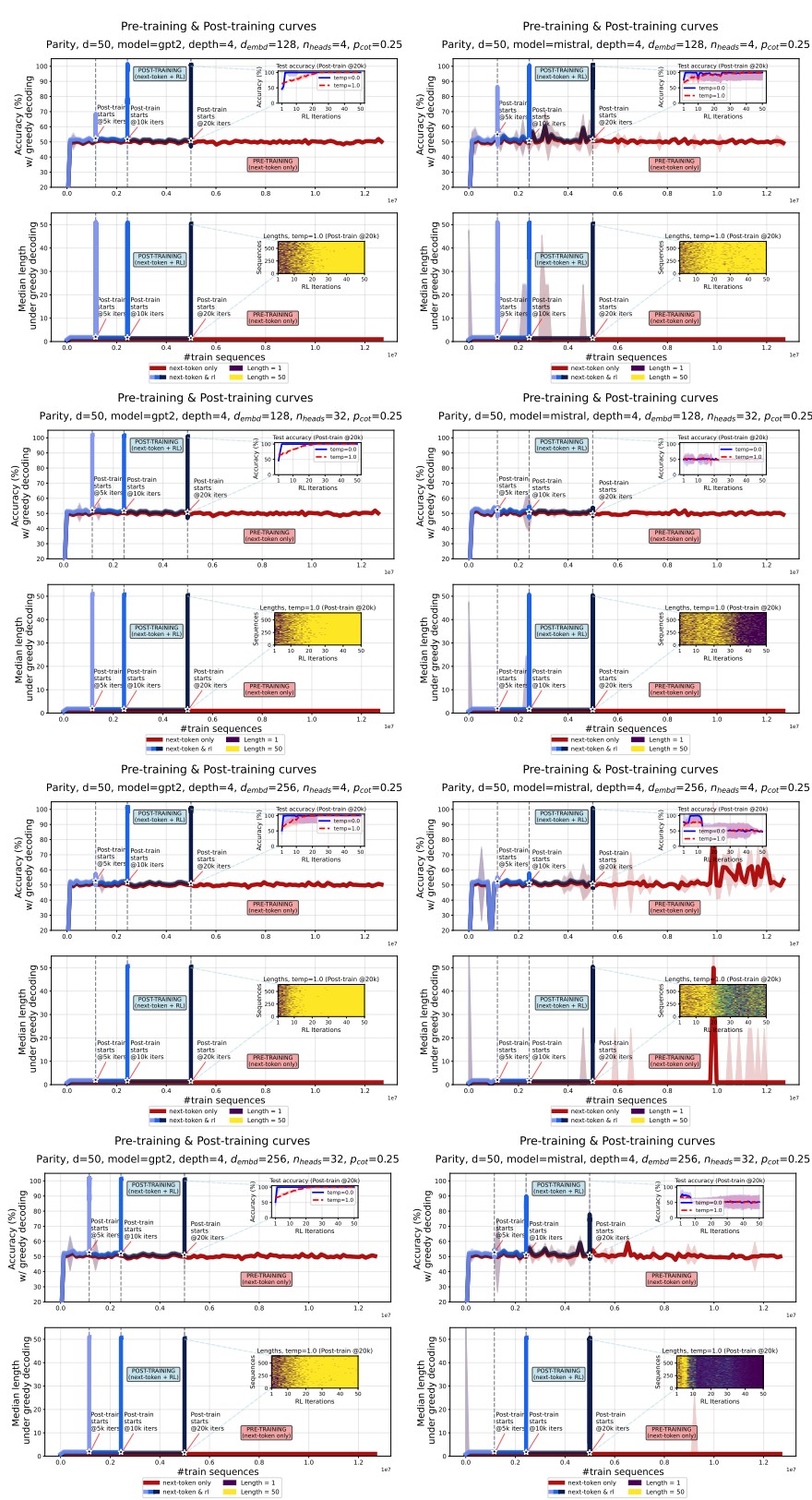

Figure 6: **Pre-training and post-training curves combined on the parity task for GPT2 and Mistral architectures.** We vary the embedding dimension and the number of heads. Depth $L$=4.

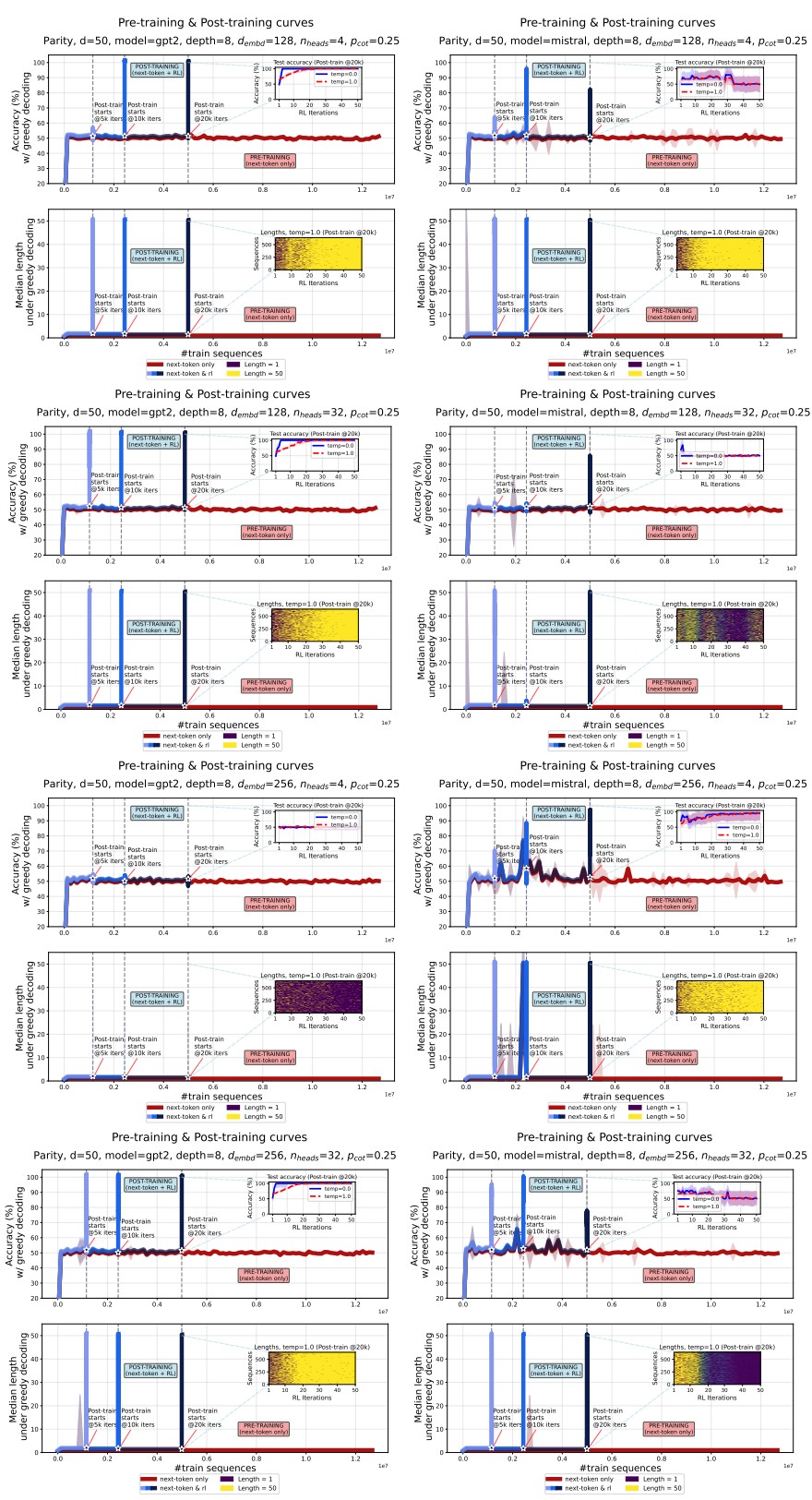

Figure 7: **Pre-training and post-training curves combined on the parity task for GPT2 and Mistral architectures.** We vary the embedding dimension and the number of heads. Depth $L$=8.

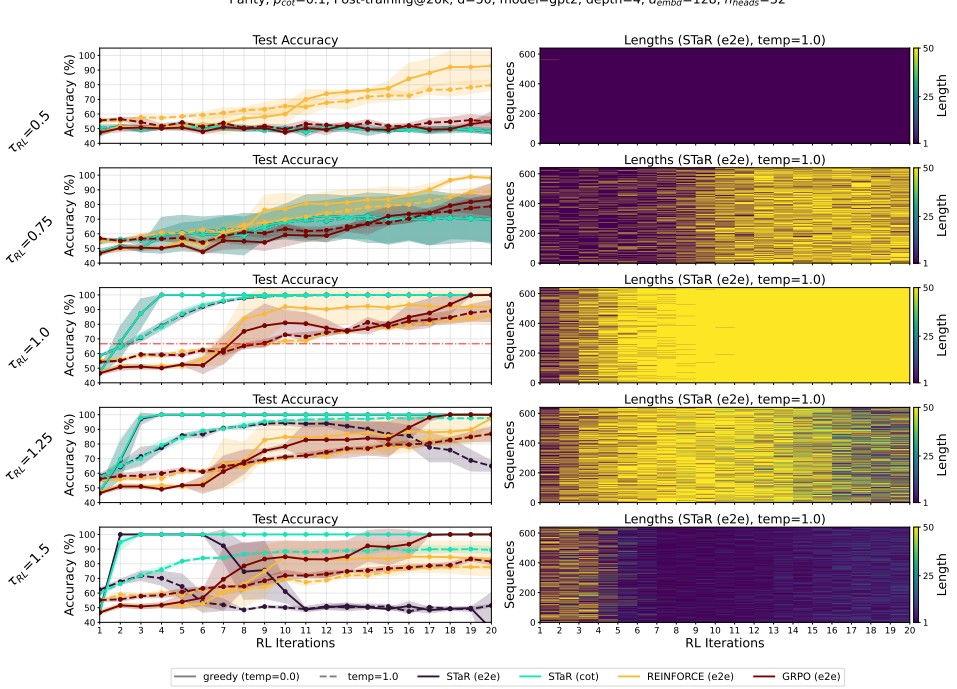

Figure 8: **Post-training of transformers on mixture of long and short sequences encoding the parity of $d$ bits with various RL methods and generation temperatures ($\tau_{\mathrm{RL}}$).** Mixture coefficient: $p_{\mathrm{cot}}$=0.1. Pre-training iterations: 20k.

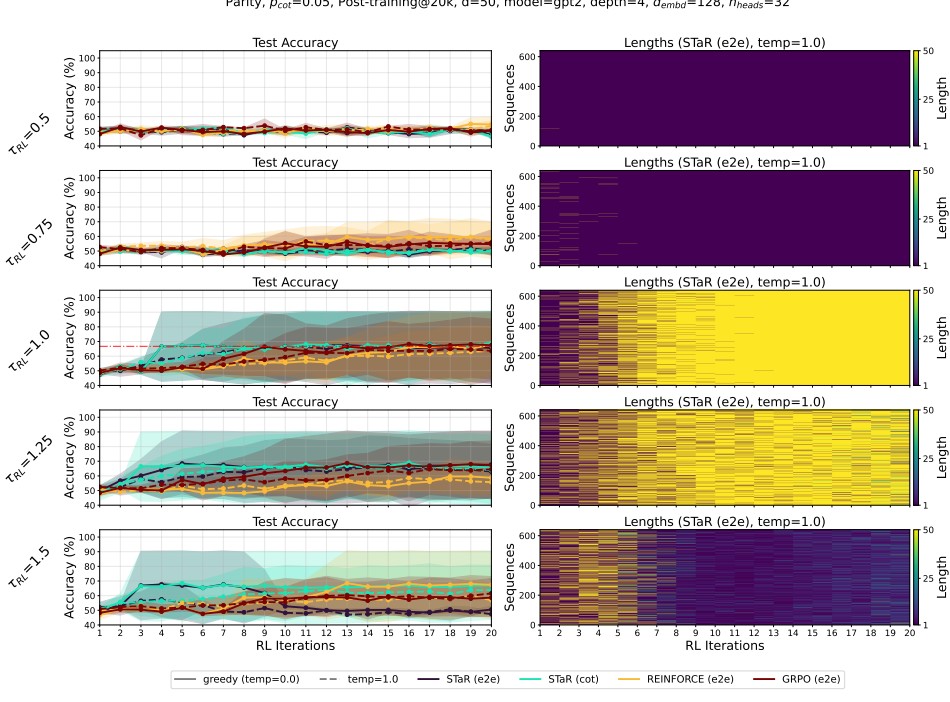

Figure 9: **Post-training of transformers on mixture of long and short sequences encoding the parity of $d$ bits with various RL methods and generation temperatures ($\tau_{\mathrm{RL}}$).** Mixture coefficient: $p_{\mathrm{cot}}$=0.05. Pre-training iterations: 20k.

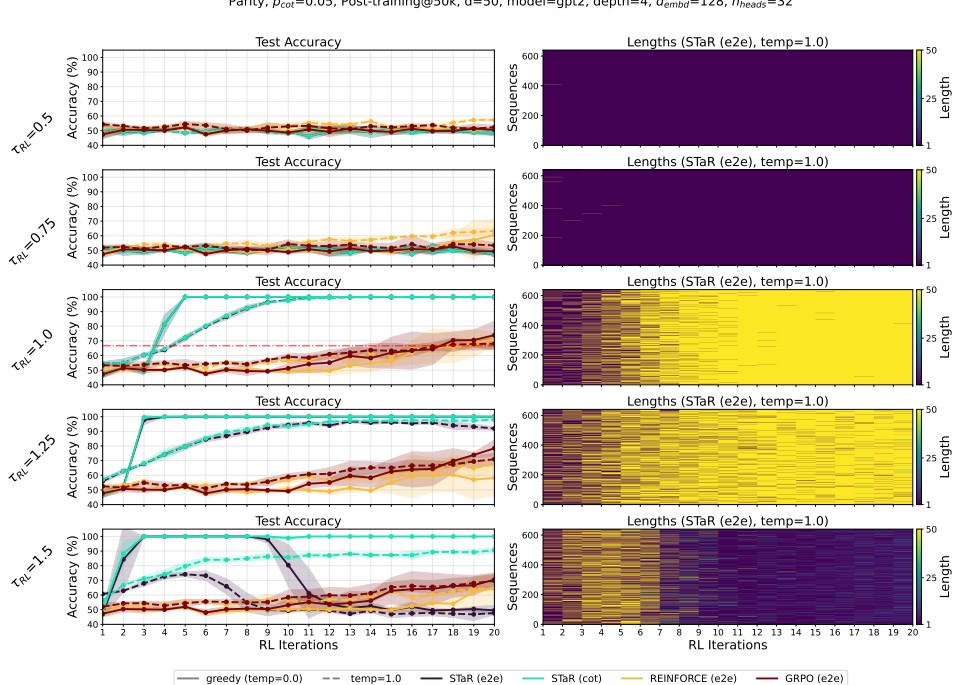

Figure 10: **Post-training of transformers on mixture of long and short sequences encoding the parity of $d$ bits with various RL methods and generation temperatures ($\tau_{\mathrm{RL}}$).** Mixture coefficient: $p_{\mathrm{cot}}$=0.05. Pre-training iterations: 50k.

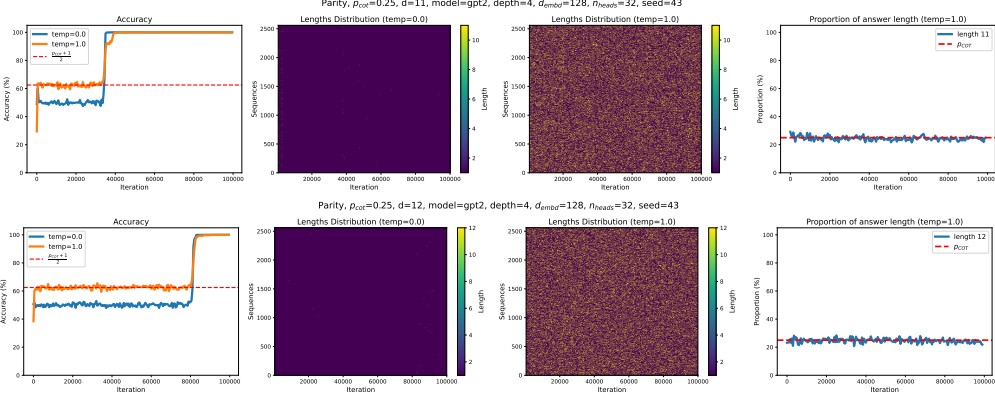

Figure 11: **Pre-training on the parity task leads to a generalizing model after many iterations.** The average response is short (length equal to 1). *Top row*: $d$=11. *Bottom row*: $d$=12.

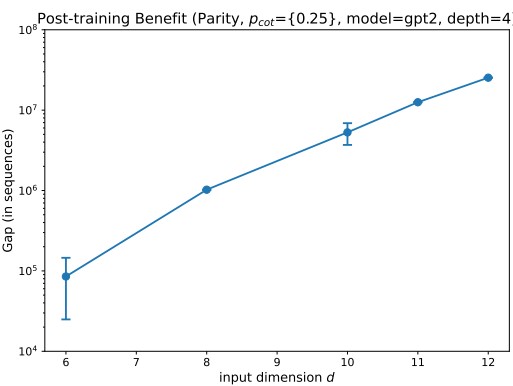

Figure 12: **Sample complexity advantage of pre-training & post-training over mere pre-training vs input dimension.** Note the logarithmic scale in the y-axis. The figure shows mean and one standard deviation across 3 random seeds.

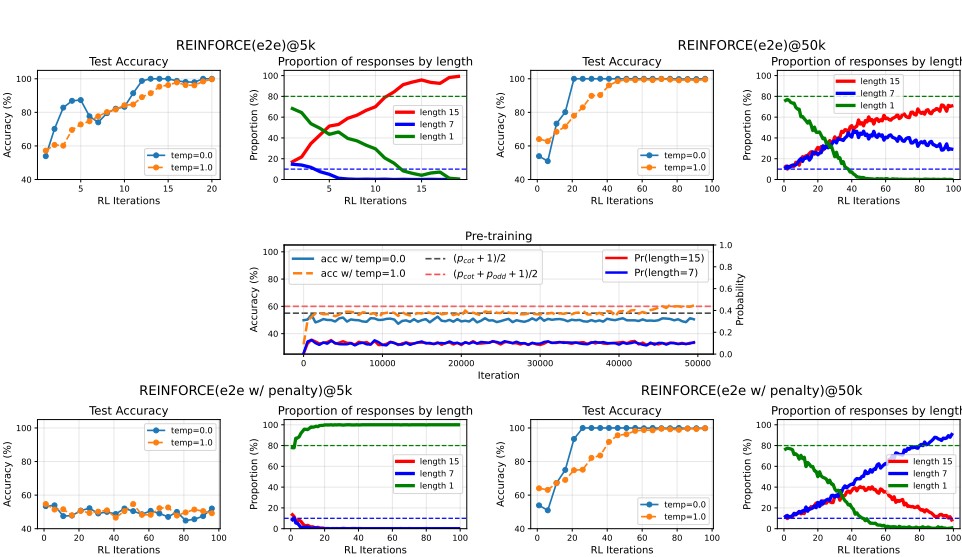

Figure 13: **Parity with partial chain of thought data and effectiveness of length penalties.** Post-training experiments using REINFORCE with (bottom row) and without (top row) a length penalty on pre-trained models that correspond to the training shown in the middle. Notice that leaps in accuracy during pre-training control length distributions at the end of post-training and success of length-penalized RL.

leap of order 2. The leap between terms characterizes the additional sample complexity for learning each (sub)function.

We repeat pre-training and post-training on this distribution and present results in Figure 13 for $d$=15 and $(p_{\text{cot}}, p_{\text{odd}})$=$(0.1, 0.1)$.

**Pre-training staircase** As before, improvement during pre-training happens in stages: early at training, model accuracy under sampling with temperature 1 is around 50%, yet it suddenly jumps to $\frac{p_{\text{cot}}+1}{2} \times 100\%$ as the model learns from the long demonstrations. Later in training, there is another phase transition indicating the model has learned the medium-sized sequences, as the accuracy leaps to $\frac{p_{\text{cot}}+p_{\text{odd}}+1}{2} \times 100\%$. As before, if $p_{\text{cot}}, p_{\text{odd}}$ are not large enough, then the models under greedy decoding do not generalize in reasonable time. In fact, the critical values of $p_{\text{cot}}, p_{\text{odd}}$ can be calculated as in our main setting. It holds:

$$\mathbb{P}\left[y_1 = +1|\mathbf{x}\right] = (p_{\text{cot}} + p_{\text{odd}}) \frac{x_1 + 1}{2} + (1 - p_{\text{cot}} - p_{\text{odd}}) \frac{\prod_{i=1}^d x_i + 1}{2}. \tag{12}$$

In absence of a parity feature, this conditional probability simplifies to (see Section E.2 for details):

$$\mathbb{P}\left[y_1 = +1|\mathbf{x}\right] = \frac{(p_{\text{cot}} + p_{\text{odd}}) x_1 + 1}{2}, \tag{13}$$

or in other words:

$$\mathbb{P}\left[y_1 = x_1|\mathbf{x}\right] = \frac{p_{\text{cot}} + p_{\text{odd}} + 1}{2}. \tag{14}$$

This implies that first token output of a greedy decoder will always be $x_1$, as long as $p_{\text{cot}} + p_{\text{odd}} > 0$. For the second position we have:

$$
\begin{aligned}
\mathbb{P}\left[y_2 = x_1 x_2 | y_1 = x_1, \mathbf{x}\right] &= \frac{\mathbb{P}\left[y_2 = x_1 x_2, y_1 = x_1 | \mathbf{x}\right]}{\mathbb{P}\left[y_1 = x_1 | \mathbf{x}\right]} \\
&= \frac{p_{\text{cot}} + \frac{x_3+1}{2} p_{\text{odd}}}{\frac{p_{\text{cot}}+p_{\text{odd}}+1}{2}} \\
&= \frac{2p_{\text{cot}} + (x_3 + 1) p_{\text{odd}}}{p_{\text{cot}} + p_{\text{odd}} + 1},
\end{aligned}
\tag{15}
$$

and likewise:

$$\mathbb{P}\left[y_2 = x_1 x_2 x_3 | y_1 = x_1, \mathbf{x}\right] = \frac{2p_{\text{odd}} + (x_3 + 1) p_{\text{cot}}}{p_{\text{cot}} + p_{\text{odd}} + 1}, \tag{16}$$

$$\mathbb{P}\left[y_2 = \texttt{<EOS>} | y_1 = x_1, \mathbf{x}\right] = \frac{1 - (x_3 + 2)(p_{\text{cot}} + p_{\text{odd}})}{p_{\text{cot}} + p_{\text{odd}} + 1}. \tag{17}$$

Thus, a model under greedy decoding will be stopping generation at the second token as long as:

$$1 - (x_3 + 2)(p_{\text{cot}} + p_{\text{odd}}) \geq \max\left(2p_{\text{cot}} + (x_3 + 1) p_{\text{odd}}, 2p_{\text{odd}} + (x_3 + 1) p_{\text{cot}}\right). \tag{18}$$

This certainly shows that a model trained on a distribution endowed with additional "partial" chain of thought data can generate long responses and generalize during pre-training already, even if $p_{\text{cot}} < 1/3$ which was the critical threshold in our main setting. Such an example is presented in Figure 14 for $(p_{\text{cot}}, p_{\text{odd}})$=$(0.1, 0.35)$.

**Pre-training checkpoints & length penalties** In this setting, the point when we switch to post-training, can not only affect the eventual success of RL, but also the length distribution of the model responses at the end of it:

- Post-training at 5k iterations yields a model that generalizes, yet it learns to almost exclusively generate long responses (Figure 13, top left).
- Post-training at 50k iterations, on the other hand, results in a model that not only manages to reach 100% test accuracy but also has more diversity in its length distribution, generating both long and medium-sized responses (Figure 13, top right).

Figure 14: **Pre-training on the parity task with "partial" chain of thought data leads to a generalizing model even for** $p_{\text{cot}} < 1/3$**.** The average response length is "medium".

The previous observation is both good and bad news: good news, as the 50k checkpoint ends up being cheaper at inference than the post-trained 5k counterpart; but bad news, as the model could have been even faster at inference without sacrificing accuracy. Indeed, after 50k pre-training iterations the model has learned to generate correct medium-sized responses, hence, in principle, it need not learn from the long self-generated ones. To test this, we consider reinforcement learning with a length-penalized reward. In particular, we consider the following length-penalized reward:

$$r_{\text{e2e},\lambda_{\text{len}}}(\mathbf{x}, y) = \mathbb{1}\left\{y[-1] = \prod_{i=1}^{d} x_i\right\} - \lambda_{\text{len}}\frac{|y|}{d}, \ \lambda_{\text{len}} \in [0, 1]. \tag{19}$$

We use $\lambda_{\text{len}}$=0.4. As we confirm in Figure 13 (bottom row), a length penalty applied during post-training of a late checkpoint is effective, as it leads to a model that is both accurate and fast. On the contrary, if we apply the same length penalty at an earlier checkpoint, post-training fails. This is because the pre-trained model has not managed to learn from the medium-sized sequences early in pre-training.

## C.3 NOISY DATA

In this subsection, we introduce and study variations of our main setting in which noise may be present in the data. Specifically, we consider cases where the noise affects the pre-training distribution. We observe that many of the previously observed learning phenomena continue to hold in the presence of moderate amounts of noise, even in cases that noise seems to be "dominating" pre-training performance.

We consider two different noise models.

**Final token noise** In this case, the final token is flipped with probability $\eta \in (0, 1)$. That is, $x_1, \ldots, x_d \sim \text{Rad}(1/2)$ and

$$(y_1, \ldots, y_{d+1}) = \begin{cases} \left(x_1, x_1 x_2, \ldots, \prod_{i=1}^{d-1} x_i, \xi_d, \texttt{<EOS>}\right), \text{ w.p. } p_{\text{cot}}, \\ (\xi_d, \texttt{<EOS>}), \text{ w.p. } 1 - p_{\text{cot}}, \end{cases} \tag{20}$$

where $\xi_d = \begin{cases} \prod_{i=1}^{d} x_i, \text{ w.p. } 1 - \eta, \\ -\prod_{i=1}^{d} x_i, \text{ w.p. } \eta. \end{cases}$ . For this type of noise, we do not anticipate a change in the critical threshold of pre-training, since there is no noise in the first token of the long reasoning chain (and in absence of a parity feature, the noise in the short path does not affect things). On the other hand, the pre-training accuracy under sampling with temperature 1 is expected to change to $(1 - \eta)p_{\text{cot}} + \frac{1 - p_{\text{cot}}}{2} = \frac{(1 - 2\eta)p_{\text{cot}} + 1}{2}$. We present pre-training simulations that confirm the above predictions for $\eta \in \{0.1, 0.2\}$ in Figures 15 and 16.

This small change in the pre-training accuracy does not alter significantly the post-training behaviour. Indeed, pre-training noise of this type only affects the starting point of the model during reinforcement learning, and as a result it does not affect significantly the sample complexity of post-training. Experimental results for $p_{\text{cot}}$=0.25 and $\eta \in \{0.1, 0.2\}$ can be found in Figures 17 and 18, respectively.

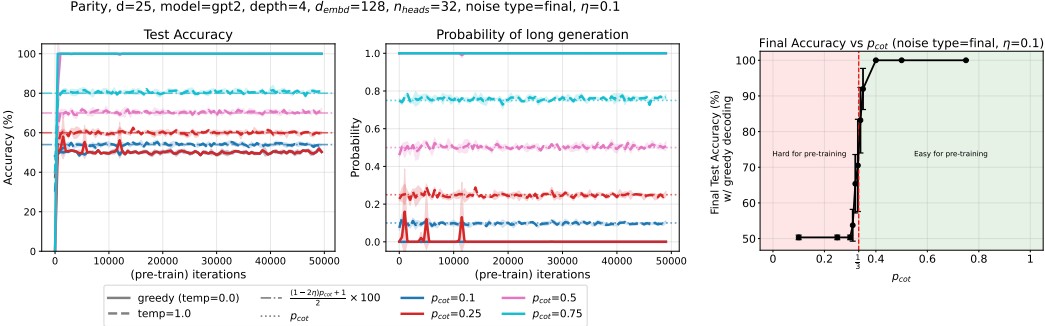

Figure 15: **Pre-training of transformers on mixture of long and short sequences encoding the parity of** $d$ **bits under "final" type noise for** $\eta$**=0.1**. *Left*: Test accuracy during the course of pre-training. *Center*: The probability that a model's generation length is equal to the maximum length present in the training distribution (which equals $d$). Solid lines correspond to greedy decoding, while dashed lines correspond to sampling with temperature 1. Each color denotes a training distribution with a separate mixture coefficient $p_{\mathrm{cot}}$. Figure shows average and 1 standard deviation across 3 seeds. Note the change in the legend from the noise-less figures. *Right*: Test accuracy with greedy decoding at the end of pre-training (50k iterations) versus mixture coefficient $p_{\mathrm{cot}}$. The red dashed line corresponds to the critical threshold. Each bullet is the median of 3 runs.

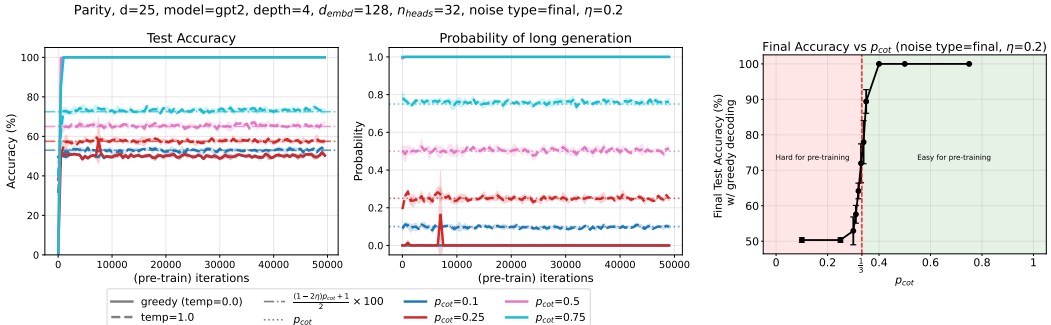

Figure 16: **Pre-training of transformers on mixture of long and short sequences encoding the parity of** $d$ **bits under "final" type noise for** $\eta$**=0.2**. *Left*: Test accuracy during the course of pre-training. *Center*: The probability that a model's generation length is equal to the maximum length present in the training distribution (which equals $d$). Solid lines correspond to greedy decoding, while dashed lines correspond to sampling with temperature 1. Each color denotes a training distribution with a separate mixture coefficient $p_{\mathrm{cot}}$. Figure shows average and 1 standard deviation across 3 seeds. Note the change in the legend from the noise-less figures. *Right*: Test accuracy with greedy decoding at the end of pre-training (50k iterations) versus mixture coefficient $p_{\mathrm{cot}}$. The red dashed line corresponds to the critical threshold. Each bullet is the median of 3 runs.

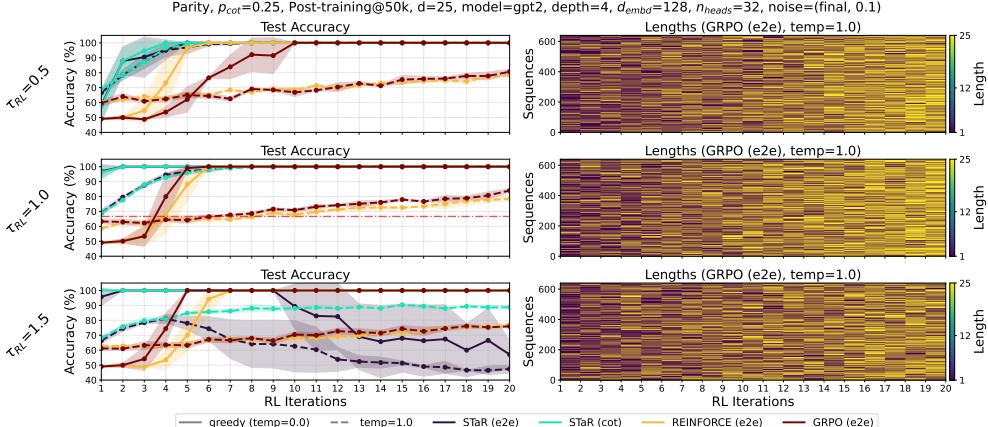

Figure 17: **Post-training of transformers on mixture of long and short sequences encoding the parity of** $d$ **bits under "final" type noise ($\eta$=0.1) with various RL methods and generation temperatures ($\tau_{\mathrm{RL}}$)**. *Left*: Test accuracy during the course of post-training with greedy decoding (solid lines) and sampling with temperature 1 (dashed lines) for various RL methods. Figure shows average and 1 standard deviation across 3 seeds. *Right*: Length of generated response (sampled with temperature 1) during the course of a post-training run (GRPO with end-to-end reward) for 640 test inputs, after 20k pre-training iterations. **Note:** The sample size $n$ of each RL iteration differs amongst the RL algorithms: $n$=64 for GRPO, REINFORCE and $n$=3,200 sequences for STaR.

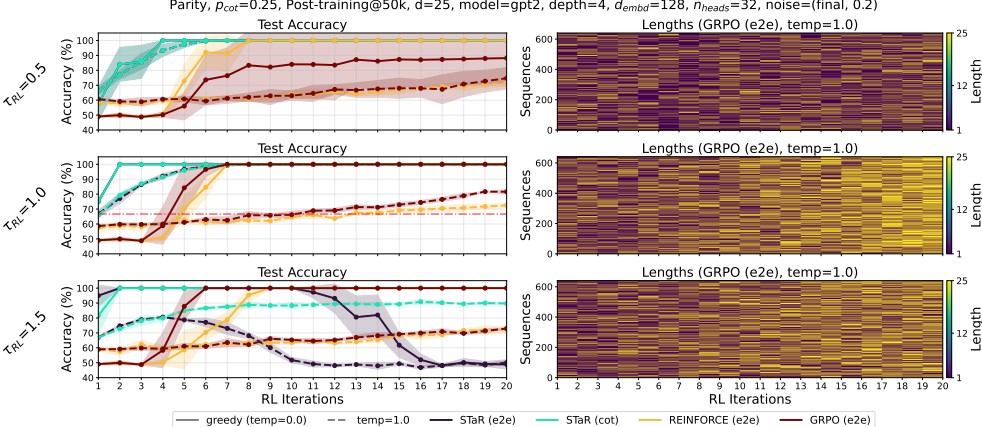

Figure 18: **Post-training of transformers on mixture of long and short sequences encoding the parity of** $d$ **bits under "final" type noise ($\eta$=0.2) with various RL methods and generation temperatures ($\tau_{\mathrm{RL}}$)**. *Left*: Test accuracy during the course of post-training with greedy decoding (solid lines) and sampling with temperature 1 (dashed lines) for various RL methods. Figure shows average and 1 standard deviation across 3 seeds. *Right*: Length of generated response (sampled with temperature 1) during the course of a post-training run (GRPO with end-to-end reward) for 640 test inputs, after 20k pre-training iterations. **Note:** The sample size $n$ of each RL iteration differs amongst the RL algorithms: $n$=64 for GRPO, REINFORCE and $n$=3,200 sequences for STaR.

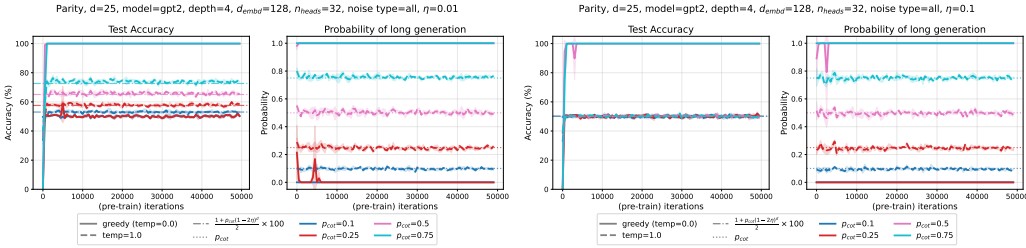

Figure 19: **Pre-training of transformers on mixture of long and short sequences encoding the parity of** $d$ **bits under "all" type noise for** $\eta$**=0.01 and** $\eta$**=0.1**. *Left*: Test accuracy during the course of pre-training. *Center*: The probability that a model's generation length is equal to the maximum length present in the training distribution (which equals $d$). Solid lines correspond to greedy decoding, while dashed lines correspond to sampling with temperature 1. Each color denotes a training distribution with a separate mixture coefficient $p_{\text{cot}}$. Figure shows average and 1 standard deviation across 3 seeds. Note the change in the legend from the noise-less figures. *Right*: Test accuracy with greedy decoding at the end of pre-training (50k iterations) versus mixture coefficient $p_{\text{cot}}$. The red dashed line corresponds to the critical threshold. Each bullet is the median of 3 runs.

**All tokens noise** In this case, there is noise in each token of the output that gets amplified ("snow-balls") in the case of long reasoning chains. That is, $x_1, \ldots, x_d \sim \text{Rad}(1/2)$ and

$$(y_1, \ldots, y_{d+1}) = \begin{cases} (\xi_1, \xi_2, \ldots, \xi_d, \texttt{<EOS>}), \text{ w.p. } p_{\text{cot}}, \\ (\xi_d, \texttt{<EOS>}), \text{ w.p. } 1 - p_{\text{cot}}, \end{cases} \tag{21}$$

where $\xi_1 = \begin{cases} x_1, \text{ w.p. } 1 - \eta, \\ -x_1, \text{ w.p. } \eta. \end{cases}$ and $\xi_i = \begin{cases} x_i \xi_{i-1}, \text{ w.p. } 1 - \eta, \\ -x_i \xi_{i-1}, \text{ w.p. } \eta. \end{cases}$ for $i > 1$. It is clear that this noise disproportionately affects the long reasoning chains. The critical threshold of pre-training here will change, as the noise affects the distribution of the first and second output tokens. Additionally, the pre-training accuracy under sampling with temperature 1 will now equal $p_{\text{cot}}\mathbb{P}\left[\xi_d = \prod_{i=1}^d x_i\right] + \frac{1-p_{\text{cot}}}{2} = p_{\text{cot}}\mathbb{P}\left[\text{even sucs in d trials}\right] + \frac{1-p_{\text{cot}}}{2} = p_{\text{cot}}\frac{1+(1-2\eta)^d}{2} + \frac{1-p_{\text{cot}}}{2} = \frac{(1-2\eta)^d p_{\text{cot}}+1}{2}$. We present pre-training simulations in Figure 19 for $\eta \in \{0.01, 0.1\}$. We confirm the analytical calculation for the pre-training accuracy under sampling, while we observe that when the noise level is moderate ($\eta$=0.1) pre-training seems to be failing to induce any correlation with the target. However, and perhaps surprisingly, post-training with reinforcement learning can still lead to generalizing models. Post-training experimental results for $p_{\text{cot}}$=0.25 and $\eta \in \{0.01, 0.1\}$ can be found in Figures 20 and 21, respectively.

In both cases, we observe that there exists a combination of post-training algorithm and sampling temperature that yields a generalizing model. We observe a moderate increase in the sample complexity in comparison to the noise-less setting, and a lower tolerance to large sampling temperatures, even with "perfect" (i.e., $r_{\text{cot}}$) verification. We defer a theoretical understanding of these interesting phenomena to future study.

## C.4 NUMBERS MULTIPLICATION

Here, we describe the setting and our results on the task of multiplication in more detail.

It has been reported that even LLMs as capable as GPT4 (Achiam et al., 2023) have struggled with accurate multiplication of 4-digit numbers (Shen et al., 2023) without the use of an external calculator program[9]. We encode the task in sequences of characters, including the n digits of the multiplier and multiplicand (both in reversed order), the execution trace of the grade-school multiplication algorithm, and the final answer. Prior work has reported that reversing the input order helps transformers with absolute positional embeddings generalize better on arithmetic tasks (Shen et al., 2023). As stated in the main text, we construct datasets that contain the full sequence, including the

---

[9]Otherwise, the problem reduces to copying input (from the calculator program) to output and transformers are well-suited for it (Jelassi et al., 2024).

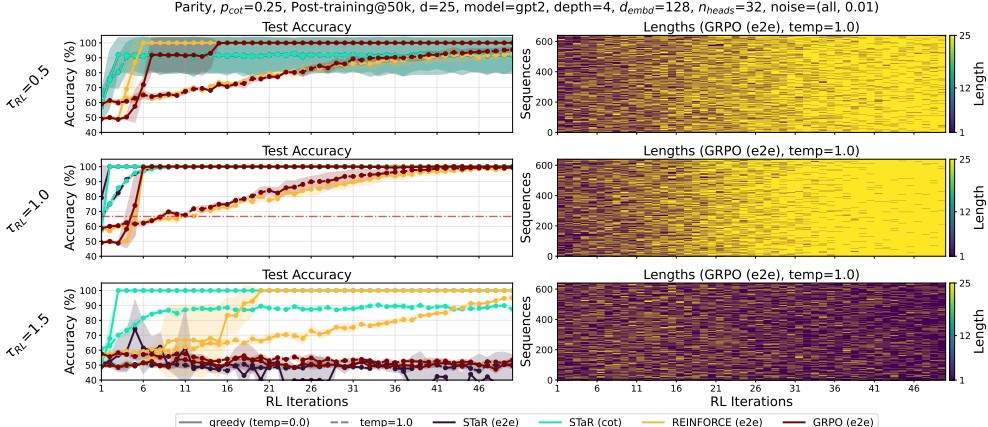

Figure 20: **Post-training of transformers on mixture of long and short sequences encoding the parity of** $d$ **bits under "final" type noise (**$\eta$**=0.01) with various RL methods and generation temperatures (**$\tau_{\mathrm{RL}}$**)**. *Left*: Test accuracy during the course of post-training with greedy decoding (solid lines) and sampling with temperature 1 (dashed lines) for various RL methods. Figure shows average and 1 standard deviation across 3 seeds. *Right*: Length of generated response (sampled with temperature 1) during the course of a post-training run (GRPO with end-to-end reward) for 640 test inputs, after 20k pre-training iterations. **Note:** The sample size $n$ of each RL iteration differs amongst the RL algorithms: $n$=64 for GRPO, REINFORCE and $n$=3,200 sequences for STaR.

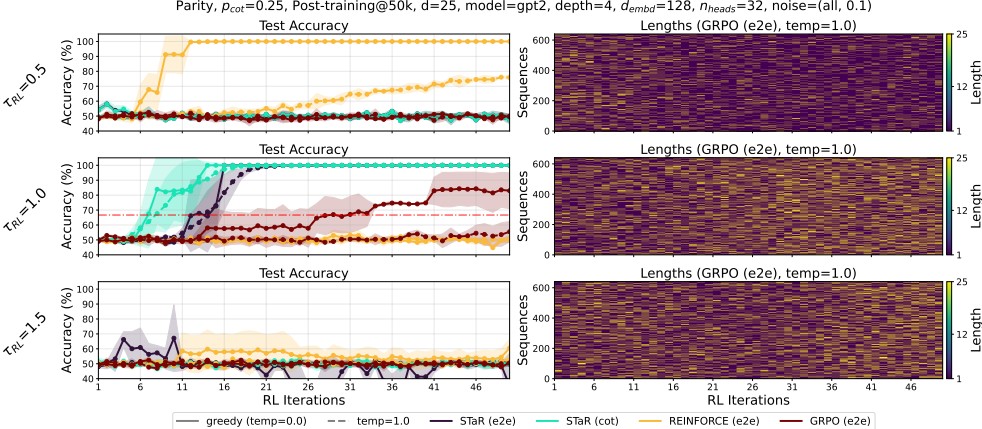

Figure 21: **Post-training of transformers on mixture of long and short sequences encoding the parity of** $d$ **bits under "all" type noise (**$\eta$**=0.1) with various RL methods and generation temperatures (**$\tau_{\mathrm{RL}}$**)**. *Left*: Test accuracy during the course of post-training with greedy decoding (solid lines) and sampling with temperature 1 (dashed lines) for various RL methods. Figure shows average and 1 standard deviation across 3 seeds. *Right*: Length of generated response (sampled with temperature 1) during the course of a post-training run (GRPO with end-to-end reward) for 640 test inputs, after 20k pre-training iterations. **Note:** The sample size $n$ of each RL iteration differs amongst the RL algorithms: $n$=64 for GRPO, REINFORCE and $n$=3,200 sequences for STaR.

chain of thought, with probability $p_{\text{cot}}$ and just the problem and answer with probability $1 - p_{\text{cot}}$ for various values of $p_{\text{cot}}$ – see Figure 22.

We pre-train randomly initialized `GPT2` transformers on either $4 \times 4$, $5 \times 5$ or $7 \times 7$ datasets for multiple epochs over a training set of 808k examples using the next-token prediction objective. At various checkpoints, we switch to reinforcement learning (with GRPO) and compare performance during pre-training vs post-training. Full experimental details are provided in Appendix B.2.

Figures 23 to 25 show pre-training results for all values of $n \in \{4, 5, 7\}$ and $p_{\text{cot}} \in \{0, 0.1, 0.25, 0.5, 1.0\}$. When $p_{\text{cot}}$=1.0 (i.e., the algorithm's trace is included in all samples), models during pre-training quickly learn to answer correctly with long responses, leveraging the intermediate calculations present in the training data. By contrast, when $p_{\text{cot}}$ is small, models struggle to generalize: performance under greedy decoding remains close to 0% throughout training. In some runs, we observe sudden increase in accuracy under sampling with temperature 1.0, jumping to approximately $p_{\text{cot}} \times 100\%$. This outcome is consistent with the parity experiments: the objective enforces length calibration and, once the model learns from long sequences, its output becomes a mixture of short random guesses (with probability of success $10^{-2 \times n}$) and long correct responses (with probability of success 1). Similar to the parity experiments, we observed unstable behavior under greedy decoding when training with $p_{\text{cot}}$ close to the critical threshold (which is close to 0.5 here).

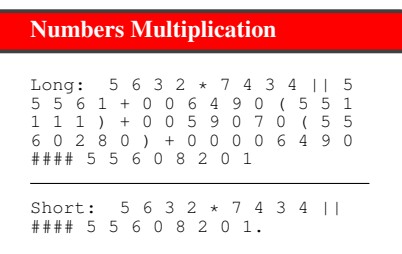

Figure 22: Example of a 4x4 sequence, encoding 2365*4374 (digits appear in reverse order). Top row: long format with full sequence. Bottom row: short format without chain of thought.

After switching to reinforcement learning (Figure 4 (left) for (n=5, $p_{\text{cot}}$=0.25) and Figure 26 for the rest of the values), the model's accuracy improves rapidly and the average response length increases, provided the pre-training checkpoint has developed some correlation with the target. These observations hold for most values of n and $p_{\text{cot}}$. For the most challenging setting we considered, $(n, p_{\text{cot}}) = (7, 0.1)$, the model failed to learn from long sequences under any of the 3 random seeds, even after 38 GPU hours, preventing post-training from being successful in this case. Note that this section provides an example of a task where the capability obtained during RL (accuracy under greedy decoding on task of multiplication) appears to be almost absent during pre-training (accuracy goes from $\approx 0$ to $100\%$) – for example, when $p_{\text{cot}}$=0.1.

**Miscellaneous observations**   Notice that the length of the response is very large for all runs at initialization, as the model has not learned to generate the end-of-sequence token. For the $4 \times 4$ task, we observe success after RL for at least one checkpoint for all values of $p_{\text{cot}}$. If post-training starts early, the model is not able to benefit from reinforcement learning. As an anecdote, some of the early checkpoints in $p_{\text{cot}} = 0.5$ (top left corner) seemingly failed to yield a generalizing model, but we found that this was actually due to a slight mismatch between our evaluation protocol and the reward function; the model did learn to generate correct answers to the multiplication questions but did not learn to end the generation afterward, generating tokens until the maximum limit was exceeded (indeed, see in the plot that the average length is much larger than in the rest of the checkpoints). The evaluation code, however, deemed such responses incorrect, and thus RL was recorded as a failure. This illustrates the various ways that RL might not go as planned if applied to an "immature" checkpoint.

## C.5   MATH BENCHMARK

MATH results with the instruct version of Llama 3.1 8B and Llama 3.2 3B models are presented in Figures 28 and 29. As in the GSM8k experiments, we observe that SFT on a dataset with small $p_{\text{cot}}$ does not enable the model to generalize as well as SFT with full chain-of-thought data. During RL, checkpoints with smaller $p_{\text{cot}}$ benefit the most in terms of relative performance gains, which are accompanied by an increase in response length. However, we also find quantitative evidence that this improvement is sometimes (Figures 34 and 35) (but not always Figures 36 to 39) due to out-of-distribution gains, as model completions can differ significantly from the SFT dataset. We believe

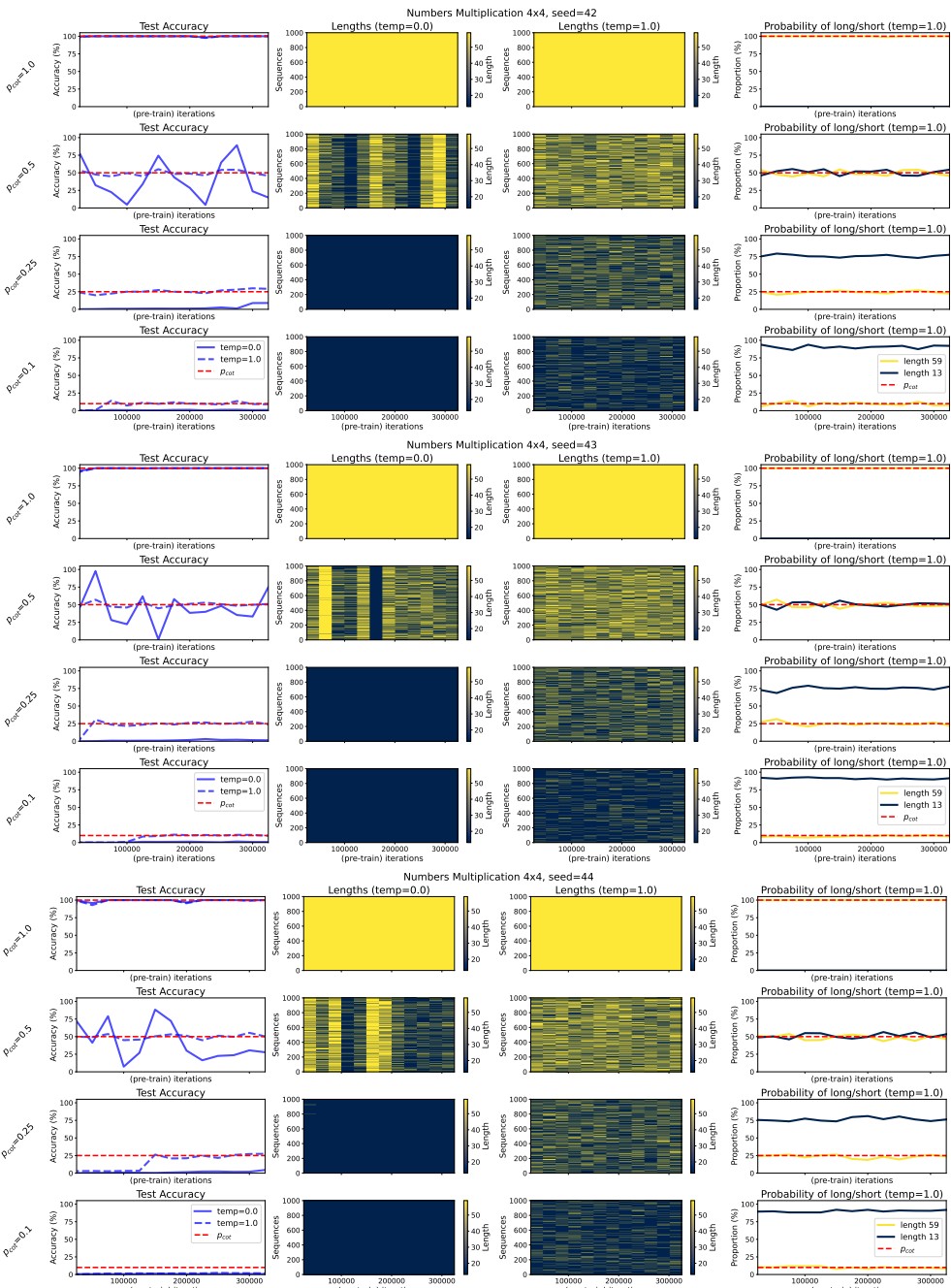

Figure 23: **Pre-training of GPT2 transformers on mixture of long and short sequences encoding the multiplication of 4-digits numbers for 3 different seeds and for various values of** $p_{\mathrm{cot}}$. *Left*: Test accuracy during the course of pre-training, under greedy decoding and sampling with temperature 1. *Center, Left*: The length of the greedy response for 1000 test samples during the course of pre-training. *Center, Right*: The length of the sampled response (temperature of 1) for 1000 test samples during the course of pre-training. **Right**: The probability that the length of a model's autoregressive generation equals the maximum or minimum length present in the training distribution.

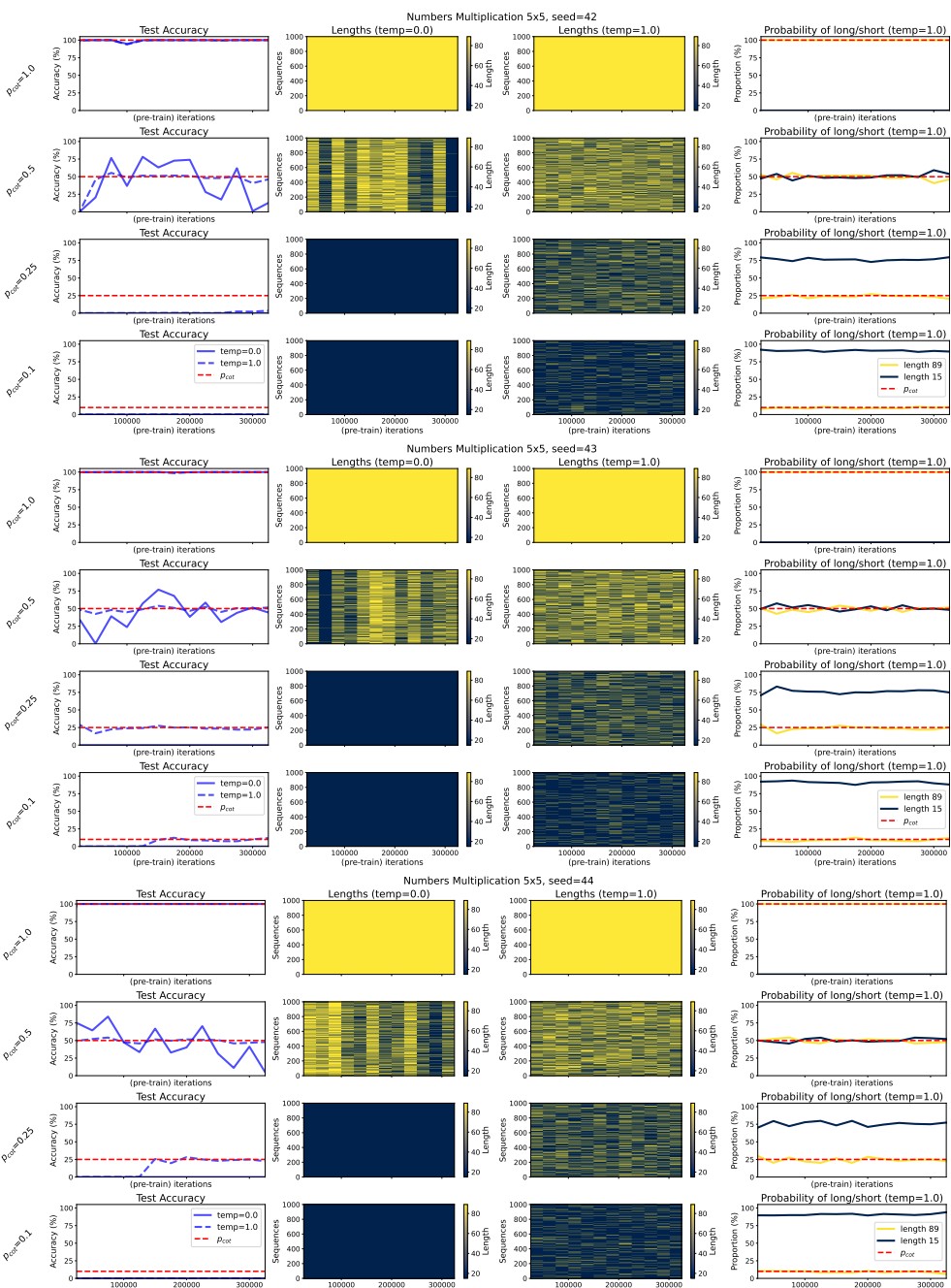

Figure 24: **Pre-training of GPT2 transformers on mixture of long and short sequences encoding the multiplication of 5-digits numbers for 3 different seeds and for various values of** $p_{\mathrm{cot}}$. *Left*: Test accuracy during the course of pre-training, under greedy decoding and sampling with temperature 1. *Center, Left*: The length of the greedy response for 1000 test samples during the course of pre-training. *Center, Right*: The length of the sampled response (temperature of 1) for 1000 test samples during the course of pre-training. **Right**: The probability that the length of a model's autoregressive generation equals the maximum or minimum length present in the training distribution.

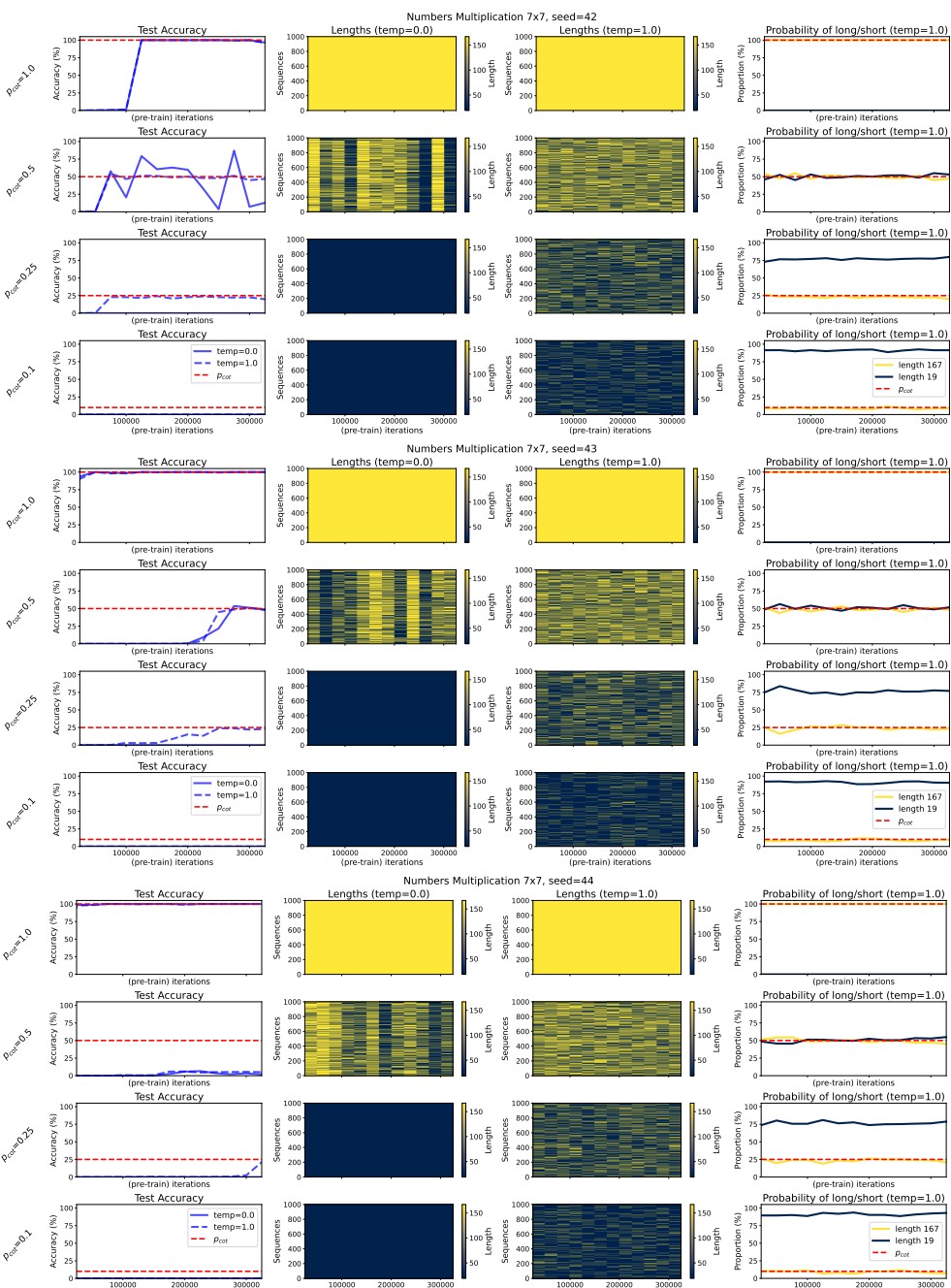

Figure 25: **Pre-training of GPT2 transformers on mixture of long and short sequences encoding the multiplication of 7-digits numbers for 3 different seeds and for various values of** $p_{\text{cot}}$. *Left*: Test accuracy during the course of pre-training, under greedy decoding and sampling with temperature 1 . *Center, Left*: The length of the greedy response for 1000 test samples during the course of pre-training. *Center, Right*: The length of the sampled response (temperature of 1) for 1000 test samples during the course of pre-training. **Right**: The probability that the length of a model's autoregressive generation equals the maximum or minimum length present in the training distribution.

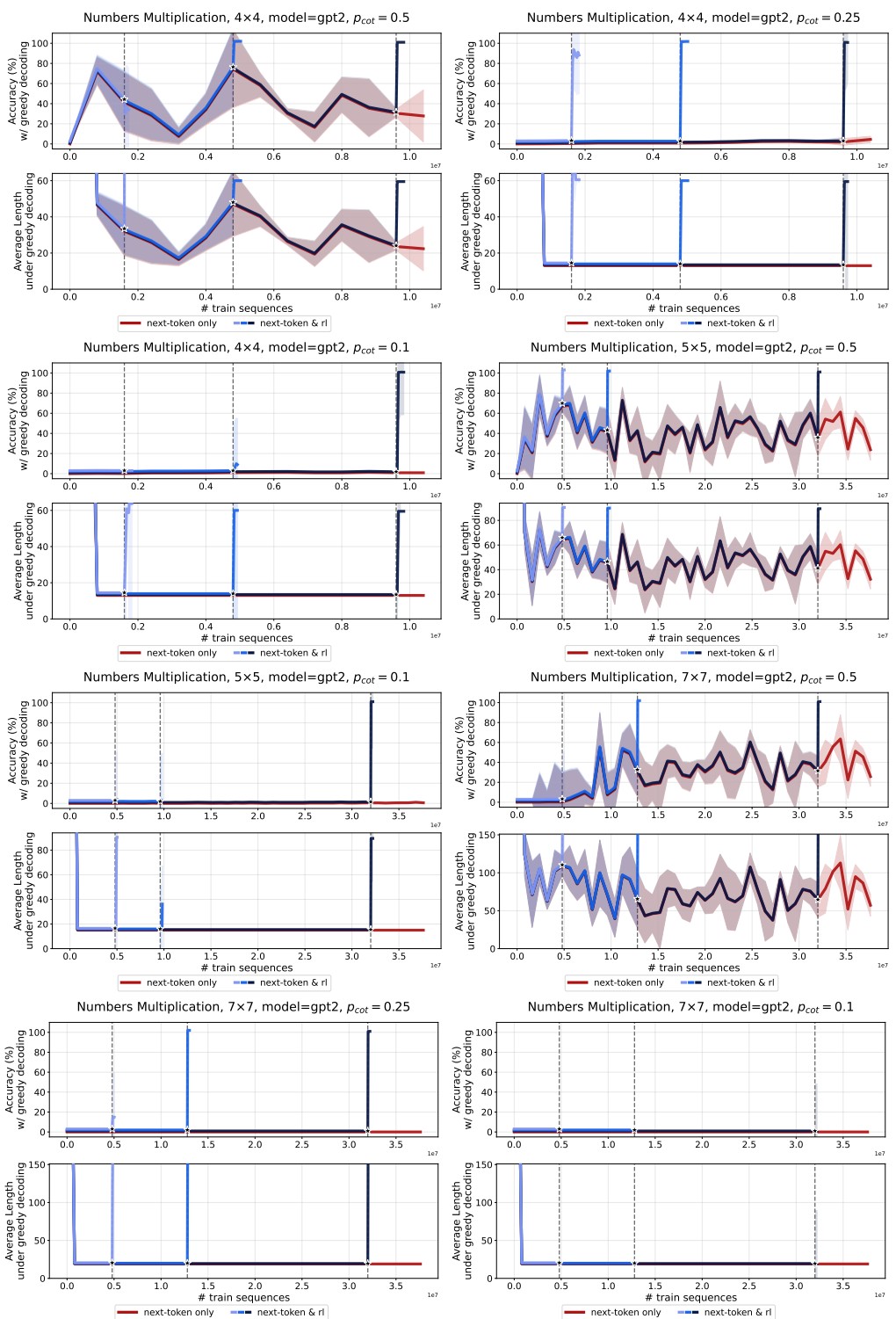

Figure 26: **Pre-training and post-training combined of a randomly initialized GPT2 on numbers multiplication** for various values of numbers of digits $n$ and fraction of chain of thought data $p_{\text{cot}}$ in the mix.

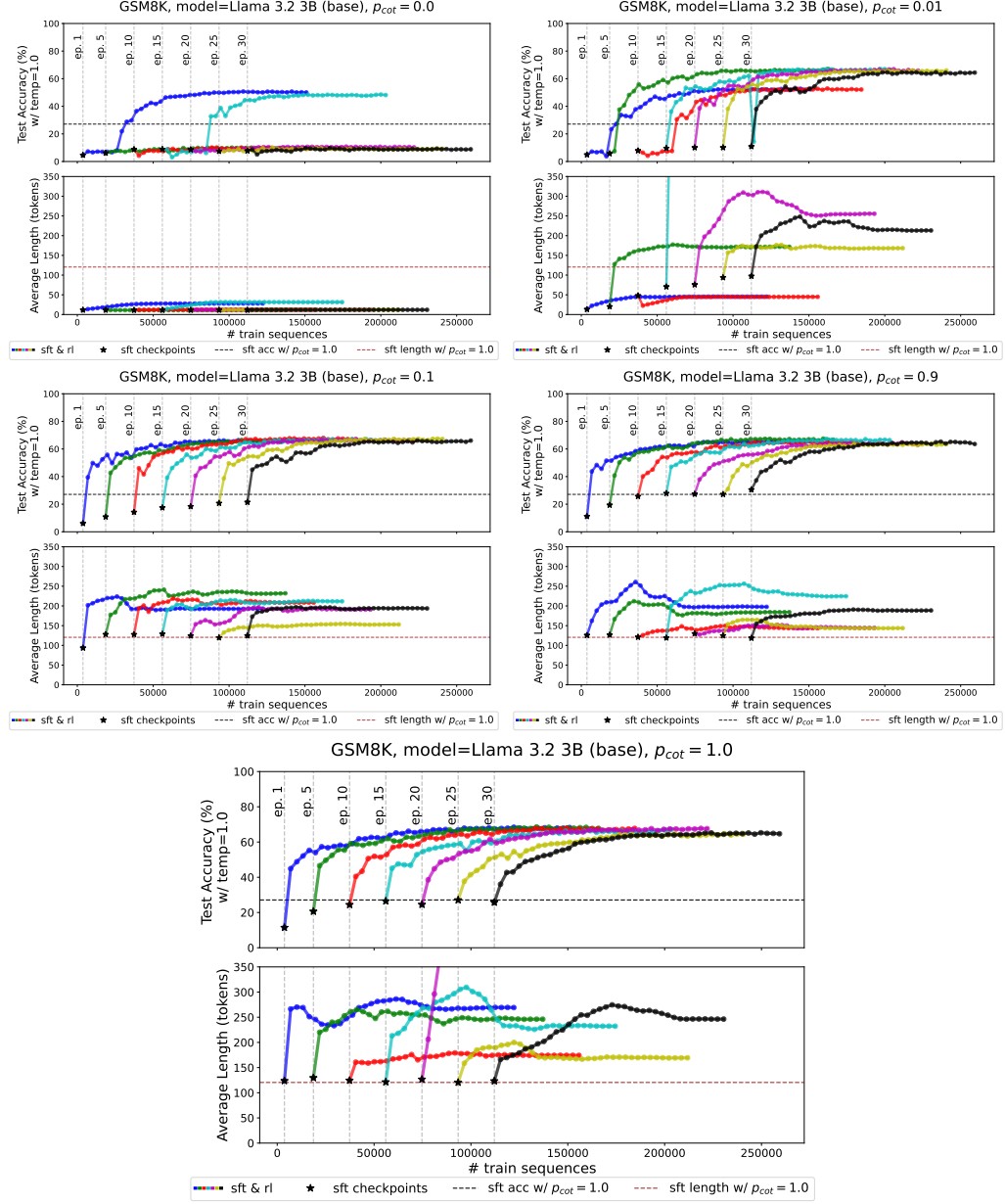

Figure 27: **SFT and GRPO of Llama 3.2 3B (base) on the GSM8k dataset** for various values of $p_{\text{cot}}$. Some observations: For $p_{\text{cot}}$=0, only two checkpoints succeed to generalize during RL. Interestingly, the length does not grow in these cases. The model obviously leverages some kind of pre-training data. For $p_{\text{cot}}$=0.01, we observe length increase for almost all checkpoints and accompanying performance gains. When the length does not grow (SFT epochs 1 and 10), test accuracy plateaus to a small value (in comparison to the other checkpoints). RL consistently induces generalizing models for larger values of $p_{\text{cot}}$. We observe that test accuracy and average token length are very similar across these checkpoints.

that these models were heavily finetuned on the MATH dataset prior to its release, introducing additional confounding factors.

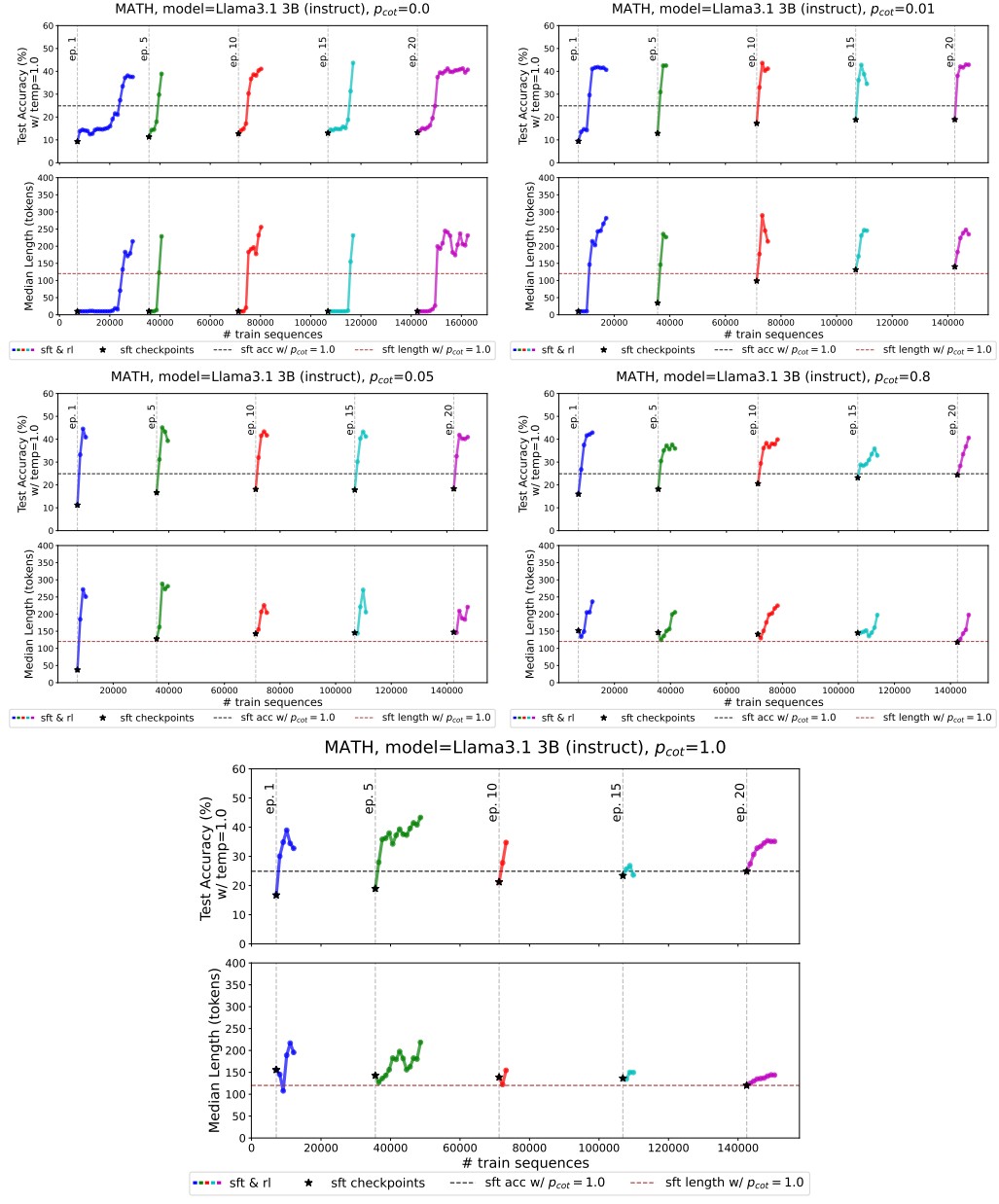

Figure 28: **SFT and GRPO of Llama 3.2 3B (instruct) on the MATH dataset** for various values of $p_{\mathrm{cot}}$. The situation is similar to the GSM8k experiments, yet RL gains might be due to out of distribution gains – see qualitative analysis. We early stopped the curves before the RL runs collapse, which we attribute to large learning rate.

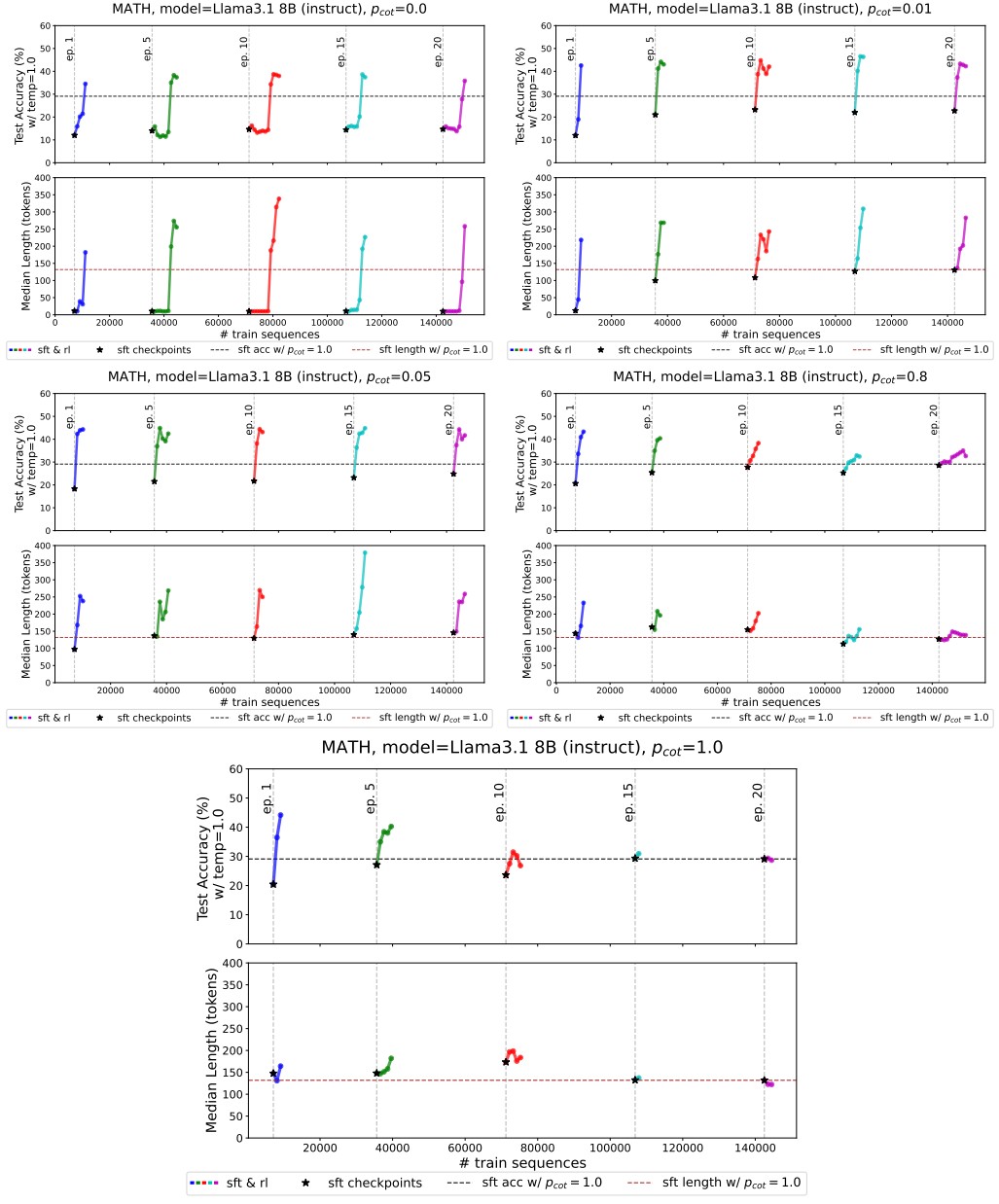

Figure 29: **SFT and GRPO of Llama 3.1 8B (instruct) on the MATH dataset** for various values of $p_{\text{cot}}$. The situation is similar to the GSM8k experiments, yet RL gains might be due to out of distribution gains – see qualitative analysis. We early stopped the curves before the RL runs collapse, which we attribute to large learning rate.

**GSM8k Completions ($p_{cot}$ = 0.05, epoch=1, RL step=1)**

**Question: In four years, Peter will be twice as old as Harriet. If Peter's age is currently half of his mother's age, who is 60, how old is Harriet now?**
**Ground truth cot: If Peter's age is currently half of his mother's age, who is 60, Peter is 1/2\*60 = <<30=30>>30 years old. In four years, Peter will be 30+4 = <<30+4=34>>34 years old. Since Peter's age in four years will be twice Harriet's age, Harriet will be 34/2 = <<34/2=17>>17 years old in four years. Currently, Harriet is 17-4 = <<17-4=13>>13 years old. Answer: 13**

*Generation 1*; `reasoning process here</think> <answer>32</answer>` *Reward: 10*

*Generation 2*; `reasoning process here</think> <answer>35</answer>` *Reward: 10*

*Generation 3*; `...]</think> <answer>24</answer>` *Reward: 10*

*Generation 4*; `reasoning process here</think> <answer>34</answer>` *Reward: 10*

*Generation 5*; `?</think> <answer>40</answer>` *Reward: 10*

*Generation 6*; `processing online </think> <answer>35</answer>` *Reward: 10*

*Generation 7*; `reasoning process here</think> <answer>18</answer>` *Reward: 10*

*Generation 8*; `2020-11-23T11:49:57.397405+00:00</think> <answer>40</answer>` *Reward: 10*

*Generation 9*; `thinking...  </think> <answer>21</answer>` *Reward: 10*

*Generation 10*; `Harriet's age right now =</think> <answer>2</answer>` *Reward: 10*

*Generation 11*; `reasoning process here</think> <answer>32</answer>` *Reward: 10*

*Generation 12*; `2016-09-19T00:00:00Z</think> <answer>24</answer>` *Reward: 10*

*Generation 13*; `3</think> <answer>36</answer>` *Reward: 10*

*Generation 14*; `reasoning process here</think> <answer>40</answer>` *Reward: 10*

*Generation 15*; `Harriet = 4</think> <answer>84</answer>` *Reward: 10*

*Generation 16*; `reasoning process here</think> <answer>48</answer>` *Reward: 10*

*Generation 17*; `</think> <answer>25</answer>` *Reward: 10*

*Generation 18*; `  For Peter to be twice as old as Harriet, Harriet's age has to be half the current age of Peter.  Start with the current age of Peter as x years.  Then half of x is half of mother's age=60, or` $\frac{x}{2} = 60 \Rightarrow x = 60 \times 2 =$ `120[0] Harriet's age=`$x - 2 = 120 - 2 = 118$`[1] Harriet's age is 118 years, and Peter is 120 years.  The answer is 118.  [2]November 6, 2021` *Reward: 0*

*Generation 19*; `(please wait a few seconds)</think> <answer>18</answer>` *Reward: 10*

*Generation 20*; `my reasonning here</think> <answer>18</answer>` *Reward: 10*

*Generation 21*; `</think> <answer>20</answer>` *Reward: 10*

*Generation 22*; `could not solve the problem</think> <answer>13</answer>` *Reward: 100*

*Generation 23*; `2016-03-11 20:54:37</think> <answer>36</answer>` *Reward: 10*

*Generation 24*; `reasoning process here</think> <answer>40</answer>` *Reward: 10*

*Generation 25*; `         IN FOUR YEARS, PETER WILL BE TWICE AS OLD</think> <answer>8</answer>` *Reward: 10*

*Generation 26*; `He is fourty years now</think> <answer>35</answer>` *Reward: 10*

*Generation 27*; `10</think> <answer>25</answer>` *Reward: 10*

*Generation 28*; `Harriet's current age?</think> <answer>24</answer>` *Reward: 10*

*Generation 29*; `    Why is Harriet's age h?  Who is w?  Who is d?  <think> <answer>37</answer>` *Reward: 5*

*Generation 30*; `</think> <answer>36</answer>` *Reward: 10*

*Generation 31*; `include forgive-me.fr</think> <answer>32</answer>` *Reward: 10*

*Generation 32*; `    Ernesto thought about reasoning process here </think> <answer>42</answer>` *Reward: 10*

Figure 30: Llama 3.2 3B completion on a GSM8k prompt for $p_{cot}$ = 0.05, SFT epochs=1, RL step=1. Most completions are short (following the SFT format of `</think><answer>ANS</answer>`) and inaccurate.

---

**GSM8k Completions ($p_{cot}$ = 0.05, epoch=1, RL step=58)**

**Question: Mrs. Watson is grading 120 final exams from her American History class. On Monday, she grades 60% of the exams. On Tuesday, she grades 75% of the remaining exams. On Wednesday, how many more exams does she need to grade?**

**Ground truth cot: On Monday, Mrs. Watson grades 120 x 60% = <<120\*60\*.01=72>>72 final exams. At the end of Monday, she has 120 - 72 = <<120-72=48>>48 exams left to grade. On Tuesday, she grades 48 x 75% = <<48\*75\*.01=36>>36 final exams. On Wednesday, she has 48 - 36 = <<48-36=12>>12 exams left to grade.**

**Answer: 12**

---

*Generation 1*; ...</think> <answer>30</answer> *Reward: 10*

*Generation 2*; She graded 60% of 120 = 72 exams on Monday.</think> <answer>24</answer> *Reward: 10*

*Generation 3*; ...</think> <answer>15</answer> *Reward: 10*

*Generation 4*; Let's get a variable for Mrs. Watson's total number of exams.</think> <answer>30</answer> *Reward: 10*

*Generation 5*; 120 final exams are left to grade, or 30. She grades 60% of them on Monday, which is 18.</think> <answer>12</answer> *Reward: 100*

*Generation 6*; Mrs. Watson needs to grade 30% of the exams</think> <answer>40</answer> *Reward: 10*

*Generation 7*; ...</think> <answer>30</answer> *Reward: 10*

*Generation 8*; She's finished 60% of the total. How many are left? We have 100\*60=60 exams left. Next, on Tuesday, she grades another 75% of those. She now has finished 60+75=135 out of 120 exams. So how many exams are left? There are 120-135=15 exams left.</think> <answer>45</answer> *Reward: 10*

[...]

*Generation 22*; There is a 60% completion rate of all exams. On Monday, 60% of all grades were given. This means that 60% of 120, or 72, were graded so far. Thus, 48 have yet to be graded. 75% of the remaining exams that haven't been graded will be graded on Tuesday. There are then 48%×75% = 36 more exams to be graded.</think> <answer>36</answer> *Reward: 10*

*Generation 23*; ...</think> <answer>10</answer> *Reward: 10*

*Generation 24*; 120</think> <answer>30</answer> *Reward: 10*

*Generation 25*; That's a really easy word problem. It's a good thing I worked through so many examples last night.</think> <answer>30</answer> *Reward: 10*

*Generation 26*; She has graded them at a rate of 60% and then 75% of them remaining, so she will need to grade...</think> <answer>30</answer> *Reward: 10*

*Generation 27*; ...once she grades 60% of the exams, she grades the remaining...</think> <answer>30</answer> *Reward: 10*

*Generation 28*; Well, we know that we have 120 exams that need to be graded. Mrs. Watson graded 60% on Monday so that was 60 exams. That means that she graded 60 120 = 60 exams on Monday.</think> <answer>30</answer> *Reward: 10*

*Generation 29*; Let's see....</think> <answer>15</answer> *Reward: 10*

*Generation 30*; ...</think> <answer>15</answer> *Reward: 10*

*Generation 31*; ...</think> <answer>20</answer> *Reward: 10*

*Generation 32*; He asked how many more exams she needs to grade...wish she had thought of this one...</think> <answer>6</answer> *Reward: 10*

Figure 31: Llama 3.2 3B completion on a GSM8k prompt for $p_{cot}$ = 0.05, SFT epochs=1, RL step=58. The model starts generating longer responses that mimic the long SFT demonstrations, while continuing to respect the SFT answering format.

**GSM8k Completions** ($p_{\text{cot}}$ = 0.05, epoch=1, RL step=118)

**Question: Ethan is reading a sci-fi book that has 360 pages. He read 40 pages on Saturday morning and another 10 pages at night. The next day he read twice the total pages as on Saturday. How many pages does he have left to read?**
**Ground truth cot: Ethan read a total of 40 pages + 10 pages = $<<40+10=50>>$50 pages on Saturday. On the next day, he read a total of 50 pages x 2 = $<<50*2=100>>$100 pages. The total number of pages that Ethan had read is 50 pages + 100 pages = $<<50+100=150>>$150 pages. Therefore, Ethan has 360 pages - 150 pages = $<<360-150=210>>$210 pages left to read.**
**Answer: 210**

---

*Generation 1*; Ethan was reading a sci-fi book that had 360 pages.Ethan read 40 pages on Saturday morning and then another 10 pages on Saturday night.So far he had read 50 pages on Saturday.So the next day he read 2*50 = 100 more pages So far he had read 50 + 100 = 150 pages So the next day he has read a total of 150 pages.Ethan has 360 - 150 = 210 pages left to read.</think> <answer>210</answer> *Reward: 100*

*Generation 2*; Our book had 360 pages in total.Ethan read 40 pages on Saturday so there are 320 pages left. Ethan read 10 more pages at night so there are now 330 pages left. So far he has read 50 pages in 2 days. So if he read one day on Saturday then he must have read 25 on Sunday. So now there are 320-25 = 295 pages left to read.</think> <answer>295</answer> *Reward: 10*

*Generation 3*; Ethan is reading a sci fi book that has 360 pages. He read 40 pages on Saturday morning and 10 pages at night. So the total number of pages he read on Saturday was 40 + 10 = 50 pages. The next day he read twice the number of pages he read on Saturday. So the number of pages he read on Sunday was 2 * 50 = 100. So the total number of pages he read on Saturday and Sunday was 50 + 100 = 150 pages. So the total number of pages he read was 50 + 100 = 150. So the remaining number of pages were 360 - 150 = 210 pages. </think> <answer>210</answer> *Reward: 100*

[...]

*Generation 30*; Ethans book has 360 pages.Ethan read 40 pages on Saturday morning.Ethan read an additional 10 pages at night.Ethan read on Saturday 50 pages = 40 + 10 = 50.Ethans book has on the next day twice as many pages as on Saturday.So on the next day Ethan would have read 50 * 2 = 100 pages.Ethans book had a Total of 360 - 50 - 100 = 210 pages he had left to read.</think> <answer>210</answer> *Reward: 100*

*Generation 31*; Ethan has a 360 page science fiction book. He read 40 pages on Saturday morning and then 10 pages on Saturday night. So far he has read 50 pages. The next day he read 2 times 50 which equals 100 pages. So he has read a total of 150 pages. So he has 360 - 150 = 210 pages left to read.</think> <answer>210</answer> *Reward: 100*

*Generation 32*; Ethan is reading a sci-fi book that has 360 pages. He read 40 pages on Saturday morning and another 10 pages on Saturday at night. So far today he has read 50 pages. The next day he read 2 times 50 which equals 100 pages He has read so far today and the next day so he has read a total of 150 pages. So far he has read 360 minus 150 which equals 210 pages. So he has 210 pages left to read.</think> <answer>210</answer> *Reward: 100*

Figure 32: Llama 3.2 3B completion on a GSM8k prompt for $p_{\text{cot}}$ = 0.05, SFT epochs=1, RL step=117. After only 100 RL steps, the overwhelming majority of model generations consist of long, elaborate responses that give the correct answer.

**GSM8k Completions** ($p_{cot}$ = 0.05, epoch=1, RL step=4676)

**Question: Gabe has three songs on his playlist. "The Best Day" is 3 minutes, "Raise the Roof" is 2 minutes, and "Rap Battle" is 3 minutes. How many times can Gabe listen to his entire playlist on the 40-minute ride to his wrestling match?**
**Ground truth cot: If you add all the songs together, Gabe's playlist is 3 + 2 + 3 = <<3+2+3=8>>8 minutes long. During the 40-minute car ride, Gabe can listen to his 8-minute playlist 40 / 8 = <<40/8=5>>5 times.**
**Answer: 5**

---

*Generation 1*; We have the following information:
```
Gabe has three songs on his playlist.
"The Best Day" is 3 minutes
Raise the Roof" is 2 minutes
"Rap Battle" is 3 minutes
So the total length of the three songs in minutes = 3 + 2 + 3 = 8 minutes
So the number of minutes in the ride to his wrestling match is 40 minutes
So the number of times that the entire playlist can be played during the
40-minute ride = 40 / 8 = 5 plays of the playlist
</think> <answer>5</answer>
```
*Reward: 100*

*Generation 2*; We have the following information:
```
Gabe has three songs on his playlist.
"The Best Day" is 3 minutes long
"Raise the Roof" is 2 minutes long
"Rap Battle" is 3 minutes long
So the total length of the three songs on Gabe's playlist = 3 minutes for
"The Best Day"
+ 2 minutes for "Raise the Roof"
+ 3 minutes for "Rap Battle"
So the total length of the playlist = 3 + 2 + 3 = 8 minutes
So the length of the ride to his wrestling match is 40 minutes
So the number of times that the entire playlist can fit into the 40-minute
ride = 40 / 8 = 5 total plays of the playlist
</think> <answer>5</answer>
```
*Reward: 100*
```
[...]
```
*Generation 32*; We have the following information:
```
Gabe has three songs on his playlist.
\The Best Day" is 3 minutes long
\Raise the Roof" is 2 minutes long
\Rap Battle" is 3 minutes long
So the total length of the three songs on Gabe's playlist = 3 minutes for
\The Best Day"
+ 2 minutes for \Raise the Roof"
+ 3 minutes for \Rap Battle"
So the total length of the playlist = 3 + 2 + 3 = 8 minutes
Gabe's ride to his wrestling match lasts 40 minutes.
So the number of times that the entire playlist can fit into the 40-minute
ride = 40 / 8 = 5 complete plays of the playlist
So the answer to how many times can Gabe listen to his entire playlist on
the 40-minute ride to his wrestling match = 5 plays
</think> <answer>5</answer>
```
*Reward: 100*

Figure 33: Llama 3.2 3B completion on a GSM8k prompt for $p_{cot}$ = 0.05, SFT epochs=1, RL step=4676. After 1000s of RL steps, model responses cease to resemble SFT data and are much more verbose.

---

**MATH Completions ($p_{\text{cot}}$ = 0.01, epoch=20, RL step=1)**

**Question: In triangle** $ABC$, $AB = 13$, $BC = 15$ **and** $CA = 17$. **Point** $D$ **is on** $\overline{AB}$, $E$ **is on** $\overline{BC}$, **and** $F$ **is on** $\overline{CA}$. **Let** $AD = p \cdot AB$, $BE = q \cdot BC$, **and** $CF = r \cdot CA$, **where** $p$, $q$, **and** $r$ **are positive and satisfy** $p + q + r = 2/3$ **and** $p^2 + q^2 + r^2 = 2/5$. **The ratio of the area of triangle** $DEF$ **to the area of triangle** $ABC$ **can be written in the form** $m/n$, **where** $m$ **and** $n$ **are relatively prime positive integers. Find** $m + n$.

**Ground truth cot: [asy] /* – arbitrary values, I couldn't find nice values for pqr please replace if possible – */ real p = 0.5, q = 0.1, r = 0.05; /* – arbitrary values, I couldn't find nice values for pqr please replace if possible – */ pointpen = black; pathpen = linewidth(0.7) + black; pair A=(0,0),B=(13,0),C=IP(CR(A,17),CR(B,15)), D=A+p\*(B-A), E=B+q\*(C-B), F=C+r\*(A-C); D(D(MP("A",A))–D(MP("B",B))–D(MP("C",C,N))–cycle); D(D(MP("D",D))–D(MP("E",E,NE))–D(MP("F",F,NW))–cycle); [/asy] We let** [. . .] **denote area; then the desired value is** $\frac{m}{n} = \frac{[DEF]}{[ABC]} = \frac{[ABC]-[ADF]-[BDE]-[CEF]}{[ABC]}$

**Using the formula for the area of a triangle** $\frac{1}{2}ab\sin C$, **we find that** $\frac{[ADF]}{[ABC]} = \frac{\frac{1}{2} \cdot p \cdot AB \cdot (1-r) \cdot AC \cdot \sin \angle CAB}{\frac{1}{2} \cdot AB \cdot AC \cdot \sin \angle CAB} = p(1 - r)$ **and similarly that** $\frac{[BDE]}{[ABC]} = q(1 - p)$ **and** $\frac{[CEF]}{[ABC]} = r(1 - q)$. **Thus, we wish to find**

$$\frac{[DEF]}{[ABC]} = 1 - \frac{[ADF]}{[ABC]} - \frac{[BDE]}{[ABC]} - \frac{[CEF]}{[ABC]}$$
$$= 1 - p(1 - r) - q(1 - p) - r(1 - q)$$
$$= (pq + qr + rp) - (p + q + r) + 1$$

**We know that** $p + q + r = \frac{2}{3}$, **and also that** $(p+q+r)^2 = p^2 + q^2 + r^2 + 2(pq+qr+rp) \Longleftrightarrow pq + qr + rp = \frac{\left(\frac{2}{3}\right)^2 - \frac{2}{5}}{2} = \frac{1}{45}$. **Substituting, the answer is** $\frac{1}{45} - \frac{2}{3} + 1 = \frac{16}{45}$, **and** $m + n = \boxed{61}$. **Answer: 61**

---

*Generation 1*; `</think> <answer>103</answer>` *Reward: 10*
*Generation 2*; `</think> <answer>103</answer>` *Reward: 10*
*Generation 3*; `</think> <answer>103</answer>` *Reward: 10*
*Generation 4*; `</think> <answer>103</answer>` *Reward: 10*
*Generation 5*; `DEEZNIT</think> <answer>123</answer>` *Reward: 10*
*Generation 6*; `</think> <answer>103</answer>` *Reward: 10*
*Generation 7*; `></think> <answer>103</answer>` *Reward: 10*
*Generation 8*; `forcedeterminant</think> <answer>121</answer>` *Reward: 10*
*Generation 9*; `think</think> <answer>103</answer>` *Reward: 10*
*Generation 10*; `</think> <answer>59</answer>` *Reward: 10*
*Generation 11*; `<think> <think></think></think> <answer>137</answer>` *Reward: 10*
*Generation 12*; `<answer>103</answer>` *Reward: 5*
*Generation 13*; `</think> <answer>103</answer>` *Reward: 10*
*Generation 14*; `</think> <answer>937</answer>` *Reward: 10*
*Generation 15*; `></think> <answer>101</answer>` *Reward: 10*
*Generation 16*; `</think> <answer>103</answer>` *Reward: 10*
*Generation 17*; `></think> <answer>13</answer>` *Reward: 10*
*Generation 18*; `commands- i</think> <answer>9</answer>` *Reward: 10*
*Generation 19*; `</think> <answer>243</answer>` *Reward: 10*
*Generation 20*; `</think> <answer>103</answer>` *Reward: 10*

Figure 34: Llama 3.1 8B (instruct) completion on a MATH prompt for $p_{\text{cot}}$ = 0.01, SFT epochs=20, RL step=1. The model generates short inaccurate responses.

---

**MATH Completions ($p_{\text{cot}} = 0.01$, epoch=20, RL step=181)**

**Question: Let $a$, $b$, $c$ be distinct complex numbers such that**

$$\frac{a}{1-b} = \frac{b}{1-c} = \frac{c}{1-a} = k.$$

**Find the sum of all possible values of $k$.**
**Ground truth cot: From the given equation,**

$$a = k(1-b),$$
$$b = k(1-c),$$
$$c = k(1-a).$$

**Then**

$$a = k(1-b)$$
$$= k(1 - k(1-c))$$
$$= k(1 - k(1 - k(1-a))).$$

**Expanding, we get $ak^3 + a - k^3 + k^2 - k = 0$, which factors as**

$$(k^2 - k + 1)(ak + a - k) = 0.$$

**If $ak + a - k = 0$, then $a = \frac{k}{k+1}$, in which case $b = c = \frac{k}{k+1}$. This is not allowed, as $a$, $b$, and $c$ are distinct, so $k^2 - k + 1 = 0$. The sum of the roots is $\boxed{1}$. Note: The roots of $k^2 - k + 1 = 0$ are**

$$\frac{1 \pm i\sqrt{3}}{2}.$$

**For either value of $k$, we can take $a = 0$, $b = 1$, and $c = k$. Answer: 1**

---

*Generation 1;* `></think> <answer>`From the given equation, we have: $a = k(1-b), b = k(1-c), c = k(1-a)$. Substituting $b = k(1-c)$ into $a = k(1-b)$ gives: $a = k(1 - k(1-c)) = k - k^2 + k^2 c$. Substituting $a = k(1-b)$ into $c = k(1-a)$ gives: $c = k(1 - k(1-b)) = k - k^2 + k^2 b$. Equating $a = k - k^2 + k^2 c$ and $a = k - k^2 + k^2 b$, we get: $k^2 c = k^2 b$, so $c = b$. Substituting $c = b$ into $a = k - k^2 + k^2 c$ gives: $a = k - k^2 + k^2 b$. Also, we know that $b = k - k^2 + k^2 a$. Equating $a = k - k^2 + k^2 b$ and $a = k - k^2 + k^2 a$, we get: $b = a$. Since $a, b, c$ are distinct, we have a contradiction. Therefore, the initial assumption that there are distinct complex numbers $a, b, c$ is incorrect. There can be only one value of $k$. Adding the equations $a = k - k^2 + k^2 b, b = k - k^2 + k^2 a, c = k - k^2 + k^2 a$, we get: $a + b + c = 3k - 3k^2 + k^2(a + b + c)$. Equating $3k - 3k^2 = 0$, we get: $3k(1 - k) = 0$. Solving for $k$, we get: $k = 0$ or $k = 1$. We are only interested in non-trivial solutions, so the sum of all possible values of $k$ is: $0 + 1 = 1$. `</answer>` *Reward: 100*

[...]

*Generation 14;* `></think> <answer>`$\frac{a}{1-b} + \frac{b}{1-c} + \frac{c}{1-a} = k + k + k = 3k = 3 \cdot \frac{a}{1-b} = 3 \cdot \frac{b}{1-c} = 3 \cdot \frac{c}{1-a}$. Therefore, $k$ can be any value such that the equation $\frac{a}{1-b} = \frac{b}{1-c} = \frac{c}{1-a} = k$ holds. It can be shown that the sum of all possible values of $k$ is $3$. `</answer>` *Reward: 20*

[...]

Figure 35: Llama 3.1 8B (instruct) completion on a MATH prompt for $p_{\text{cot}} = 0.01$, SFT epochs=20, RL step=181. The model starts generating longer responses, which however do not respect the SFT format. This behavior is likely due to out of distribution generalization.

---

**MATH Completions ($p_{\text{cot}}$ = 0.01, epoch=1, RL step=1)**

**Question: In triangle** $ABC$, $\angle B = 60°$ **and** $\angle C = 45°$. **The point** $D$ **divides** $\overline{BC}$ **in the ratio** $1 : 3$. **Find**

$$\frac{\sin \angle BAD}{\sin \angle CAD}.$$

**Answer:** $\frac{\sqrt{6}}{6}$

---

*Generation 1*; `Categories problem. Inclines BC</think> <answer>`$\sqrt{2}$`</answer>` *Reward: 10*

*Generation 2*; `</think> <answer>2</answer>` *Reward: 10*

*Generation 3*; `</think> <answer>2</answer>` *Reward: 10*

*Generation 4*; `</think> <answer>2</answer>` *Reward: 10*

*Generation 5*; `</think> <answer>` $\boxed{2}$ `</answer>` *Reward: 10*

*Generation 6*; `></think> <answer>3</answer>` *Reward: 10*

*Generation 7*; `></think> <answer>3</answer>` *Reward: 10*

*Generation 8*; `</think> <answer>1</answer>` *Reward: 10*

*Generation 9*; `</think> <answer>2</answer>` *Reward: 10*

*Generation 10*; `</think> <answer>`$\frac{\sqrt{3}}{2+\sqrt{3}}$`</answer>` *Reward: 10*

*Generation 11*; `</think> <answer>`$\sqrt{3}$`</answer>` *Reward: 10*

*Generation 12*; `</think> <answer>`$\sqrt{3},$`</answer>` *Reward: 10*

*Generation 13*; `No </think> <answer>`$\sqrt{3}$`</answer>` *Reward: 10*

*Generation 14*; `</think> <answer> `$\sqrt{3}$` </answer>` *Reward: 10*

*Generation 15*; `</think> <answer>`$2 - \sqrt{3}$`</answer>` *Reward: 10*

*Generation 16*; `</think> <answer>`$\frac{2-\sqrt{3}}{1+\sqrt{3}}$` end answer>` *Reward: 5*

*Generation 17*; `</think> <answer>`$\sqrt{7}$`</answer>` *Reward: 10*

*Generation 18*; `</think> <answer>2</answer>` *Reward: 10*

*Generation 19*; `</think> <answer>`$3\sqrt{3}$`</answer>` *Reward: 10*

*Generation 20*; `</think> <answer> 1 </answer>` *Reward: 10*

Figure 36: Llama 3.2 3B (instruct) completion on a MATH prompt for $p_{\text{cot}}$ = 0.01, SFT epochs=1, RL step=1. The model generates short inaccurate responses, which nevertheless adhere to the SFT format.

---

**MATH Completions ($p_{\text{cot}}$ = 0.01, epoch=1, RL step=87)**

**Question: Triangle $ABC$ has sides of $6$ units, $8$ units, and $10$ units. The width of a rectangle, whose area is equal to the area of the triangle, is $4$ units. What is the perimeter of this rectangle, in units?**
**Answer: 20**

---

*Generation 1*; `</think> <answer>`2(6 + 8) = 28`</answer>` *Reward: 10*
*Generation 2*; `</think> <answer>`2(6+8) = 28`</answer>` *Reward: 10*
*Generation 3*; `</think> <answer>`2(6+8+6+4) = 44`</answer>` *Reward: 10*
*Generation 4*; `</think> <answer>`44`</answer>` *Reward: 10*
*Generation 5*; `</think> <answer>`$2 \cdot (6+8) = 28$`</answer>` *Reward: 10*
*Generation 6*; `</think> <answer>`2(6+8)=28`</answer>` *Reward: 10*
*Generation 7*; `</think> <answer>`2(6+4)+2(8+4)=36`</answer>` *Reward: 10*
*Generation 8*; `</think> <answer>`$2 \cdot (5+6) = 2 \cdot 11 = 22$`</answer>` *Reward: 10*
*Generation 9*; `</think> <answer>`20 + 8`</answer>` *Reward: 20*
*Generation 10*; `</think> <answer>`$2 \times (5 \times 8) = 40 </answer>$` *Reward: 10*
*Generation 11*; `</think> <answer>`2(6+8) = 28`</answer>` *Reward: 10*
*Generation 12*; `</think> <answer>`2(10 + 6) = 32`</answer>` *Reward: 10*
*Generation 13*; `</think> <answer>`2(6 + 8 + 10 + 4) = 76`</answer>` *Reward: 10*
*Generation 14*; `</think> <answer>`$2 \cdot (10+6) = 32$`</answer>` *Reward: 10*
*Generation 15*; `</think> <answer>`2(10 + 6) = 32`</answer>` *Reward: 10*
*Generation 16*; `</think> <answer>`2(4+12+8) = 52`</answer>` *Reward: 10*
*Generation 17*; `</think> <answer>`2(10+4)=36`</answer>` *Reward: 10*
*Generation 18*; `</think> <answer>`2 * (4 + 5) = 18`</answer>` *Reward: 10*
*Generation 19*; `</think> <answer>`2(4+5)=18`</answer>` *Reward: 10*
*Generation 20*; `</think> <answer>`2(6 + 8) = 28`</answer>` *Reward: 10*

Figure 37: Llama 3.2 3B (instruct) completion on a MATH prompt for $p_{\text{cot}}$ = 0.01, SFT epochs=1, RL step=87. The model learns to generate some reasoning in the answer part of the response, which does not seem to be a very successful strategy.

---

**MATH Completions ($p_{\text{cot}}$ = 0.01, epoch=1, RL step=92)**

### Question: Simplify

$$\frac{1}{\log_{15} 2 + 1} + \frac{1}{\log_{10} 3 + 1} + \frac{1}{\log_6 5 + 1}.$$

### Answer: 2

---

*Generation 1*; `></think> <answer>1</answer>` *Reward: 10*

*Generation 2*; `</think> <answer>1</answer>` *Reward: 10*

*Generation 3*; `</think> <answer>1</answer>` *Reward: 10*

*Generation 4*; `</think> <answer>1</answer>` *Reward: 10*

*Generation 5*; `</think> <answer>1</answer>` *Reward: 10*

*Generation 6*; `</think> <answer>1</answer>` *Reward: 10*

*Generation 7*; `</think> <answer>1</answer>` *Reward: 10*

*Generation 8*; `Recall the change-of-base formula for logarithms:` $\log_b a = \frac{\log_c a}{\log_c b}$ `for any positive numbers` $a, b$, `and` $c$. `Also note that` $\frac{1}{x+1} = \frac{\log(x+1)}{\log(x+1)^{x+1}}$. `We can rewrite each term of the given expression using these properties. After the rewriting, we obtain`

$$\frac{\log 15}{\log 15 + \log 2} + \frac{\log 10}{\log 10 + \log 3} + \frac{\log 6}{\log 6 + \log 5},$$

`where we have removed the denominators to make the arguments in the denominator of each term equal` $\log_{15} 2, \log_{10} 3,$ `and` $\log_6 5$. `Since the 2nd and 3rd terms are already in lowest terms, we focus on simplifying the 1st term:`

$$\frac{\log 15}{\log 15 + \log 2} = \frac{\log 15}{\log(15 \cdot 2)} = \frac{1}{\log_{30} 15}.$$

`Thus, the given expression can be written as`

$$\frac{1}{\log_{30} 15} + \frac{\log_{10} 3}{\log_{10} 10 + \log_{10} 3} + \frac{\log_6 5}{\log_6 6 + \log_6 5}.$$

`By using a change-of-base formula, we find that` $\frac{1}{\log_{30} 15} = \log_{15} 30$. `The final result is the following:` *Reward: 0*

*Generation 9*; `</think> <answer>1</answer>` *Reward: 10*

*Generation 10*; `></think> <answer>1</answer>` *Reward: 10*

*Generation 11*; `</think> <answer>1</answer>` *Reward: 10*

*Generation 12*; `</think> <answer>1</answer>` *Reward: 10*

*Generation 13*; `</think> <answer>1</answer>` *Reward: 10*

*Generation 14*; `Let's use the change-of-base formula to simplify each of the logarithmic expressions in the denominators of the fractions.</think> <answer>1</answer>` *Reward: 10*

*Generation 15*; `</think> <answer>1</answer>` *Reward: 10*

*Generation 16*; `</think> <answer>1</answer>` *Reward: 10*

*Generation 17*; `</think> <answer>1</answer>` *Reward: 10*

*Generation 18*; `Using the identity` $\log_b a + 1 = \log_b ab$, `the expression can be simplified as` $\frac{1}{\log_{15} 30} + \frac{1}{\log_{10} 30} + \frac{1}{\log_6 30}$, `then applying the change of base formula,` $\log_b a = \frac{\log_c a}{\log_c b}$, `we get` $\frac{1}{\frac{\log 30}{\log 15}} + \frac{1}{\frac{\log 30}{\log 10}} + \frac{1}{\frac{\log 30}{\log 6}}$, `then` $\frac{\log 15}{\log 30} + \frac{\log 10}{\log 30} + \frac{\log 6}{\log 30}$, `and after cancelling out common terms, we find` $\frac{\log 15 + \log 10 + \log 6}{\log 30}$, `and finally` $\frac{\log(15 \cdot 10 \cdot 6)}{\log 30}$, `which is` $\frac{\log 900}{\log 30}$, `equal to` $\frac{\log(30^2)}{\log 30}$, `equal to 2, and the final value is` $\boxed{2}$ `.</think> <answer>2</answer>` *Reward: 100*

*Generation 19*; `</think> <answer>1</answer>` *Reward: 10*

*Generation 20*; `</think> <answer>1</answer>` *Reward: 10*

---

Figure 38: Llama 3.2 3B (instruct) completion on a MATH prompt for $p_{\text{cot}}$ = 0.01, SFT epochs=1, RL step=92. The model starts exploring longer responses in the thinking part of the response.

---

**MATH Completions ($p_{\text{cot}}$ = 0.01, epoch=1, RL step=102)**

**Question: Let $G$ be the centroid of triangle $ABC$. If $GA^2 + GB^2 + GC^2 = 58$, then find $AB^2 + AC^2 + BC^2$.**
**Answer: 174**

---

*Generation 1*; Since $G$ is the centroid of triangle $ABC$, by the properties of a centroid, $G$ divides the medians of $\triangle ABC$ into six equal segments, with $GA = \frac{1}{3}AD$, $GB = \frac{1}{3}BE$, and $GC = \frac{1}{3}CF$, where $D, E$, and $F$ are the vertices of trisections of the medians $AD, BE$, and $CF$. Then by the Midpoint Theorem applied to triangles $AGD, BGE, CFC$, we have $GD = 2(AG)$, $GE = 2(BG)$, and $FC = 2(CF)$, and since $AD = AE, BE = BG$, $CF = CE$, we get $GD^2 = 4AG^2, GE^2 = 4BG^2$, and $FC^2 = 4CF^2$, where $D, E$ and $F$ are the midpoints of sides $AB, BC$, and $CA$. Now, $GA^2 + GB^2 + GC^2 = \left(\frac{1}{3}AD\right)^2 + \left(\frac{1}{3}BE\right)^2 + \left(\frac{1}{3}CF\right)^2 = \frac{1}{9} \cdot (AD^2 + BE^2 + CF^2)$. Thus, $\frac{9}{1} \cdot (GA^2 + GB^2 + GC^2) = AD^2 + BE^2 + CF^2 = 3 \cdot (GA^2 + GB^2 + GC^2) = 3 \cdot 58$, then $AB^2 + AC^2 + BC^2 = AD^2 + BE^2 + CF^2$. </think> <answer>174</answer> *Reward: 100*

*Generation 2*; Using the fact that the centroid divides each median into two segments with a 2:1 ratio, we have $AG = \frac{2}{3}AD$, $BG = \frac{2}{3}BE$, and $CG = \frac{2}{3}CF$, where $D$, $E$, and $F$ are the midpoints of $\overline{BC}$ $\overline{AC}$, and $\overline{AB}$ respectively.</think> <think>Therefore, $GA^2 + GB^2 + GC^2 = \frac{4}{9}(AD^2 + BE^2 + CF^2)$. Since $AD = \frac{1}{2}AC$, $BE = \frac{1}{2}AB$, and $CF = \frac{1}{2}BC$, we have $GA^2 + GB^2 + GC^2 = \frac{4}{9}(AC^2 + AB^2 + BC^2) = 58$.</think> <think>It follows that $AC^2 + AB^2 + BC^2 = \frac{9}{4} \cdot 58 = \boxed{132}$.</think> <answer>132</answer> *Reward: 5*

*Generation 3*; By the relationship between the centroid and the circumcenter of a triangle, we know that $G$ is the circumcenter of triangle $ABC$. Thus, $AG = BG = CG$, since the circumcenter is equidistant from the vertices of the triangle. Letting $x = AG = BG = CG$, we see that

$$x^2 + x^2 + x^2 = 58$$
$$3x^2 = 58$$
$$x^2 = 58/3.$$

</think> <answer>58 + 58/3</answer> *Reward: 10*

*Generation 4*; By the configparser proposition, we know that $GA^2 + GB^2 + GC^2 = \frac{1}{3}(AB^2 + AC^2 + BC^2)$, so $AB^2 + AC^2 + BC^2 = 3 \cdot 58 = 174$.</think> <answer>174</answer>
*Reward: 100*

[...]

*Generation 17*; By the formula for the centroid, $GA^2 + GB^2 + GC^2 = \frac{AB^2 + BC^2 + AC^2}{3} \Rightarrow \frac{1}{3}(AB^2 + BC^2 + AC^2) = 58 \Rightarrow AB^2 + BC^2 + AC^2 = 174 </think>$ <answer>174</answer>
*Reward: 100*

*Generation 18*; By Apollonius's Theorem, $AB^2 + AC^2 = 2(GA^2 + GB^2)$ and $AB^2 + BC^2 = 2(GB^2 + GC^2)$ and $AC^2 + BC^2 = 2(GA^2 + GC^2)$. Adding these together, we have $AB^2 + AC^2 + BC^2 = 2(GA^2 + GB^2 + GC^2) = 2(58) = 116$.</think> <answer>116</answer> *Reward: 10*

*Generation 19*; By the definition of the centroid, we know that $G$ is the point $(1 : 1 : 1)$ in the 3-2-1 coordinate system. Also, by Stewart's Theorem, $GA^2 + GB^2 + GC^2 = \frac{1}{3}(AB^2 + BC^2 + CA^2)$. We want to solve for $AB^2 + BC^2 + CA^2$, so we can multiply both sides by 3 and get $3(GA^2 + GB^2 + GC^2) = AB^2 + BC^2 + CA^2$. Since we are given that $GA^2 + GB^2 + GC^2 = 58$, it follows that $AB^2 + BC^2 + CA^2 = 3(58) = 174$.</think> <answer>174</answer> *Reward: 100*

*Generation 20*; Recall that the centroid divides the median of each triangle into segments whose lengths hold the weights of 2:1, with the longer length being towards the vertex. Then, by the properties of medians, we know

$$GA^2 + GB^2 + GC^2 = \frac{1}{8}(AB^2 + AC^2 + BC^2).$$

If we know this, then the answer is

$$AB^2 + AC^2 + BC^2 = 8(GA^2 + GB^2 + GC^2) = 8(58) = \boxed{464}.$$

</think> <answer>464</answer> *Reward: 10*

Figure 39: Llama 3.2 3B (instruct) completion on MATH prompt for $p_{\text{cot}}$ = 0.01, SFT epochs=1, RL step=102. After 100 RL steps, the model generates long correct responses similar to SFT samples.

## D    USE OF LARGE LANGUAGE MODELS

LLMs were used at several stages during the development of this work:

1. for assistance with coding and figure creation (especially in the Llama experiments);
2. for checking grammar and syntax in parts of the paper;
3. for brainstorming different directions for the proofs of Section 4;
4. for writing this Section.

## E    PROOFS

Here, we present our theoretical analysis. For pre-training, we consider learning the mixture of long and short demonstrations of the parity task with a linear (time-dependent) autoregressive hypothesis class. That is, we consider a series of $d+1$ linear models. For post-training, we consider reinforcement learning with the STaR objective and a chain-of-thought correctness reward. The goal is to show two main learning results: a negative one during pre-training and a positive one after post-training.

*Remark* 4. Linear autoregressive architectures were introduced in (Malach, 2024) to demonstrate the power of autoregressive learning, independently of the specifics of self-attention networks. Leveraging the equivalence between binary computable functions and binary circuits, Malach (2024) proved that such simple autoregressive models can approximate and learn any function efficiently computed by a Turing machine given a dataset that contains appropriate "chain-of-thought" data.

We first cover some preliminary results. We then define our architecture as a series of linear models, and then define pre- & post-training algorithms.

### E.1    PRELIMINARIES & SETUP

We make use of a standard definition and theorem for convex learning as presented in (Shalev-Shwartz & Ben-David, 2014). First, we recall the definition of a convex (and Lipschitz) learning problem, which is based on Definition 12.12 in (Shalev-Shwartz & Ben-David, 2014).

**Definition 1.** (Adaptation of Definition 12.12 in (Shalev-Shwartz & Ben-David, 2014)) Let $\mathcal{H}$ be a hypothesis set, $\mathcal{Z}$ be a measurable instance set and $l : \mathcal{H} \times \mathcal{Z} \mapsto \mathbb{R}$ be a measurable loss function. A learning problem $(\mathcal{H}, \mathcal{Z}, l)$ is called *Convex and $\rho$-Lipschitz* for $\rho > 0$ if:

- The hypothesis set $\mathcal{H}$ is convex.

- For all $z \in \mathcal{Z}$, the (partially applied) loss function $l(\cdot; z)$ is convex and $\rho$-Lipschitz.

The original definition in (Shalev-Shwartz & Ben-David, 2014) is about bounded hypothesis sets, but in our analysis we will use unbounded ones (as a regularization term in the loss function will effectively bound the solution space).

The learning algorithm will be Stochastic Gradient Descent (SGD) for $\ell_2$ regularized learning problems (Algorithm 2 below). Note that the algorithm returns the average of the weights across $T$ iterations, as it is typically the case in online learning.

---

**Algorithm 2** Stochastic Gradient Descent (SGD) for minimizing $\mathbb{E}_{z \sim \mathcal{D}}\left[l(\mathbf{w}, z)\right] + \frac{\lambda}{2}\|\mathbf{w}\|^2$

---

**Require:** Integer $T > 0$
1: Initialize $\mathbf{w}^{(1)} = 0$
2: **for** $t = 1, 2, \ldots, T$ **do**
3:     Sample $z \sim \mathcal{D}$
4:     Set $\mathbf{v}^{(t)} = \nabla_{\mathbf{w}^{(t)}} l(\mathbf{w}^{(t)}, z)$
5:     Set $\eta_t = \frac{1}{\lambda t}$
6:     Set $\mathbf{w}^{(t+1)} = \mathbf{w}^{(t)} - \eta_t \left(\mathbf{v}^{(t)} + \lambda \mathbf{w}^{(t)}\right)$
7: **end for**
8: Output $\bar{\mathbf{w}} = \frac{1}{T} \sum_{t=1}^{T} \mathbf{w}^{(t)}$.

---

We study strongly convex learning objectives, as they permit tight bounds on the calibration of the output of SGD later on. Analyzing their convex (only) counterparts would have yielded weaker calibration guarantees – see, for instance, Chapter 4.7 in Mohri et al. (2012) and the analysis of early stopped SGD in Wu et al. (2025).

We now state a guarantee on the output of SGD after $T$ iterations for convex and Lipschitz objectives plus an additional $\ell_2$ regularization term. The proof follows from Theorem 14.11 in Shalev-Shwartz & Ben-David (2014) by applying it to a learning objective and combining it with Markov's inequality. We present it here for completeness.

**Theorem 3.** *(Adaptation of Theorem 14.11 in Shalev-Shwartz & Ben-David (2014)) Consider a Convex, $\rho$-Lipschitz learning problem ($\mathcal{H} = \mathbb{R}^d, \mathcal{Z}, l$) and a distribution $\mathcal{D}$ over $\mathcal{Z}$. For every $\delta \in (0,1)$, if we run the SGD method (Algorithm 2) for minimizing $L_{\mathcal{D},\lambda}(\mathbf{w}) = \mathbb{E}_{z \sim \mathcal{D}}[l(\mathbf{w}, z)] + \frac{\lambda}{2}\|\mathbf{w}\|^2, \lambda > 0,$ for $T$ iterations and with learning rate $\eta_t = \frac{1}{\lambda t}$, then the output $\bar{\mathbf{w}}$ of SGD satisfies with probability at least $1 - \delta$ over the sampling $z_1, \ldots, z_T \sim \mathcal{D}$:*

$$L_{\mathcal{D},\lambda}(\bar{\mathbf{w}}) \leq \min_{\mathbf{w} \in \mathcal{H}} L_{\mathcal{D},\lambda}(\mathbf{w}) + \frac{2\rho^2}{\delta \lambda T}(1 + \ln T). \tag{22}$$

*Proof.* The objective $L_{\mathcal{D},\lambda}(\mathbf{w}) = \mathbb{E}_{z \sim \mathcal{D}}[l(\mathbf{w}, z)] + \frac{\lambda}{2}\|\mathbf{w}\|^2, \lambda > 0$ is $\lambda$-strongly convex. Let $z \sim \mathcal{D}$, then $\mathbf{v}^{(t)} = \nabla_{\mathbf{w}^{(t)}} l(\mathbf{w}^{(t)}, z)$ and, from the Lipschitz condition, it holds: $\|\mathbf{v}^{(t)}\| \leq \rho$. Let the update direction be: $\mathbf{g}_t = \mathbf{v}^{(t)} + \lambda \mathbf{w}^{(t)}$. Then, we have for the weight vector $\mathbf{w}^{(t+1)}$:

$$\begin{aligned} \mathbf{w}^{(t+1)} &= \mathbf{w}^{(t)} - \eta_t \left(\mathbf{v}^{(t)} + \lambda \mathbf{w}^{(t)}\right) \\ &= \mathbf{w}^{(t)}\left(1 - \frac{1}{t}\right) - \frac{1}{\lambda t}\mathbf{v}^{(t)} \\ &= -\frac{1}{\lambda t}\sum_{i=1}^{t} \mathbf{v}^{(i)}. \end{aligned} \tag{23}$$

Hence, we have $\|\mathbf{w}^{(t)}\| \leq \frac{1}{\lambda(t-1)}(t-1)\rho = \frac{\rho}{\lambda}$, which implies that for the update vector it holds: $\|\mathbf{g}_t\| \leq \rho + \lambda\frac{\rho}{\lambda} = 2\rho$. From Theorem 14.11 (SGD guarantee for strongly convex functions) in Shalev-Shwartz & Ben-David (2014), we get:

$$\mathbb{E}_{z_1,\ldots,z_T \sim \mathcal{D}}[L_{\mathcal{D},\lambda}(\bar{\mathbf{w}})] \leq \min_{\mathbf{w} \in \mathcal{H}} L_{\mathcal{D},\lambda}(\mathbf{w}) + \frac{(2\rho)^2}{2\lambda T}(1 + \ln T) = \min_{\mathbf{w} \in \mathcal{H}} L_{\mathcal{D},\lambda}(\mathbf{w}) + \frac{2\rho^2}{\lambda T}(1 + \ln T). \tag{24}$$

Finally, we apply Markov's inequality on the non-negative random variable $L_{\mathcal{D},\lambda}(\bar{\mathbf{w}}) - \min_{\mathbf{w} \in \mathcal{H}} L_{\mathcal{D},\lambda}(\mathbf{w})$. Let $0 < \delta < 1$, then by leveraging the above guarantee, we get:

$$\begin{aligned} \mathbb{P}_{z_1,\ldots,z_T \sim \mathcal{D}}\left[L_{\mathcal{D},\lambda}(\bar{\mathbf{w}}) - \min_{\mathbf{w} \in \mathcal{H}} L_{\mathcal{D},\lambda}(\mathbf{w}) \geq \tau\right] &\leq \frac{\mathbb{E}_{z_1,\ldots,z_T \sim \mathcal{D}}[L_{\mathcal{D},\lambda}(\bar{\mathbf{w}})] - \min_{\mathbf{w} \in \mathcal{H}} L_{\mathcal{D},\lambda}(\mathbf{w})}{\tau} \\ &\leq \frac{\frac{2\rho^2}{\lambda T}(1 + \ln T)}{\tau} < \delta, \end{aligned} \tag{25}$$

then with probability less than $\delta$: $L_{\mathcal{D},\lambda}(\bar{\mathbf{w}}) - \min_{\mathbf{w} \in \mathcal{H}} L_{\mathcal{D},\lambda}(\mathbf{w}) \geq \tau > \frac{2\rho^2(1+\ln T)}{\lambda T \delta}$. In other words, with probability at least $1 - \delta$ over $z_1, \ldots, z_T \sim \mathcal{D}$, it holds:

$$L_{\mathcal{D},\lambda}(\bar{\mathbf{w}}) \leq \min_{\mathbf{w} \in \mathcal{H}} L_{\mathcal{D},\lambda}(\mathbf{w}) + \frac{2\rho^2}{\delta \lambda T}(1 + \ln T). \tag{26}$$

$\square$

*Remark* 5. We assumed that the loss function is differentiable, but the proof goes through for any continuous function using subgradients.

*Remark* 6. It is possible to tighten the $\delta$ dependency on the previous bound from $\Theta(1/\delta)$ to $\Theta(\ln(1/\delta))$ by using Azuma's instead of Markov's inequality.

### E.1.1 SETUP: DATA DISTRIBUTION, ARCHITECTURE AND LEARNING ALGORITHMS

**Data** As a reminder, we consider a parity mixture distribution. Let $\mathcal{X} = \{\pm 1\}^d$ be the input space of $d \geq 2$ bits and $\mathcal{Y} = \{\pm 1, \texttt{<EOS>}\}^*$ be the output space of sequences, where $\texttt{<EOS>}$ is a special symbol denoting the end of a string. Let $\mathcal{D}(p_{\text{cot}})$ be a distribution over $\mathcal{X} \times \mathcal{Y}$, parameterized by $p_{\text{cot}} \in [0, 1]$, such that:

$$x_1, \ldots, x_d \sim \text{Rad}(1/2)$$

$$(y_1, \ldots, y_{d+1}) = Z(x_1, x_1 x_2, \ldots, \prod_{i=1}^{d} x_i, \texttt{<EOS>}) + (1 - Z) \left( \prod_{i=1}^{d} x_i, \texttt{<EOS>} \right), \quad (27)$$

where $Z \sim \text{Ber}(p_{\text{cot}})$.

**Model** We consider an architecture that consists of $d+1$ linear models. For the first output position, we consider the hypothesis class $\mathcal{H}_1 = \{\mathbf{x} \mapsto \langle \mathbf{w}_1, \mathbf{x} \rangle : \mathbf{x} \in \{\pm 1\}^d, \mathbf{w}_1 \in \mathbb{R}^d\}$. We break the second decision into two parts and consider two separate linear classes: $\mathcal{H}_{2a} = \{\mathbf{x} \mapsto \langle \mathbf{w}_{2a}, \phi_2(\mathbf{x}) \rangle + b_{2a} : \mathbf{x} \in \{\pm 1\}^{d+1}, \mathbf{w}_{2a} \in \mathbb{R}^{2d+1}, b_{2a} \in \mathbb{R}\}$ and $\mathcal{H}_2 = \{\mathbf{x} \mapsto \langle \mathbf{w}_2, \phi_2(\mathbf{x}) \rangle : \mathbf{x} \in \{\pm 1\}^{d+1}, \mathbf{w}_2 \in \mathbb{R}^{2d+1}\}$, where the first model decides between $\texttt{<EOS>}$ and $\{\pm 1\}$ while the second model decides between $-1$ and $+1$, in case the first model did not predict $\texttt{<EOS>}$. For the rest of the positions, we consider hypothesis classes $\mathcal{H}_l = \{\mathbf{x} \mapsto \langle \mathbf{w}_l, \phi_l(\mathbf{x}) \rangle : \mathbf{x} \in \{\pm 1\}^{2d+l-1}, \mathbf{w}_l \in \mathbb{R}^{2d+l-1}\}$. The feature embedding is defined as follows $\phi_l : \mathbf{x} \mapsto [x_1 \quad \ldots \quad x_{d+l-1} \quad x_{d+l-1} x_1 \quad \ldots x_{d+l-1} x_d]^T \in \{\pm 1\}^{2d+l-1}$ for $2 \leq l \leq d$. For position $d + 1$, the output is a constant function of the input, as the output symbol is always $\texttt{<EOS>}$. As learning is trivial in this case and, for not complicating the analysis any further, we assume access to this deterministic function $o : \{\pm 1\}^{2d} \mapsto \texttt{<EOS>}$ for which it holds: $o(\mathbf{x}) = \texttt{<EOS>}$ for any $\mathbf{x} \in \{\pm 1\}^{2d}$.

Our final model is a function $h$ that belongs to the linear autoregressive hypothesis class $\mathcal{H}_{\text{AR}}^{\text{Lin}}$, defined as:

$$\mathcal{H}_{\text{AR}}^{\text{Lin}} = \mathcal{H}_1 \times \mathcal{H}_{2a} \times \mathcal{H}_2 \ldots \times \mathcal{H}_d. \quad (28)$$

Note that we can learn any sparse parity inside $\mathcal{H}_{\text{AR}}^{\text{Lin}}$, hence we argue that our definitions are not particularly tailored for the problem at hand. Given an $h \in \mathcal{H}_{\text{AR}}^{\text{Lin}}$ and the corresponding $d + 1$ models, $\mathbf{w}_1, \boldsymbol{\theta}_{2a}, \mathbf{w}_2, \mathbf{w}_3, \ldots, \mathbf{w}_d$, we define a deterministic autoregressive process $\hat{h} : \mathbb{R}^d \times \mathbb{R}^d \times \mathbb{R}^{2d+2} \times \mathbb{R}^{2d+1} \times \ldots \times \mathbb{R}^{3d-1} \mapsto \{-1, +1, \epsilon, \texttt{<EOS>}\}^*$ as follows:

$$\hat{h}_1(\mathbf{x}; \mathbf{w}_1) = \text{sgn}(\langle \mathbf{w}_1, \mathbf{x} \rangle),$$

$$\hat{h}_{2a}(\mathbf{x}; \boldsymbol{\theta}_{2a}) = \texttt{dict}\left( \text{sgn}\left( \left\langle \mathbf{w}_{2a}, \phi_2\left(\mathbf{x}, \hat{h}^{(1)}(\mathbf{x}; \mathbf{w}_1)\right) \right\rangle + b_{2a} \right) \right),$$

$$\boldsymbol{\theta}_{2a} := \left( \mathbf{w}_{2a}^T, b_{2a} \right),$$

$$\hat{h}_2(\mathbf{x}; \mathbf{w}_2) = \begin{cases} \epsilon, & \text{if } \hat{h}_{2a}(\mathbf{x}; \boldsymbol{\theta}_{2a}) = \texttt{<EOS>}, \\ \text{sgn}\left( \left\langle \mathbf{w}_2, \phi_2\left(\mathbf{x}, \hat{h}^{(1)}(\mathbf{x}; \mathbf{w}_1)\right) \right\rangle \right), & \text{o.w.,} \end{cases}$$

$$\hat{h}_3(\mathbf{x}; \mathbf{w}_3) = \begin{cases} \epsilon, & \text{if } \hat{h}_{2a}(\mathbf{x}; \boldsymbol{\theta}_{2a}) = \texttt{<EOS>}, \\ \text{sgn}\left( \left\langle \mathbf{w}_3, \phi_3\left(\mathbf{x}, \hat{h}^{(1)}(\mathbf{x}; \mathbf{w}_1), \hat{h}^{(2)}(\mathbf{x}; \mathbf{w}_2)\right) \right\rangle \right), & \text{o.w.,} \end{cases}$$

$$\vdots$$

$$\hat{h}_d(\mathbf{x}; \mathbf{w}_d) = \begin{cases} \epsilon, & \text{if } \hat{h}_{2a}(\mathbf{x}; \boldsymbol{\theta}_{2a}) = \texttt{<EOS>}, \\ \text{sgn}\left( \left\langle \mathbf{w}_d, \phi_d\left(\mathbf{x}, \hat{h}^{(1)}(\mathbf{x}; \mathbf{w}_1), \ldots, \hat{h}^{(d-1)}(\mathbf{x}; \mathbf{w}_{d-1})\right) \right\rangle \right), & \text{o.w.,} \end{cases}$$

$$\hat{h}_{d+1}(\mathbf{x}) = \begin{cases} \epsilon, & \text{if } \hat{h}_{2a}(\mathbf{x}; \boldsymbol{\theta}_{2a}) = \texttt{<EOS>}, \\ o(\mathbf{x}), & \text{o.w.,} \end{cases}$$

$$\hat{h}\left(\mathbf{x}; \{\mathbf{w}_1, \boldsymbol{\theta}_{2a}, \mathbf{w}_2, \ldots, \mathbf{w}_d\}\right) = \left( \hat{h}^{(1)}(\mathbf{x}; \mathbf{w}_1), \ldots, \hat{h}^{(d+1)}(\mathbf{x}) \right),$$

$$(29)$$

where $\texttt{dict} : \{\pm 1\} \mapsto \{\varepsilon, \texttt{<EOS>}\}$ is a deterministic function (dictionary) that maps label $-1$ to the empty string and label $+1$ to the $\texttt{<EOS>}$ token. Likewise, we define a randomized sequence to sequence model $\tilde{h}$ as follows:

$$\tilde{h}_1(\mathbf{x}; \mathbf{w}_1) \sim \mathrm{Rad}\left(\frac{1}{1 + e^{-\langle \mathbf{w}_1, \mathbf{x} \rangle}}\right),$$

$$\tilde{h}_{2a}(\mathbf{x}; \boldsymbol{\theta}_{2a}) \sim \texttt{dict}\left(\mathrm{Rad}\left(\frac{1}{1 + e^{-\langle \mathbf{w}_{2a}, \phi_2(\mathbf{x}, \tilde{h}_1(\mathbf{x}; \mathbf{w}_1)) \rangle - b_{2a}}}\right)\right),$$

$$\boldsymbol{\theta}_{2a} := \left(\mathbf{w}_{2a}^T, b_{2a}\right),$$

$$\tilde{h}_2(\mathbf{x}; \mathbf{w}_2) \sim \begin{cases} \epsilon, \text{if } \tilde{h}_{2a}(\mathbf{x}) = \texttt{<EOS>}, \\ \mathrm{Rad}\left(\frac{1}{1 + e^{-\langle \mathbf{w}_2, \phi_2(\mathbf{x}, \tilde{h}_1(\mathbf{x}; \mathbf{w}_1)) \rangle}}\right), \text{ o.w.,} \end{cases}$$

$$\tilde{h}_3(\mathbf{x}; \mathbf{w}_3) \sim \begin{cases} \epsilon, \text{if } \tilde{h}_{2a}(\mathbf{x}) = \texttt{<EOS>}, \\ \mathrm{Rad}\left(\frac{1}{1 + e^{-\langle \mathbf{w}_3, \phi_3(\mathbf{x}, \tilde{h}_1(\mathbf{x}; \mathbf{w}_1), \tilde{h}_2(\mathbf{x}; \mathbf{w}_2)) \rangle}}\right), \text{ o.w.,} \end{cases} \tag{30}$$

$$\vdots$$

$$\tilde{h}_d(\mathbf{x}; \mathbf{w}_d) \sim \begin{cases} \epsilon, \text{if } \tilde{h}_{2a}(\mathbf{x}) = \texttt{<EOS>}, \\ \mathrm{Rad}\left(\frac{1}{1 + e^{-\langle \mathbf{w}_d, \phi_d(\mathbf{x}, \tilde{h}_1(\mathbf{x}; \mathbf{w}_1), \dots, \tilde{h}_{d-1}(\mathbf{x}; \mathbf{w}_{d-1})) \rangle}}\right), \text{ o.w.,} \end{cases}$$

$$\tilde{h}_{d+1}(\mathbf{x}) = \begin{cases} \epsilon, \text{ if } \tilde{h}_{2a}(\mathbf{x}) = \texttt{<EOS>}, \\ o(\mathbf{x}), \text{ o.w.,} \end{cases}$$

$$\tilde{h}(\mathbf{x}; \{\mathbf{w}_1, \boldsymbol{\theta}_{2a}, \mathbf{w}_2, \dots, \mathbf{w}_d\}) = \left(\tilde{h}_1(\mathbf{x}; \mathbf{w}_1), \dots, \tilde{h}_{d+1}(\mathbf{x})\right).$$

We adopt the convention that $\alpha \epsilon \beta = \alpha \beta$, i.e., concatenation with the empty string character $\epsilon$ does not change a string. We define the probability measure induced by $h$ as:

$$\pi_h(y \mid \mathbf{x}) = \begin{cases} \mathbb{P}\left[\tilde{h}_1(\mathbf{x}; \mathbf{w}_1) = y_1\right] \cdot \mathbb{P}\left[\tilde{h}_{2a}(\mathbf{x}; \boldsymbol{\theta}_{2a}) = \texttt{<EOS>} \,\middle|\, \tilde{h}_1(\mathbf{x}; \mathbf{w}_1) = y_1\right], & \text{if } |y| = 2, \\ \mathbb{P}\left[\tilde{h}_1(\mathbf{x}; \mathbf{w}_1) = y_1\right] \cdot \dots \cdot \mathbb{P}\left[\tilde{h}_d(\mathbf{x}; \mathbf{w}_d) = y_d \,\middle|\, \tilde{h}_1(\mathbf{x}; \mathbf{w}_1) = y_1, \dots, \\ \qquad\qquad\qquad\qquad\qquad \tilde{h}_{d-1}(\mathbf{x}; \mathbf{w}_{d-1}) = y_{d-1}\right], & \text{if } |y| > 2. \end{cases} \tag{31}$$

For any other $y \in \{\pm 1, \epsilon, \texttt{<EOS>}\}^*$, it is $\pi_h(y \mid \mathbf{x}) = 0$.

The two autoregressive models, $\hat{h}$ and $\tilde{h}$, correspond to *greedy decoding* and *sampling with temperature 1*, respectively.

**Pre-training loss function** During pre-training, the loss function is the next token prediction objective, together with an $\ell_2$ regularization term. Let $(\mathbf{x}, y) \sim \mathcal{D}(p_{\mathrm{cot}})$. We define the loss functions corresponding to each linear model:

1. Position 1: $l^{(1)}(\mathbf{w}_1, (\mathbf{x}, y_1)) = \ln\left(1 + e^{-y\langle \mathbf{w}_1, \mathbf{x} \rangle}\right) + \frac{\lambda_1}{2}\|\mathbf{w}_1\|^2$, $\lambda_1 > 0$.

2. Position 2a: $l^{(2a)}\left((\mathbf{w}_{2a}, b_{2a}), ((\mathbf{x}, y_1), \tilde{y}_2)\right) = \ln\left(1 + e^{-y(\langle \mathbf{w}_{2a}, \phi_2((\mathbf{x}, y_1)) \rangle + b_{2a})}\right) + \frac{\lambda_{2a}}{2}\left(\|\mathbf{w}_{2a}\|^2 + b_{2a}^2\right)$ with $\lambda_{2a} > 0$ and $\tilde{y}_2 = \begin{cases} +1, & y_2 = \texttt{<EOS>}, \\ -1, & y_2 \in \{\pm 1\}. \end{cases}$

3. Positions 2 to $d$:

$$l^{(l)}(\mathbf{w}_l, ((\mathbf{x}, y_1, \dots, y_{l-1}), y_l)) = \ln\left(1 + e^{-y\langle \mathbf{w}_l, \phi_l((\mathbf{x}, y_1, \dots, y_{l-1})) \rangle}\right) + \frac{\lambda_l}{2}\|\mathbf{w}_l\|^2,$$

with $\lambda_l > 0$ for all $2 \leq l \leq d$.

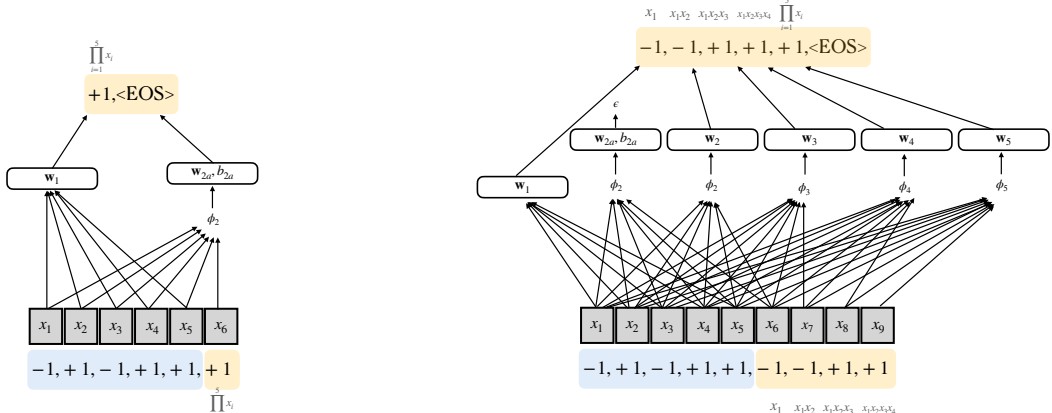

Figure 40: An illustration of next-token prediction training (for $d$=5) with the linear (time-inhomogeneous) autoregressive architecture in the case of a short (left) and long (right) training sequence.

We included the regularization term in the definition of the loss functions. Then, we seek to solve the following optimization problem:

$$
\min_{\mathbf{w}_1, \mathbf{w}_{2a}, b, \mathbf{w}_2, \ldots, \mathbf{w}_d} \mathbb{E}_{(\mathbf{x}, y) \sim \mathcal{D}(p_{\mathrm{cot}})} \Bigg[ \mathbb{1}\{|y| = 2\} \left( l^{(1)}(\mathbf{w}_1, (\mathbf{x}, y_1)) + l^{(2a)}(\{\mathbf{w}_{2a}, b\}, ((\mathbf{x}, y_1), \tilde{y}_2)) \right)
$$

$$
+ \mathbb{1}\{|y| = d+1\} \left( l^{(1)}(\mathbf{w}_1, (\mathbf{x}, y_1)) + \ldots + l^{(d)}(\mathbf{w}_d, ((\mathbf{x}, y_1, \ldots, y_{d-1}), y_d)) \right) \Bigg].
$$

$$\text{(LIN-NTP)}$$

Note that the regularization term can have a different coefficient for each parameter vector. Also, observe that Problem LIN-NTP corresponds to $d$+1 binary classification problems with respect to the logistic loss.

**Pre-training Algorithm** We minimize the previous objective with Stochastic Gradient Descent – see Algorithm 3.

---

**Algorithm 3** Stochastic Gradient Descent (SGD) for solving Problem equation LIN-NTP

---

**Require:** Integers $T, T_1, T_{2a}, T_2, \ldots, T_d > 0$, Real numbers $\lambda_1, \lambda_{2a}, \lambda_2, \ldots, \lambda_d > 0$.

1: Initialize $\mathbf{w}_1^{(1)}, \mathbf{w}_{2a}^{(1)}, \mathbf{w}_2^{(1)}, \ldots, \mathbf{w}_d^{(1)} = \mathbf{0}, b_{2a} = 0$

2: Set $t_{\text{long}} = 0$

3: **for** $t = 1, 2, \ldots, T$ **do**

4:      Sample $(\mathbf{x}, y) \sim \mathcal{D}(p_{\text{cot}})$

5:      **if** $t \leq T_1$ **then**

6:          Set $\eta_t = \frac{1}{\lambda_1 t}$

7:          Set $\mathbf{w}_1^{(t+1)} = \mathbf{w}_1^{(t)} - \eta_t \nabla_{\mathbf{w}_1^{(t)}} l^{(1)} \left( \mathbf{w}_1^{(t)}, (\mathbf{x}, y_1) \right)$

8:      **end if**

9:      **if** $t \leq T_{2a}$ **then**

10:          Set $\tilde{y}_2 = \begin{cases} +1, & \text{if } y_2 = \texttt{<EOS>}, \\ -1, & \text{if } y_2 \in \{\pm 1\} \end{cases}$

11:          Set $\eta_t = \frac{1}{\lambda_{2a} t}$

12:          Set $\mathbf{w}_{2a}^{(t+1)} = \mathbf{w}_{2a}^{(t)} - \eta_t \nabla_{\mathbf{w}_{2a}^{(t)}} l^{(2a)} \left( \left\{ \mathbf{w}_{2a}^{(t)}, b_{2a}^{(t)} \right\}, ([\mathbf{x}, y_1], \tilde{y}_2) \right)$

13:          Set $b_{2a}^{(t+1)} = b_{2a}^{(t)} - \eta_t \frac{\partial}{\partial b_{2a}} l^{(2a)} \left( \left\{ \mathbf{w}_{2a}^{(t)}, b_{2a}^{(t)} \right\}, ([\mathbf{x}, y_1], \tilde{y}_2) \right)$

14:      **end if**

15:      **if** $|y| > 2$ **then**

16:          Set $t_{\text{long}} = t_{\text{long}} + 1$

17:          **for** $l = 2, \ldots, d$ **do**

18:              **if** $t_{\text{long}} \leq T_l$ **then**

19:                  Set $\eta_{t_{\text{long}}} = \frac{1}{\lambda_l t_{\text{long}}}$

20:                  Set $\mathbf{w}_l^{(t_{\text{long}}+1)} = \mathbf{w}_l^{(t_{\text{long}})} - \eta_{t_{\text{long}}} \nabla_{\mathbf{w}_l^{(t_{\text{long}})}} l^{(l)} \left( \mathbf{w}_l^{(t_{\text{long}})}, ([\mathbf{x}, y_1, \ldots, y_{l-1}], y_l) \right)$

21:              **end if**

22:          **end for**

23:      **end if**

24: **end for**

25: Output $\bar{\mathbf{w}}_1 = \frac{1}{T_1} \sum_{t=1}^{T_1} \mathbf{w}_1^{(t)}, \bar{\mathbf{w}}_{2a} = \frac{1}{T_{2a}} \sum_{t=1}^{T_{2a}} \mathbf{w}_{2a}^{(t)}, \bar{b}_{2a} = \frac{1}{T_{2a}} \sum_{t=1}^{T_{2a}} b_{2a}^{(t)}, \bar{\mathbf{w}}_2 = \frac{1}{T_2} \sum_{t=1}^{T_2} \mathbf{w}_2^{(t)}, \ldots, \bar{\mathbf{w}}_d = \frac{1}{T_d} \sum_{t=1}^{T_d} \mathbf{w}_d^{(t)}$.

---

**Post-training loss function** For post-training, we consider the STaR algorithm (Zelikman et al., 2022). Recall that the STaR algorithm involves $n$ reinforcement learning rounds, where each round involves optimization of a next-token prediction loss on model sampled responses (from the model of the previous round) that are correct according to some reward function. We use a reward $r_{\text{cot}} : \mathcal{X} \times \mathcal{Y}$ that assesses whether the whole sequence is valid:

$$r_{\text{cot}}(\mathbf{x}, y) = \mathbb{1}\left\{ y = \left( x_1, x_1 x_2, \ldots, \prod_{i=1}^{d} x_i, \texttt{<EOS>} \right) \vee y = \left( \prod_{i=1}^{d} x_i, \texttt{<EOS>} \right) \right\}. \tag{32}$$

Namely, at the $k + 1$ round, we minimize the following objective:

$$\mathbb{E}_{\substack{\mathbf{x} \sim \text{Rad}(1/2)^{\otimes d}, \\ y \sim \pi_{h^{(k)}}(\cdot | \mathbf{x})}} \left[ \mathbb{1}\left\{ |y| = 2 \right\} \left( l^{(1)}(\mathbf{w}_1, (\mathbf{x}, y_1)) + l^{(2a)}(\{\mathbf{w}_{2a}, b\}, ([\mathbf{x}, y_1], \tilde{y}_2)) \right) \right.$$

$$+ \mathbb{1}\left\{ |y| = d + 1 \right\} \left( l^{(1)}(\mathbf{w}_1, (\mathbf{x}, y_1)) + l^{(2a)}(\{\mathbf{w}_{2a}, b\}, ([\mathbf{x}, y_1], \tilde{y}_2)) + \ldots \right.$$

$$\left. + \ldots + l^{(d)}(\mathbf{w}_d, ([\mathbf{x}, y_1, \ldots, y_{d-1}], y_d)) \right) \Big| r_{\text{cot}}(\mathbf{x}, y) = 1 \Big],$$

$$\text{(LIN-RL)}$$

where $h^{(k)}$ is the model returned at the end of the $k$'th round.

**Post-training Algorithm** The algorithm consists of $n$ rounds, where at each one we minimize the previous objective with Stochastic Gradient Descent. At each round, we start optimization from a freshly initialized model at the origin.

### E.2 WHY IS THE CRITICAL THRESHOLD 1/3?

Before proceeding with the theoretical analysis of the pre-training and post-training stages, we explain why we should expect the critical threshold of pre-training to equal 1/3. Our argument is in some sense generic – see Lemma 1 below.

Recall that from the definition of $\mathcal{D}(p_{\text{cot}})$, we have for the distribution of the first token: $y_1 = Zx_1 + (1-Z)\prod_{i=1}^{d} x_i$, where $Z \sim \text{Ber}(p_{\text{cot}})$. We have:

$$\mathbb{P}\left[y_1 = +1 \mid \mathbf{x}\right] = p_{\text{cot}}\mathbb{1}\left\{x_1 = +1\right\} + (1-p_{\text{cot}})\mathbb{1}\left\{\prod_{i=1}^{d} x_i = +1\right\}$$

$$= p_{\text{cot}}\frac{x_1+1}{2} + (1-p_{\text{cot}})\frac{\prod_{i=1}^{d} x_i + 1}{2}. \tag{33}$$

Any model that matches this conditional probability should be able to implement the parity function $\mathbf{x} \mapsto \prod_{i=1}^{d} x_i$. In absence of such a parity feature, the conditional probability simplifies to:

$$\mathbb{P}\left[y_1 = +1 \mid \mathbf{x}\right] = p_{\text{cot}}\frac{x_1+1}{2} + \frac{1-p_{\text{cot}}}{2} = \frac{p_{\text{cot}}x_1+1}{2}. \tag{34}$$

In other words, $\mathbb{P}\left[y_1 = x_1 \mid \mathbf{x}\right] = \frac{p_{\text{cot}}+1}{2} > 1/2$ for all $p_{cot} > 0$. Hence, for any model that matches this conditional probability, the "greedy" decision on the first token will be $x_1$, since $x_1 = \arg\max_{c \in \{x_1, -x_1\}} \mathbb{P}\left[y_1 = c \mid \mathbf{x}\right]$.

For the second position, we have:

$$\mathbb{P}\left[y_2 = x_1x_2 \mid \mathbf{x}, y_1 = x_1\right] = \frac{\mathbb{P}\left[y_2 = x_1x_2, y_1 = x_1 \mid \mathbf{x}\right]}{\mathbb{P}\left[y_1 = x_1 \mid \mathbf{x}\right]} = \frac{p_{\text{cot}}}{\frac{p_{\text{cot}}+1}{2}} = \frac{2p_{\text{cot}}}{p_{\text{cot}}+1}, \tag{35}$$

and also $\mathbb{P}\left[y_2 = \texttt{<EOS>} \mid \mathbf{x}, y_1 = x_1\right] = 1 - \mathbb{P}\left[y_2 = x_1x_2 \mid \mathbf{x}, y_1 = x_1\right] = \frac{1-p_{\text{cot}}}{p_{\text{cot}}+1}$. Hence, as long as $1 - p_{\text{cot}} > 2p_{\text{cot}} \iff p_{\text{cot}} < 1/3$, the "greedy" generation of a model will output the sequence $(x_1, \texttt{<EOS>})$.

We formally explain now why all logistic regression models without a parity feature will match the conditional probability expression of eq. equation 34. The proof uses simple ideas from the Fourier analysis of Boolean functions.

**Lemma 1.** *Denote the parity functions with support $S \subseteq [d]$ by $\chi_S(\mathbf{x}) = \prod_{i \in S} x_i$. Let $\mathcal{F} = \text{span}\left\{\chi_S : S \subseteq [d], S \neq [d]\right\}$ be the class of functions spanned by all but the complete parity $\chi_{[d]} = \prod_{i=1}^{d} x_i$. Then, if $f^\star = \arg\min_{f \in \mathcal{F}} \mathbb{E}_{\mathbf{x}\sim\text{Rad}(1/2)^{\otimes d}, y\sim Zx_1+(1-Z)\prod_{i=1}^{d} x_i}\left[\log\left(1 + e^{-yf(\mathbf{x})}\right)\right]$ is the output of logistic regression in class $\mathcal{F}$, then it holds:*

$$\mathbb{P}_{f^\star}\left[y = +1|\mathbf{x}\right] := \frac{1}{1 + e^{-f^\star(\mathbf{x})}} = \frac{p_{\text{cot}}x_1+1}{2}. \tag{36}$$

*Proof.* We write any function $f \in \mathcal{F}$ in its Fourier expansion: $f(\mathbf{x}) = \sum_{S \subset [d]} c_S \chi_S(\mathbf{x})$, where $c_S = \mathbb{E}_{\mathbf{x}\sim\text{Rad}(1/2)^{\otimes d}}\left[f(\mathbf{x})\chi_S(\mathbf{x})\right]$. Then, the objective becomes:

$$\mathcal{L}\left(c_\emptyset, \ldots, c_{[d-1]}\right) = \mathbb{E}_{\mathbf{x}\sim\text{Rad}(1/2)^{\otimes d}, y\sim Zx_1+(1-Z)\prod_{i=1}^{d} x_i}\log\left(1 + e^{-y\sum_{S\subset[d]} c_S\chi_S(\mathbf{x})}\right). \tag{37}$$

This is a strictly convex objective with respect to the Fourier coefficients. Denote by $\eta(\mathbf{x}) = \mathbb{P}\left[y_1 = +1 \mid \mathbf{x}\right] = p_{\text{cot}}\frac{x_1+1}{2} + (1-p_{\text{cot}})\frac{\prod_{i=1}^{d} x_i+1}{2}$. We calculate the partial derivatives for all $S \subset [d]$:

$$\frac{\partial L}{\partial c_S} = \mathbb{E}_{\mathbf{x}\sim\text{Rad}(1/2)^{\otimes d}, y\sim Zx_1+(1-Z)\prod_{i=1}^{d} x_i}\left[\frac{\chi_S(\mathbf{x})e^{-y\sum_{S'\subset[d]} c_{S'}\chi_{S'}(\mathbf{x})}}{1 + e^{-y\sum_{S'\subset[d]} c_{S'}\chi_{S'}(\mathbf{x})}}\right]$$

$$= \mathbb{E}_{\mathbf{x}\sim\text{Rad}(1/2)^{\otimes d}}\left[\left(\frac{1}{1 + e^{-\sum_{S'\subset[d]} c_{S'}\chi_{S'}(\mathbf{x})}} - \eta(\mathbf{x})\right)\chi_S(\mathbf{x})\right]. \tag{38}$$

Therefore, at the optimum, we have $2^d - 1$ orthogonality conditions,

$$\widehat{\sigma \circ f^\star}(S) := \mathbb{E}_{\mathbf{x} \sim \text{Rad}(1/2)^{\otimes d}} \left[ \frac{1}{1 + e^{-\sum_{S' \subset [d]} c^\star_{S'} \chi_{S'}(\mathbf{x})}} \chi_S(\mathbf{x}) \right] = \mathbb{E}_{\mathbf{x} \sim \text{Rad}(1/2)^{\otimes d}} \left[ \eta(\mathbf{x}) \chi_S(\mathbf{x}) \right],$$
(39)

for all $S \subset [d]$. In particular, if we denote $\sigma(u) = \frac{1}{1+e^{-u}}$, then: $\widehat{\sigma \circ f^\star}(\emptyset) = \frac{1}{2}, \widehat{\sigma \circ f^\star}(\{1\}) = \frac{p_{\text{cot}}}{2}$ and $\widehat{\sigma \circ f^\star}(S) = 0$ for $S \neq \{\emptyset, \{1\}, [d]\}$. Observe that the full parity coefficient is unconstrained. Due to symmetry, we can observe the form of the optimal solution $f^\star(\mathbf{x}) = \alpha x_1 + \beta$, then the optimality conditions yield:

$$\mathbb{E}_{\mathbf{x} \sim \text{Rad}(1/2)^{\otimes d}} \left[ \frac{1}{1 + e^{-\alpha x_1 + \beta}} \right] = \frac{1}{2},$$
(40)

or:

$$\sigma(\alpha - \beta) + \sigma(-\alpha - \beta) = 1$$
(41)

This implies that $\beta = 0$. From the second equation:

$$\mathbb{E}_{\mathbf{x} \sim \text{Rad}(1/2)^{\otimes d}} \left[ \frac{x_1}{1 + e^{-\alpha x_1}} \right] = \frac{p_{\text{cot}}}{2},$$
(42)

which implies $\alpha = \ln \left( \frac{1+p_{\text{cot}}}{1-p_{\text{cot}}} \right)$. This is the only possible solution as the objective is strictly convex. Therefore, we showed that:

$$\mathbb{P}_{f^\star} \left[ y = +1 | \mathbf{x} \right] = \frac{1}{1 + e^{-\ln\left(\frac{1+p_{\text{cot}}}{1-p_{\text{cot}}}\right)^{x_1}}} = \frac{(1 + p_{\text{cot}})^{x_1}}{(1 + p_{\text{cot}})^{x_1} + (1 - p_{\text{cot}})^{x_1}} = \frac{p_{\text{cot}} x_1 + 1}{2}.$$
(43)

$\square$

The intuition is that a "reasonable" architecture, such as an auto-regressive transformer, requires exponentially many iterations to form a parity feature (and therefore to "escape" $\mathcal{F}$). As a result, a pre-trained transformer will exhibit the same threshold.

### E.3 PRE-TRAINING

We proceed by providing guarantees for each one of the models $\bar{\mathbf{w}}_1, \bar{\boldsymbol{\theta}}_{2a}, \bar{\mathbf{w}}_2, \bar{\mathbf{w}}_3, \ldots, \bar{\mathbf{w}}_d$ independently, and then state and prove our main pre-training theorem for the behavior of induced models $\hat{h}, \tilde{h}$, leveraging the per-position results. All results are stated for a mixture coefficient $p \in (0, 3/4)$. The upper bound needs to be greater than $1/2$ to obtain the post-training result later on, but the specific 3/4 value is arbitrary.

#### E.3.1 POSITION 1

For the first position of the output, we consider a binary classification problem where $\mathcal{X}_1 = \{\pm 1\}^d, \mathcal{Y}_1 = \{\pm 1\}$ with a distribution $\mathcal{D}_1(p)$ over $\mathcal{X}_1 \times \mathcal{Y}_1$ such that: $x_1, \ldots, x_d \sim \text{Rad}(1/2)$ and $y = Zx_1 + (1 - Z) \prod_{i=1}^d x_i$ where $Z \sim \text{Ber}(p)$, $0 < p < 3/4$. We consider the hypothesis class $\mathcal{H}_1 = \{\mathbf{x} \mapsto \langle \mathbf{w}, \mathbf{x} \rangle : \mathbf{w} \in \mathbb{R}^d\}$ and the logistic loss plus an additional $\ell_2$ regularization term: $l^{(1)}(\mathbf{w}, (\mathbf{x}, y)) = \ln\left(1 + e^{-y\langle \mathbf{w}, \mathbf{x} \rangle}\right) + \frac{\lambda_1}{2} \|\mathbf{w}\|^2$, $\lambda_1 > 0$.

We prove the following guarantees on the hypothesis returned by SGD.

**Proposition 1.** *Consider running SGD (Algorithm 2) for minimizing* $L_{\mathcal{D}_1(p), \lambda_1}(\mathbf{w}) = \mathbb{E}_{(\mathbf{x}, y) \sim \mathcal{D}_1(p)} \left[ \ln\left(1 + e^{-y\langle \mathbf{w}, \mathbf{x} \rangle}\right) \right] + \frac{\lambda_1}{2} \|\mathbf{w}\|^2$ *with* $\lambda_1 > 0$. *Then, after* $T_1$ *iterations and for any* $\delta \in (0, 1)$, *with probability at least* $1 - \delta$ *over the sampled* $\{(\mathbf{x}_i, y_i)\}_{i=1}^{T_1} \sim \mathcal{D}_1(p)$, *it holds for all* $\mathbf{x} \in \{\pm 1\}^d$:

$$\left| \left\langle \bar{\mathbf{w}}_1 - \ln\left(\frac{1+p}{1-p}\right) \mathbf{e}_1, \mathbf{x} \right\rangle \right| \leq \frac{2d}{\lambda_1} \sqrt{\frac{1 + \ln T_1}{\delta T_1}} + \frac{4 \ln\left(\frac{1+p}{1-p}\right)}{1 - p^2} \lambda_1,$$

$$\left| \frac{1}{1 + e^{-\langle \bar{\mathbf{w}}_1, \mathbf{x} \rangle}} - \frac{1}{1 + \left(\frac{1-p}{1+p}\right)^{x_1}} \right| < \frac{d}{2\lambda_1} \sqrt{\frac{1 + \ln T_1}{\delta T_1}} + \frac{\ln\left(\frac{1+p}{1-p}\right)}{1 - p^2} \lambda_1,$$
(44)

where $\bar{\mathbf{w}}_1$ is the output of SGD and $\mathbf{e}_1 = [1, 0, \ldots, 0]^T \in \mathbb{R}^d$.

*Proof.* We first show that learning $\mathcal{H}_1$ with respect to (unregularized loss) $\ln\left(1 + e^{-y(\langle \mathbf{w}, \mathbf{x}\rangle)}\right)$ corresponds to a Convex and Lipschitz learning problem. The loss is convex with respect to its first argument. For the Lipschitz constant, we have for all $\mathbf{x} \in \{\pm 1\}^d$, $y \in \{\pm 1\}$ and $\mathbf{w} \in \mathbb{R}^d$:

$$\left\| \nabla_{\mathbf{w}} \ln\left(1 + e^{-y\langle \mathbf{w}, \mathbf{x}\rangle}\right) \right\| = \left\| -\frac{y\mathbf{x}}{1 + e^{y\langle \mathbf{w}, \mathbf{x}\rangle}} \right\| \leq \sqrt{d}. \tag{45}$$

Therefore, applying Theorem 3, we have that SGD after $T_1$ iterations returns a hypothesis $\bar{\mathbf{w}}_1$ such that for any $\delta \in (0, 1)$ with probability at least $1 - \delta$ it holds:

$$L_{\mathcal{D}_1(p), \lambda_1}(\bar{\mathbf{w}}_1) \leq L_{\mathcal{D}_1(p), \lambda_1}(\hat{\mathbf{w}}) + \frac{2d}{\delta \lambda_1 T_1}\left(1 + \ln T_1\right), \tag{46}$$

where $\hat{\mathbf{w}} = \arg\min_{\mathbf{w} \in \mathcal{H}_1} L_{\mathcal{D}_1(p), \lambda_1}(\mathbf{w})$. From the strong convexity of $L_{\mathcal{D}_1(p), \lambda_1}$, this implies:

$$\left\|\bar{\mathbf{w}}_1 - \hat{\mathbf{w}}\right\|^2 \leq \frac{2}{\lambda_1}\left(L_{\mathcal{D}_1(p), \lambda_1}(\bar{\mathbf{w}}_1) - L_{\mathcal{D}_1(p), \lambda_1}(\hat{\mathbf{w}})\right) \leq \frac{4d}{\lambda_1^2 \delta T_1}\left(1 + \ln T_1\right). \tag{47}$$

The previous bound on the parameter space can be translated to a guarantee on the estimated probability of the output being equal to 1. Recall that for a hypothesis $\mathbf{w}$, this probability is defined as the output of the hypothesis passed through the sigmoid function $\sigma(u) = \frac{1}{1+e^{-u}}$:

$$p_1(\mathbf{w} \mid \mathbf{x}) = \sigma(\langle \mathbf{w}, \mathbf{x}\rangle) = \frac{1}{1 + e^{-\langle \mathbf{w}, \mathbf{x}\rangle}}. \tag{48}$$

We calculate the Lipschitz constant of $p_1(\mathbf{w} \mid \mathbf{x})$. For any $\mathbf{x} \in \{\pm 1\}^d$, $\mathbf{w} \in \mathbb{R}^d$, we have:

$$\left\|\nabla_{\mathbf{w}} p_1(\mathbf{w} \mid \mathbf{x})\right\| = \left\|\frac{\mathbf{x} e^{-\langle \mathbf{w}, \mathbf{x}\rangle}}{\left(1 + e^{-\langle \mathbf{w}, \mathbf{x}\rangle}\right)^2}\right\| \leq \frac{\sqrt{d}}{4}. \tag{49}$$

Therefore, combining eqs. 47, 49, we have:

$$\left|p_1(\bar{\mathbf{w}}_1 \mid \mathbf{x}) - p_1(\hat{\mathbf{w}} \mid \mathbf{x})\right| \leq \frac{\sqrt{d}}{4}\left\|\bar{\mathbf{w}}_1 - \hat{\mathbf{w}}\right\| \leq \frac{d}{2\lambda_1}\sqrt{\frac{1 + \ln T_1}{\delta T_1}}. \tag{50}$$

It remains to estimate the value of $p_1(\hat{\mathbf{w}} \mid \mathbf{x})$. In order to find the minimizer $\hat{\mathbf{w}} = \arg\min_{\mathbf{w} \in \mathcal{H}_1} L_{\mathcal{D}_1(p), \lambda_1}(\mathbf{w})$, we set the gradient of $L_{\mathcal{D}_1(p), \lambda_1}(\mathbf{w})$ to zero:

$$\mathbb{E}_{(\mathbf{x}, y) \sim \mathcal{D}_1(p)}\left[\frac{-y\mathbf{x}}{1 + e^{y\langle \hat{\mathbf{w}}, \mathbf{x}\rangle}}\right] + \lambda_1 \hat{\mathbf{w}} = 0. \tag{51}$$

The objective $L_{\mathcal{D}_1(p), \lambda_1}(\mathbf{w})$ is strongly convex, so it admits a unique solution. We observe that this solution is of the form $\hat{\mathbf{w}} = \alpha e_1$, $\alpha \in \mathbb{R}$, where $e_1 = [1, 0, \ldots, 0]^T$. Indeed, the optimality conditions become:

$$\begin{cases} \mathbb{E}_{(\mathbf{x}, y) \sim \mathcal{D}_1(p)}\left[\frac{-yx_1}{1 + e^{\alpha y x_1}}\right] + \lambda_1 \alpha = 0, \\ \mathbb{E}_{(\mathbf{x}, y) \sim \mathcal{D}_1(p)}\left[\frac{-yx_i}{1 + e^{\alpha y x_1}}\right] = 0, \quad i = 2, \ldots, d. \end{cases} \tag{52}$$

The last $d - 1$ equations are satisfied as $\mathbb{E}_{(\mathbf{x}, y) \sim \mathcal{D}_1(p)}[yx_i] = 0$ for all $i = 2, \ldots, d$. The first equation simplifies to:

$$g(\alpha) := \frac{1}{1 + e^{-\alpha}} + \lambda_1 \alpha - \frac{p+1}{2} = 0. \tag{53}$$

This equation has, indeed, a unique root as $g$ is continuous, $g'(u) = \frac{e^{-u}}{(1+e^{-u})^2} + \lambda_1 > 0$ for all $\lambda_1 > 0$, $u \in \mathbb{R}$ and $\lim_{u \to -\infty} g(u) = -\infty$, $\lim_{u \to +\infty} g(u) = +\infty$. Furthermore, $g(0) = -\frac{p}{2} < 0$, hence $\alpha > 0$. This proves that $\hat{\mathbf{w}} = \alpha e_1$, where $\alpha > 0$ is such that $g(\alpha) = 0$. Furthermore, let $\alpha_0 = \ln\left(\frac{1+p}{1-p}\right)$ be the weight of the unregularized solution $\hat{\mathbf{w}}_0$; that is, $\alpha_0$ is the solution of $g(\alpha_0) = 0$ for $\lambda_1 = 0$. We have $g(\alpha) - g(\alpha_0) = -\lambda_1 \alpha < 0$ which implies that $\alpha < \alpha_0$. From the mean value theorem there exists $\xi \in (\alpha, \alpha_0)$ such that:

$$\sigma'(\xi) = \frac{\sigma(\alpha_0) - \sigma(\alpha)}{\alpha_0 - \alpha} = \frac{\lambda_1 \alpha}{\alpha_0 - \alpha}. \tag{54}$$

But, observe that for any $u \in [0, \alpha_0]$, the derivative of the sigmoid is bounded as follows: $\frac{1-p^2}{4} \leq \sigma'(u) \leq \frac{1}{4}$. Therefore, for any $\lambda_1 > 0$, it holds:

$$|\alpha - \alpha_0| \leq \frac{4\alpha}{1-p^2}\lambda_1 < \frac{4\alpha_0}{1-p^2}\lambda_1 = \frac{4\ln\left(\frac{1+p}{1-p}\right)}{1-p^2}\lambda_1. \tag{55}$$

Given the above, we can bound the calibration of $\hat{\mathbf{w}}$. For any $\lambda_1 > 0$ and $\mathbf{x} \in \{\pm1\}^d$ we have:

$$\left|\frac{1}{1+e^{-\alpha x_1}} - \frac{1}{1+e^{-\alpha_0 x_1}}\right| \leq \frac{1}{4}|\alpha - \alpha_0| \tag{56}$$
$$< \frac{\ln\left(\frac{1+p}{1-p}\right)}{1-p^2}\lambda_1.$$

Combining the above with equation 50, we finally obtain that for any $\lambda_1 > 0$ and $\mathbf{x} \in \{\pm1\}^d$, it holds:

$$\left|\frac{1}{1+e^{-\langle\bar{\mathbf{w}}_1,\mathbf{x}\rangle}} - \frac{1}{1+\left(\frac{1-p}{1+p}\right)^{x_1}}\right| = |p_1(\bar{\mathbf{w}}_1 \mid \mathbf{x}) - p_1(\hat{\mathbf{w}}_0 \mid \mathbf{x})|$$
$$\leq |p_1(\bar{\mathbf{w}}_1 \mid \mathbf{x}) - p_1(\hat{\mathbf{w}} \mid \mathbf{x})| + |p_1(\hat{\mathbf{w}} \mid \mathbf{x}) - p_1(\hat{\mathbf{w}}_0 \mid \mathbf{x})| \tag{57}$$
$$< \frac{d}{2\lambda_1}\sqrt{\frac{1+\ln T_1}{\delta T_1}} + \frac{\ln\left(\frac{1+p}{1-p}\right)}{1-p^2}\lambda_1.$$

Furthermore, for the optimality of $\bar{\mathbf{w}}_1$ in parameter space, we have:

$$|\langle\bar{\mathbf{w}}_1 - \alpha_0\mathbf{e}_1, \mathbf{x}\rangle| = |\langle\bar{\mathbf{w}}_1 - \alpha\mathbf{e}_1, \mathbf{x}\rangle - \langle\alpha_0\mathbf{e}_1 - \alpha\mathbf{e}_1, \mathbf{x}\rangle|$$
$$\leq \sqrt{d}\|\bar{\mathbf{w}}_1 - \hat{\mathbf{w}}\| + |\alpha - \alpha_0|. \tag{58}$$

From eqs. 47 and 55, we obtain:

$$|\langle\bar{\mathbf{w}}_1 - \alpha_0\mathbf{e}_1, \mathbf{x}\rangle| \leq \frac{2d}{\lambda_1}\sqrt{\frac{1+\ln T_1}{\delta T_1}} + \frac{4\ln\left(\frac{1+p}{1-p}\right)}{1-p^2}\lambda_1. \tag{59}$$

$\square$

The guarantees of equation 44 quantify the calibration of the model returned by SGD and its proximity to the best-in-class model in parameter space. We obtain the following explicit corollary on the calibration and "hard" prediction of the model.

**Corollary 1.** *Consider running SGD (Algorithm 2) for minimizing* $L_{\mathcal{D}_1(p),\lambda_1}(\mathbf{w}) = \mathbb{E}_{(\mathbf{x},y)\sim\mathcal{D}_1(p)}\left[\ln\left(1 + e^{-y\langle\mathbf{w},\mathbf{x}\rangle}\right)\right] + \frac{\lambda_1}{2}\|\mathbf{w}\|^2$. *Then, for any* $\varepsilon > 0$, *if* $\lambda_1 = \tilde{\Theta}\left(\frac{d^{1/2}}{\delta^{1/4}T_1^{1/4}}\right)$, *and after* $T_1 = \tilde{O}\left(\frac{d^2}{\delta\varepsilon^4}\right)$ *iterations, with probability at least* $1 - \delta$ *over the sampled* $\{(\mathbf{x}_i, y_i)\}_{i=1}^{T_1} \sim \mathcal{D}_1(p)$, *it holds:*

$$\left|\frac{1}{1+e^{-\langle\bar{\mathbf{w}}_1,\mathbf{x}\rangle}} - \frac{1}{1+\left(\frac{1-p}{1+p}\right)^{x_1}}\right| < \varepsilon, \tag{60}$$

*for all* $\mathbf{x} \in \{\pm1\}^d$. *If, additionally,* $\varepsilon < \ln\left(\frac{1+p}{1-p}\right)$, *then with probability* $1 - \delta$:

$$\mathrm{sgn}\left(\langle\bar{\mathbf{w}}_1, \mathbf{x}\rangle\right) = x_1, \tag{61}$$

*for all* $\mathbf{x} \in \{\pm1\}^d$.

*Proof.* Recall that from eq. equation 44, with probability $1 - \delta$ we have:

$$\left| \left\langle \bar{\mathbf{w}}_1 - \ln\left(\frac{1+p}{1-p}\right) \mathbf{e}_1, \mathbf{x} \right\rangle \right| \leq \frac{2d}{\lambda_1} \sqrt{\frac{1 + \ln T_1}{\delta T_1}} + \frac{4 \ln\left(\frac{1+p}{1-p}\right)}{1 - p^2} \lambda_1,$$

$$\left| \frac{1}{1 + e^{-\langle \bar{\mathbf{w}}_1, \mathbf{x} \rangle}} - \frac{1}{1 + \left(\frac{1-p}{1+p}\right)^{x_1}} \right| < \frac{d}{2\lambda_1} \sqrt{\frac{1 + \ln T_1}{\delta T_1}} + \frac{\ln\left(\frac{1+p}{1-p}\right)}{1 - p^2} \lambda_1. \tag{62}$$

To instantiate the bounds, first observe that for any $0 < p < 3/4$, it holds: $\frac{\ln\left(\frac{1+p}{1-p}\right)}{1-p^2} < \frac{16 \ln 7}{7}$.

Furthermore, the first bound is always larger than the second one. We set $\lambda_1 = \sqrt{\frac{d\left(\frac{1+\ln T_1}{T_1}\right)^{1/2}}{\frac{32 \ln 7}{7}}}$ to

bound the two terms and solve the following inequality for $T_1$:

$$d^{1/2} \left(\frac{1 + \ln T_1}{\delta T_1}\right)^{1/4} \left(\frac{512 \ln 7}{7}\right)^{1/2} < \varepsilon$$

$$\frac{T_1}{1 + \ln T_1} > \frac{512^2}{49} (\ln 7)^2 \frac{d^2}{\delta \varepsilon^4}. \tag{63}$$

This proves the first part of the claim. For the decision term $\text{sgn}\left(\langle \bar{\mathbf{w}}_1, \mathbf{x}\rangle\right)$, we have:

$$\text{sgn}\left(\langle \bar{\mathbf{w}}_1, \mathbf{x}\rangle\right) = \text{sgn}\left(\left\langle \bar{\mathbf{w}}_1 - \ln\left(\frac{1+p}{1-p}\right) \mathbf{e}_1, \mathbf{x}\right\rangle + \left\langle \ln\left(\frac{1+p}{1-p}\right) \mathbf{e}_1, \mathbf{x}\right\rangle\right)$$

$$= \text{sgn}\left(\left\langle \bar{\mathbf{w}}_1 - \ln\left(\frac{1+p}{1-p}\right) \mathbf{e}_1, \mathbf{x}\right\rangle + x_1 \ln\left(\frac{1+p}{1-p}\right)\right). \tag{64}$$

If $\left| x_1 \ln\left(\frac{1+p}{1-p}\right) \right| > \left| \left\langle \bar{\mathbf{w}}_1 - \ln\left(\frac{1+p}{1-p}\right) \mathbf{e}_1, \mathbf{x}\right\rangle \right|$, then the following holds: $\text{sgn}\left(\langle \bar{\mathbf{w}}_1, \mathbf{x}\rangle\right) = \text{sgn}\left(x_1 \ln\left(\frac{1+p}{1-p}\right)\right) = x_1$. For this to happen, we additionally require having $\varepsilon < \ln\left(\frac{1+p}{1-p}\right)$. $\square$

### E.3.2 POSITION 2

**Position 2a** We consider a binary classification problem where $\mathcal{X}_{2a} = \{\pm 1\}^{d+1}, \mathcal{Y}_{2a} = \{\pm 1\}$ with a distribution $\mathcal{D}_{2a}(p)$ over $\mathcal{X}_{2a} \times \mathcal{Y}_{2a}$ such that: $x_1, \ldots, x_d \sim \text{Rad}(1/2)$ and $(x_{d+1}, y) = Z(x_1, -1) + (1 - Z)\left(\prod_{i=1}^d x_i, +1\right)$ where $Z \sim \text{Ber}(p), 0 < p < 3/4$. The $-1$ label should be interpreted as the empty string $\epsilon$, while the $+1$ output corresponds to the `<EOS>` symbol. We consider a non-homogeneous linear hypothesis class $\mathcal{H}_{2a} = \{\mathbf{x} \mapsto \langle \mathbf{w}, \phi_2(\mathbf{x})\rangle + b : \mathbf{w} \in \mathbb{R}^{2d+1}, b \in \mathbb{R}\}$, where the feature map $\phi_2 : \mathbf{x} \mapsto [x_1 \quad \cdots \quad x_{d+1} \quad x_1 x_{d+1} \quad x_2 x_{d+1} \quad \cdots \quad x_d x_{d+1}]^T \in \{\pm 1\}^{2d+1}$ augments the input with the second degree monomials that involve the inputs bits and the bit from the previous position. In this way, the linear model is capable of approximating the target via the $x_1 x_{d+1}$ feature which can help predict whether the generation is on the "long" or "short" path. Note that the dimension of the feature map – $O(d)$ – is still polynomial in the input dimension. The bias term is crucial for representing the best-in-class conditional probabilities. We consider learning with the $\ell_2$-regularized logistic loss:

$$l^{(2a)}(\boldsymbol{\theta} = \{\mathbf{w}, b\}, (\mathbf{x}, y)) = \ln\left(1 + e^{-y(\langle \mathbf{w}, \phi_2(\mathbf{x})\rangle + b)}\right) + \frac{\lambda_{2a}}{2}\left(\|\mathbf{w}\|^2 + b^2\right), \quad \lambda_{2a} > 0. \tag{65}$$

**Proposition 2.** *Consider running SGD (Algorithm 2) for minimizing* $L_{\mathcal{D}_{2a}(p), \lambda_{2a}}(\boldsymbol{\theta} = \{\mathbf{w}, b\}) = \mathbb{E}_{(\mathbf{x}, y) \sim \mathcal{D}_{2a}(p)}\left[\ln\left(1 + e^{-y(\langle \mathbf{w}, \phi_2(\mathbf{x})\rangle + b)}\right)\right] + \frac{\lambda_{2a}}{2}\left(\|\mathbf{w}\|^2 + b^2\right), \lambda_{2a} > 0.$ *Then, after $T_{2a}$ iterations and for any $\delta \in (0, 1)$, with probability at least $1 - \delta$ over the sampled $\{(\mathbf{x}_i, y_i)\}_{i=1}^{T_{2a}}$, it holds for all $\mathbf{x} \in \{\pm 1\}^{d+1}$ with $x_1 x_{d+1} = -1$:*

$$\frac{1}{1 + e^{\langle \bar{\mathbf{w}}_{2a}, \mathbf{x}\rangle + \bar{b}_{2a}}} < \frac{2(d+1)}{\lambda_{2a}} \sqrt{\frac{1 + \ln T_{2a}}{\delta T_{2a}}} + \sqrt{\frac{\lambda_{2a}}{1 - p}}, \tag{66}$$

*while for* $\mathbf{x} \in \{\pm 1\}^{d+1}$ *such that* $x_1 x_{d+1} = +1$, *it holds:*

$$\left| \frac{1}{1 + e^{-\langle \bar{\mathbf{w}}_{2a}, \mathbf{x} \rangle - \bar{b}_{2a}}} - \frac{1-p}{1+p} \right| < \frac{2(d+1)}{\lambda_{2a}} \sqrt{\frac{1 + \ln T_{2a}}{\delta T_{2a}}} + \frac{\lambda_{2a}(1+p) \left| \ln \left( \frac{2p}{1-p} \right) \right|}{8p(1-p)},$$

$$\left| \langle \bar{\mathbf{w}}_{2a} - \hat{a}_0 \mathbf{e}_{d+2}, \phi_2(\mathbf{x}) \rangle + \bar{b}_{2a} - \hat{b}_0 \right| < \frac{8(d+1)}{\lambda_{2a}} \sqrt{\frac{1 + \ln T_{2a}}{\delta T_{2a}}} + \frac{\lambda_{2a}(1+p) \left| \ln \left( \frac{2p}{1-p} \right) \right|}{2p(1-p)},$$

(67)

*where* $\bar{\mathbf{w}}_{2a}, \bar{b}_{2a}$ *is the output of SGD and* $\hat{a}_0, \hat{b}_0 \in \mathbb{R}$ *such that:* $\hat{a}_0 + \hat{b}_0 = \ln \left( \frac{1-p}{2p} \right)$.

*Proof.* We first show that learning $\mathcal{H}_{2a}$ with respect to the unregularized loss $\ln \left( 1 + e^{-y(\langle \mathbf{w}, \phi_2(\mathbf{x}) \rangle + b)} \right)$ corresponds to a Convex and Lipschitz learning problem. We will abuse notation and write $\boldsymbol{\theta} \in \mathcal{H}_{2a}$ for $\boldsymbol{\theta} = (\mathbf{w}, b)$, where $(\mathbf{w}, b) \in \mathcal{H}_{2a}$. The loss is convex with respect to $\boldsymbol{\theta}$. For the Lipschitz constant, we bound the gradient with respect to $\boldsymbol{\theta}$. For all $\mathbf{x} \in \{\pm 1\}^{d+1}$, $y \in \{\pm 1\}$, we have:

$$\nabla_{\mathbf{w}} l^{(2a)} (\boldsymbol{\theta}, (\mathbf{x}, y)) = -\frac{y \phi_2(\mathbf{x})}{1 + e^{y(\langle \mathbf{w}, \phi_2(\mathbf{x}) \rangle + b)}}, \tag{68}$$

and for the bias term:

$$\frac{\partial}{\partial b} l^{(2a)} (\boldsymbol{\theta}, (\mathbf{x}, y)) = -\frac{y}{1 + e^{y(\langle \mathbf{w}, \phi_2(\mathbf{x}) \rangle + b)}}. \tag{69}$$

As a result, we have for the Lipschitz constant:

$$\left\| \nabla_{\boldsymbol{\theta}} l^{(2a)} (\boldsymbol{\theta}, (\mathbf{x}, y)) \right\|^2 = \left\| -\frac{y \phi_2(\mathbf{x})}{1 + e^{y(\langle \mathbf{w}, \phi_2(\mathbf{x}) \rangle + b)}} \right\|^2 + \left( -\frac{y}{1 + e^{y(\langle \mathbf{w}, \phi_2(\mathbf{x}) \rangle + b)}} \right)^2 \leq (2d + 1) + 1 \tag{70}$$

which implies that the function is $\sqrt{2d + 2}$-Lipschitz with respect to $\boldsymbol{\theta}$. Therefore, applying Theorem 3, we have that SGD after $T_{2a}$ iterations returns a hypothesis $\bar{\boldsymbol{\theta}}_{2a}$ such that for any $\delta \in (0, 1)$ with probability at least $1 - \delta$ it holds:

$$L_{\mathcal{D}_{2a}(p), \lambda_{2a}}(\bar{\boldsymbol{\theta}}_{2a}) \leq L_{\mathcal{D}_{2a}(p), \lambda_{2a}}(\hat{\boldsymbol{\theta}}) + \frac{4(d+1)}{\lambda_{2a} \delta T_{2a}} (1 + \ln T_{2a}), \tag{71}$$

where $\hat{\boldsymbol{\theta}} = \arg\min_{\boldsymbol{\theta} \in \mathcal{H}_{2a}} L_{\mathcal{D}_{2a}(p), \lambda_{2a}}(\boldsymbol{\theta})$. From the strong convexity of $L_{\mathcal{D}_{2a}(p), \lambda_{2a}}$, this implies:

$$\left\| \bar{\boldsymbol{\theta}}_{2a} - \hat{\boldsymbol{\theta}} \right\|^2 \leq \frac{2}{\lambda_{2a}} \left( L_{\mathcal{D}_{2a}(p), \lambda_{2a}}(\bar{\boldsymbol{\theta}}_{2a}) - L_{\mathcal{D}_{2a}(p), \lambda_{2a}}(\hat{\boldsymbol{\theta}}) \right) \leq \frac{8(d+1)}{\lambda_{2a}^2 \delta T_{2a}} (1 + \ln T_{2a}). \tag{72}$$

We characterize now the loss minimizer $\hat{\boldsymbol{\theta}} = \left\{ \hat{\mathbf{w}}, \hat{b} \right\}$ by setting the gradient of $L_{\mathcal{D}_{2a}(p), \lambda_{2a}}$ to zero:

$$\begin{cases} \mathbb{E}_{(\mathbf{x}, y) \sim \mathcal{D}_{2a}(p)} \left[ \frac{-y \phi_2(\mathbf{x})}{1 + e^{y(\langle \mathbf{w}, \phi_2(\mathbf{x}) \rangle + b)}} \right] + \lambda_{2a} \hat{\mathbf{w}} = 0, \\ \mathbb{E}_{(\mathbf{x}, y) \sim \mathcal{D}_{2a}(p)} \left[ -\frac{y}{1 + e^{y(\langle \mathbf{w}, \phi_2(\mathbf{x}) \rangle + b)}} \right] + \lambda_{2a} \hat{b} = 0. \end{cases} \tag{73}$$

The objective $L_{\mathcal{D}_{2a}(p), \lambda_{2a}}(\boldsymbol{\theta})$ is strongly convex, so it admits a unique solution. We observe that this solution is of the form $\hat{\mathbf{w}} = \hat{a} \mathbf{e}_{d+2}$, $\hat{a} \in \mathbb{R}$, where $\mathbf{e}_{d+2} = [0, \ldots, 0, 1, 0, \ldots, 0]^T$ and $\hat{b} \in \mathbb{R}$. Indeed, the optimality conditions yield:

$$\begin{cases} \frac{1}{1 + e^{-(\hat{a} - \hat{b})}} + \frac{\lambda_{2a}}{1-p} \left( \hat{a} - \hat{b} \right) = 0, \\ \frac{2p}{1 + e^{-(\hat{a} + \hat{b})}} + \frac{p-1}{1 + e^{\hat{a} + \hat{b}}} + \lambda_{2a} \left( \hat{a} + \hat{b} \right) = 0. \end{cases} \tag{74}$$

The functions $g_1(u) = \frac{1}{1 + e^{-u}} + \lambda u$, $\lambda > 0$ and $g_2(u) = \frac{2p}{1 + e^{-u}} + \frac{p-1}{1 + e^u} + \lambda u$, $\lambda > 0$ are continuous, monotonically increasing and $\lim_{u \to -\infty} g_1(u) = \lim_{u \to -\infty} g_2(u) = -\infty$, $\lim_{u \to +\infty} g_1(u) = \lim_{u \to +\infty} g_2(u) = +\infty$, hence they have unique roots. This proves that, indeed, $\hat{\mathbf{w}} = \hat{a} \mathbf{e}_{d+2}$, where $\hat{a}, \hat{b}$ are such that $g_1(\hat{a} - \hat{b}) = 0$ and $g_2(\hat{a} + \hat{b}) = 0$. We will now bound the error in the predicted probabilities, depending on what region of the distribution the input lies. We consider the following cases:

- If $x_1 x_{d+1} = -1$, then we have:

$$
\begin{aligned}
p_{-1}\left(\bar{\boldsymbol{\theta}}_{2a}\Big| x_1 x_{d+1} = -1\right) &= \frac{1}{1 + e^{\langle \bar{\mathbf{w}}_{2a}, \phi_2(\mathbf{x})\rangle + \bar{b}_{2a}}} \\
&= \frac{1}{1 + e^{\langle \bar{\mathbf{w}}_{2a}, \phi_2(\mathbf{x})\rangle + \bar{b}_{2a}}} - \frac{1}{1 + e^{\langle \hat{\mathbf{w}}, \phi_2(\mathbf{x})\rangle + \hat{b}}} + \frac{1}{1 + e^{\langle \hat{\mathbf{w}}, \phi_2(\mathbf{x})\rangle + \hat{b}}} \\
&< \frac{\sqrt{2(d+1)}}{4}\left\|\bar{\boldsymbol{\theta}}_{2a} - \hat{\boldsymbol{\theta}}\right\| + \frac{1}{1 + e^{\langle \hat{\mathbf{w}}, \phi_2(\mathbf{x})\rangle + \hat{b}}} \\
&\leq \frac{2(d+1)}{\lambda_{2a}}\sqrt{\frac{1 + \ln T_{2a}}{\delta T_{2a}}} + \frac{1}{1 + e^{\langle \hat{\mathbf{w}}, \phi_2(\mathbf{x})\rangle + \hat{b}}},
\end{aligned}
\tag{75}
$$

where we used equation 72 and the Lipschitzness of the logistic function. For the bias error term, we have:

$$
\begin{aligned}
\frac{1}{1 + e^{\langle \hat{\mathbf{w}}, \phi_2(\mathbf{x})\rangle + \hat{b}}} &= \frac{1}{1 + e^{\hat{a} x_1 x_{d+1} + \hat{b}}} \\
&= \frac{1}{1 + e^{-(\hat{a} - \hat{b})}} \\
&= -\frac{\lambda_{2a}}{1 - p}\left(\hat{a} - \hat{b}\right) \qquad \text{(from equation 74)} \\
&< \sqrt{\frac{\lambda_{2a}}{1 - p}},
\end{aligned}
\tag{76}
$$

as the solution $\hat{a} - \hat{b}$ of the $g_1(u) = 0$ equation lies in $\left(-\sqrt{\frac{1-p}{\lambda_{2a}}}, 0\right)$.

- If $x_1 x_{d+1} = 1$, then denote by $\hat{a}_0 + \hat{b}_0$ the solution of the unregularized problem $g_2(\hat{a}_0 + \hat{b}_0) = 0$ for $\lambda = 0$ or, equivalently from equation 74, $\hat{a}_0 + \hat{b}_0 = \ln\left(\frac{1-p}{2p}\right)$. We bound the distance between the model-induced probabilities and the best in-class probabilities:

$$
\begin{aligned}
&\left|p_1\left(\bar{\boldsymbol{\theta}}_{2a}\Big| x_1 x_{d+1} = +1\right) - p_1\left(\hat{\boldsymbol{\theta}}_0 \Big| x_1 x_{d+1} = +1\right)\right| \\
&\leq \left|p_1\left(\bar{\boldsymbol{\theta}}_{2a}\Big| x_1 x_{d+1} = +1\right) - p_1\left(\hat{\boldsymbol{\theta}}\Big| x_1 x_{d+1} = +1\right)\right| \\
&\quad + \left|p_1\left(\hat{\boldsymbol{\theta}}\Big| x_1 x_{d+1} = +1\right) - p_1\left(\hat{\boldsymbol{\theta}}_0\Big| x_1 x_{d+1} = +1\right)\right| \\
&= \frac{\sqrt{2(d+1)}}{4}\left\|\bar{\boldsymbol{\theta}}_{2a} - \hat{\boldsymbol{\theta}}\right\| + \frac{1}{4}\left|\left(\hat{a} + \hat{b}\right) - \left(\hat{a}_0 + \hat{b}_0\right)\right|.
\end{aligned}
\tag{77}
$$

It remains to show that the distance $\left|\left(\hat{a} + \hat{b}\right) - \left(\hat{a}_0 + \hat{b}_0\right)\right|$ is bounded. Let us denote the unregularized part of $g_2$ as $\bar{g}$, that is $\bar{g}(u) = \frac{2p}{1 + e^{-u}} + \frac{p-1}{1 + e^u}$. First, observe that $g_2(0) = \frac{3p-1}{2}$, which means that the sign of $\hat{a} + \hat{b}$ depends on the value of $p$. It will be easier to treat these three subcases separately:

  - If $p < 1/3$, then $g_2(0) < 0$ and $g_2(\hat{a}_0 + \hat{b}_0) = \lambda\left(\hat{a}_0 + \hat{b}_0\right) > 0$, so it holds $0 < \hat{a} + \hat{b} < \hat{a}_0 + \hat{b}_0$. From the mean value theorem, there exists $\xi \in \left(\hat{a} + \hat{b}, \hat{a}_0 + \hat{b}_0\right)$ such that:

$$
\bar{g}'(\xi) = \frac{\lambda_{2a}\left(\hat{a} + \hat{b}\right)}{\left(\hat{a}_0 + \hat{b}_0\right) - \left(\hat{a} + \hat{b}\right)}.
\tag{78}
$$

But, $\bar{g}'(u) = \frac{(p+1)e^u}{(1+e^u)^2}$ and, in particular, $\bar{g}'\left(\hat{a}_0 + \hat{b}_0\right) = \bar{g}'\left(\ln\left(\frac{1-p}{2p}\right)\right) = \frac{2p(1-p)}{1+p}$. This implies that for any $u \in \left(0, \ln\left(\frac{1-p}{2p}\right)\right)$, it holds: $\frac{2p(1-p)}{1+p} \le \bar{g}'(u) \le \frac{1+p}{4}$. Thus,

$$
\begin{aligned}
\left(\hat{a}_0 + \hat{b}_0\right) - \left(\hat{a} + \hat{b}\right) &= \frac{\lambda_{2a}\left(\hat{a} + \hat{b}\right)}{\bar{g}'(\xi)} \\
&< \frac{\lambda_{2a}(1+p)\left(\hat{a}_0 + \hat{b}_0\right)}{2p(1-p)} \\
&= \frac{\lambda_{2a}(1+p)\ln\left(\frac{1-p}{2p}\right)}{2p(1-p)}.
\end{aligned}
\tag{79}
$$

- If $p = 1/3$, then $\hat{a} + \hat{b} = 0$ as $g_2(0) = 0$ and, also, $\hat{a}_0 + \hat{b}_0 = \ln\left(\frac{1-p}{2p}\right) = 0$. Hence, the bias error term is 0.
- If $p > 1/3$, then $g_2(0) > 0$, $g_2(\hat{a}_0 + \hat{b}_0) = \lambda_{2a}\left(\hat{a}_0 + \hat{b}_0\right) < 0$, so $\hat{a}_0 + \hat{b}_0 < \hat{a} + \hat{b} < 0$.

  From the mean value theorem, there exists $\xi \in \left(\hat{a}_0 + \hat{b}_0, \hat{a} + \hat{b}\right)$ such that:

$$
\bar{g}'(\xi) = -\frac{\lambda_{2a}\left(\hat{a} + \hat{b}\right)}{\left(\hat{a} + \hat{b}\right) - \left(\hat{a}_0 + \hat{b}_0\right)} < -\frac{\lambda_{2a}\left(\hat{a}_0 + \hat{b}_0\right)}{\left(\hat{a} + \hat{b}\right) - \left(\hat{a}_0 + \hat{b}_0\right)},
\tag{80}
$$

  which, further implies:

$$
\left|\left(\hat{a} + \hat{b}\right) - \left(\hat{a}_0 + \hat{b}_0\right)\right| < \frac{\lambda_{2a}(1+p)\left(\ln\left(\frac{2p}{1-p}\right)\right)}{2p(1-p)},
\tag{81}
$$

  from the same bound on $\bar{g}'$ as before.

Therefore, we showed that for any $0 < p < 3/4$, we have:

$$
\left|\left(\hat{a} + \hat{b}\right) - \left(\hat{a}_0 + \hat{b}_0\right)\right| < \frac{\lambda_{2a}(1+p)\left|\ln\left(\frac{2p}{1-p}\right)\right|}{2p(1-p)}.
\tag{82}
$$

Combining this guarantee with the bound of equation 77 and the parameter space guarantee of equation 72, we finally have:

$$
\left|p_1\left(\bar{\boldsymbol{\theta}}_{2a}\middle|x_1 x_{d+1} = +1\right) - \frac{1-p}{1+p}\right| < \frac{2(d+1)}{\lambda_{2a}}\sqrt{\frac{1 + \ln T_{2a}}{\delta T_{2a}}} + \frac{\lambda_{2a}(1+p)\left|\ln\left(\frac{2p}{1-p}\right)\right|}{8p(1-p)}.
\tag{83}
$$

Furthermore, for the optimality of $\bar{\boldsymbol{\theta}}_{2a}$ in parameter space, we have:

$$
\begin{aligned}
&\left|\langle \bar{\mathbf{w}}_{2a} - \hat{a}_0 \mathbf{e}_{d+2}, \phi_2(\mathbf{x})\rangle + \bar{b}_{2a} - \hat{b}_0\right| \\
&= \left|\langle \bar{\mathbf{w}}_{2a} - \hat{a} \mathbf{e}_{d+2}, \phi_2(\mathbf{x})\rangle + \bar{b} - \hat{b} - \left(\langle \hat{a}_0 \mathbf{e}_{d+2} - \hat{a} \mathbf{e}_{d+2}, \phi_2(\mathbf{x})\rangle + \hat{b}_0 - \hat{b}\right)\right| \\
&\le \sqrt{2(d+1)}\|\bar{\boldsymbol{\theta}}_{2a} - \hat{\boldsymbol{\theta}}\| + \left|\left(\hat{a}_0 + \hat{b}_0\right) - \left(\hat{a} + \hat{b}\right)\right| \\
&< \frac{8(d+1)}{\lambda_{2a}}\sqrt{\frac{1 + \ln T_{2a}}{\delta T_{2a}}} + \frac{\lambda_{2a}(1+p)\left|\ln\left(\frac{2p}{1-p}\right)\right|}{2p(1-p)}.
\end{aligned}
\tag{84}
$$

where the last inequality follows from eqs. 82, 72. $\qquad\square$

As before, we obtain the following explicit corollary on the calibration and "hard" prediction of the second linear model.

**Corollary 2.** *Consider running SGD (Algorithm 2) for minimizing* $L_{\mathcal{D}_{2a}(p),\lambda_{2a}}(\mathbf{w}) = \mathbb{E}_{(\mathbf{x},y)\sim\mathcal{D}_{2a}(p)}\left[\ln\left(1 + e^{-y(\langle\mathbf{w},\phi_2(\mathbf{x})\rangle+b)}\right)\right] + \frac{\lambda_{2a}}{2}\left(\|\mathbf{w}\|^2 + b^2\right)$. *Then, for any* $\varepsilon > 0$, *after* $T_{2a} = \tilde{O}\left(\max\left\{\frac{d^2}{\delta\varepsilon^6}, \frac{d^2}{p^2\delta\varepsilon^4}\right\}\right)$ *iterations with* $\lambda_{2a} = \tilde{\Theta}\left(\min\left\{\frac{d^{\frac{1}{2}}p^{\frac{1}{2}}}{\delta^{1/4}T_{2a}^{1/4}}, \frac{d^{2/3}}{\delta^{1/3}T_{2a}^{1/3}}\right\}\right)$, *with probability at least* $1 - \delta$ *over the sampled* $\{(\mathbf{x}_i, y_i)\}_{i=1}^{T_{2a}} \sim \mathcal{D}_{2a}(p)$, *it holds:*

$$\left|\frac{1}{1 + e^{-\langle\bar{\mathbf{w}}_{2a},\mathbf{x}\rangle-\bar{b}_{2a}}} - \frac{1-p}{1+p}\right| < \varepsilon, \forall\, \mathbf{x} \in \{\pm1\}^{d+1} \text{ s.t. } x_1x_{d+1} = +1,$$

$$\frac{1}{1 + e^{\langle\bar{\mathbf{w}}_{2a},\mathbf{x}\rangle+\bar{b}_{2a}}} < \varepsilon, \forall\, \mathbf{x} \in \{\pm1\}^{d+1} \text{ s.t. } x_1x_{d+1} = -1. \tag{85}$$

*If additionally* $\varepsilon < \left|\ln\left(\frac{1-p}{2p}\right)\right|$, *then with probability at least* $1 - \delta$ *we have:*

$$\text{sgn}\left(\langle\bar{\mathbf{w}}_{2a}, \phi_2(\mathbf{x})\rangle + \bar{b}_{2a}\right) = \text{sgn}\left(\ln\left(\frac{1-p}{2p}\right)\right) = \begin{cases} -1, & \text{if } p \geq 1/3, \\ +1, & \text{if } p < 1/3, \end{cases} \tag{86}$$

*for all* $\mathbf{x} \in \{\pm1\}^{d+1}$ *with* $x_1x_{d+1} = 1$.

*Proof.* Recall that from eqs. 66, 67, we have: for all $\mathbf{x} \in \{\pm1\}^{d+1}$ with $x_1x_{d+1} = -1$:

$$\frac{1}{1 + e^{\langle\bar{\mathbf{w}}_{2a},\mathbf{x}\rangle+\bar{b}_{2a}}} < \frac{2(d+1)}{\lambda_{2a}}\sqrt{\frac{1 + \ln T_{2a}}{\delta T_{2a}}} + \sqrt{\frac{\lambda_{2a}}{1-p}}$$

$$< \frac{2(d+1)}{\lambda_{2a}}\sqrt{\frac{1 + \ln T_{2a}}{\delta T_{2a}}} + \sqrt{2\lambda_{2a}} := Q_a(\lambda_{2a}, T_{2a}), \tag{87}$$

while for $\mathbf{x} \in \{\pm1\}^{d+1}$ such that $x_1x_{d+1} = +1$, it holds:

$$\left|\frac{1}{1 + e^{-\langle\bar{\mathbf{w}}_{2a},\mathbf{x}\rangle-\bar{b}_{2a}}} - \frac{1-p}{1+p}\right| < \frac{2(d+1)}{\lambda_{2a}}\sqrt{\frac{1 + \ln T_{2a}}{\delta T_{2a}}} + \frac{\lambda_{2a}(1+p)\left|\ln\left(\frac{2p}{1-p}\right)\right|}{8p(1-p)},$$

$$\left|\langle\bar{\mathbf{w}}_{2a} - \hat{a}_0\mathbf{e}_{d+2}, \phi_2(\mathbf{x})\rangle + \bar{b}_{2a} - \hat{b}_0\right| < \underbrace{\frac{8(d+1)}{\lambda_{2a}}\sqrt{\frac{1 + \ln T_{2a}}{\delta T_{2a}}} + \frac{\lambda_{2a}(1+p)\left|\ln\left(\frac{2p}{1-p}\right)\right|}{2p(1-p)}}_{Q_b(\lambda_{2a}, T_{2a})}. \tag{88}$$

We want to upper bound all three quantities by $\varepsilon \in (0,1)$. In equation 88, the second inequality upper bounds the first one. Thus, it suffices to only consider that one. Let $\lambda_a, \lambda_b$ be the optimal $\lambda_{2a}$'s for the two previous expressions (i.e., quantities $Q_a(\lambda_{2a}, T_{2a}), Q_b(\lambda_{2a}, T_{2a})$) and $T_a, T_b$ the corresponding number of iterations to get the expressions less than $\varepsilon$. We have: $\lambda_a = \tilde{\Theta}\left(\frac{d^{2/3}}{\delta^{1/3}T_a^{1/3}}\right)$, while $\lambda_b = \tilde{\Theta}\left(\frac{p^{\frac{1}{2}}d^{\frac{1}{2}}}{\delta^{1/4}T_b^{1/4}}\right)$ (by balancing the two terms of each of $Q_a, Q_b$). For these values of $\lambda_a, \lambda_b$, the two expressions are equal to:

$$Q_a(\lambda_a, T_a) = \tilde{\Theta}\left(\frac{d^{1/3}}{\delta^{1/6}T_a^{1/6}}\right),$$

$$Q_b(\lambda_b, T_b) = \tilde{\Theta}\left(\frac{d^{1/2}}{p^{\frac{1}{2}}\delta^{1/4}T_b^{1/4}}\right). \tag{89}$$

This implies that it is sufficient to take $T_a = \tilde{O}\left(\frac{d^2}{\delta\varepsilon^6}\right)$ and $T_b = \tilde{O}\left(\frac{d^2}{p^2\delta\varepsilon^4}\right)$ to satisfy $Q_a(\lambda_a, T_a) < \varepsilon$ and $Q_b(\lambda_b, T_b) < \varepsilon$, respectively. In other words, it is sufficient to take $\lambda_{2a} = \tilde{\Theta}\left(\min\left\{\frac{d^{\frac{1}{2}}p^{\frac{1}{2}}}{\delta^{1/4}T_{2a}^{1/4}}, \frac{d^{2/3}}{\delta^{1/3}T_{2a}^{1/3}}\right\}\right)$ and $T_{2a} = \tilde{O}\left(\max\left\{\frac{d^2}{\delta\varepsilon^6}, \frac{d^2}{p^2\delta\varepsilon^4}\right\}\right)$. This proves the first part of the claim. For the decision term $\text{sgn}\left(\langle\bar{\mathbf{w}}_{2a}, \phi_2(\mathbf{x})\rangle + b_{2a}\right)$, we have:

$$\text{sgn}\left(\langle\bar{\mathbf{w}}_{2a}, \phi_2(\mathbf{x})\rangle + b_{2a}\right) = \text{sgn}\left(\langle\bar{\mathbf{w}}_{2a} - \hat{a}_0\mathbf{e}_{d+2}, \phi_2(\mathbf{x})\rangle + \bar{b}_{2a} - \hat{b}_0 + \langle\hat{a}_0\mathbf{e}_{d+2}, \phi_2(\mathbf{x})\rangle + \hat{b}_0\right)$$

$$= \text{sgn}\left(\langle\bar{\mathbf{w}}_{2a} - \hat{a}_0\mathbf{e}_{d+2}, \phi_2(\mathbf{x})\rangle + \bar{b}_{2a} - \hat{b}_0 + \hat{a}_0x_1x_{d+1} + \hat{b}_0\right). \tag{90}$$

In particular, when $x_1 x_{d+1} = 1$, we have:

$$\operatorname{sgn}\left(\langle \bar{\mathbf{w}}_{2a}, \phi_2(\mathbf{x})\rangle + b_{2a}\right) = \operatorname{sgn}\left(\langle \bar{\mathbf{w}}_{2a} - \hat{a}_0 \mathbf{e}_{d+2}, \phi_2(\mathbf{x})\rangle + \bar{b}_{2a} - \hat{b}_0 + \hat{a}_0 + \hat{b}_0\right)$$

$$= \operatorname{sgn}\left(\langle \bar{\mathbf{w}}_{2a} - \hat{a}_0 \mathbf{e}_{d+2}, \phi_2(\mathbf{x})\rangle + \bar{b}_{2a} - \hat{b}_0 + \ln\left(\frac{1-p}{2p}\right)\right). \tag{91}$$

If $\left|\ln\left(\frac{1-p}{2p}\right)\right| > \left|\langle \bar{\mathbf{w}}_{2a} - \hat{a}_0 \mathbf{e}_{d+2}, \phi_2(\mathbf{x})\rangle + \bar{b}_{2a} - \hat{b}_0\right|$, then the following holds: $\operatorname{sgn}\left(\langle \bar{\mathbf{w}}_{2a}, \phi_2(\mathbf{x})\rangle + b_{2a}\right) = \operatorname{sgn}\left(\ln\left(\frac{1-p}{2p}\right)\right) = \operatorname{sgn}\left(1/3 - p\right)$. For this to happen, it suffices to additionally have $\varepsilon < \left|\ln\left(\frac{1-p}{2p}\right)\right|$.

$\square$

**Position 2b** For the second part of position 2, we consider a binary classification problem where all the data come from the "long" path. As this is similar to positions 3 to d, we treat all of these positions together next. We will denote the second part of position 2 simply as position 2 in the next subsubsection for ease of presentation.

### E.3.3   Positions 2 to d

For the $l$'th position of the output, $2 \leq l \leq d$, we consider a binary classification problem where $\mathcal{X}_l = \{\pm 1\}^{d+l-1}, \mathcal{Y}_l = \{\pm 1\}$ with a distribution $\mathcal{D}_l(p)$ over $\mathcal{X}_l \times \mathcal{Y}_l$ such that: $x_1, \ldots, x_d \sim \operatorname{Rad}(1/2)$, $(x_{d+1}, \ldots, x_{d+l-1}) = \left(x_1, \ldots, \prod_{i=1}^{l-1} x_i\right)$ and $y = \prod_{i=1}^{l} x_i$. We consider a linear hypothesis class $\mathcal{H}_l = \left\{\mathbf{x} \mapsto \langle \mathbf{w}, \phi_l(\mathbf{x})\rangle : \mathbf{w} \in \mathbb{R}^{2d+l-1}\right\}$, where the feature map $\phi_l : \mathbf{x} \mapsto \begin{bmatrix} x_1 & \cdots & x_{d+l-1} & x_{d+l-1}x_1 & \cdots x_{d+l-1}x_d \end{bmatrix}^T \in \{\pm 1\}^{2d+l-1}$ augments the input with the second degree monomials that involve the inputs bits and the bit from the previous ($l-1$'th) position. We consider learning with the logistic loss plus an additional $\ell_2$ regularization term: $l^{(l)}(\mathbf{w}, (\mathbf{x}, y)) = \ln\left(1 + e^{-y\langle \mathbf{w}, \phi_l(\mathbf{x})\rangle}\right) + \frac{\lambda_l}{2}\|\mathbf{w}\|^2, \lambda_l > 0$.

**Proposition 3.** *For any $l \in \{2, \ldots, d\}$, consider running SGD (Algorithm 2) for minimizing $L_{\mathcal{D}_l(p), \lambda_l}(\mathbf{w}) = \mathbb{E}_{(\mathbf{x},y) \sim \mathcal{D}_l(p)} \left[\ln\left(1 + e^{-y\langle \mathbf{w}, \phi_l(\mathbf{x})\rangle}\right)\right] + \frac{\lambda_l}{2}\|\mathbf{w}\|^2$ with $\lambda_l > 0$. Then, after $T_l$ iterations and for $\delta \in (0,1)$, with probability at least $1 - \delta$ over the sampled $\{(\mathbf{x}_i, y_i)\}_{i=1}^{T_l}$, it holds:*

$$\frac{1}{1 + e^{-\langle \bar{\mathbf{w}}_l, \phi_l(\mathbf{x})\rangle}} < \frac{2d + l - 1}{2\lambda_l}\sqrt{\frac{1 + \ln T_l}{\delta T_l}} + \sqrt{\lambda_l}, \tag{92}$$

*for all $\mathbf{x} \in \{\pm 1\}^{d+l-1}$ such that $x_{d+l-1} = \prod_{i=1}^{l-1} x_i \wedge \prod_{i=1}^{l} x_i = -1$, and*

$$\frac{1}{1 + e^{\langle \bar{\mathbf{w}}_l, \phi_l(\mathbf{x})\rangle}} < \frac{2d + l - 1}{2\lambda_l}\sqrt{\frac{1 + \ln T_l}{\delta T_l}} + \sqrt{\lambda_l}, \tag{93}$$

*for all $\mathbf{x} \in \{\pm 1\}^{d+l-1}$ such that $x_{d+l-1} = \prod_{i=1}^{l-1} x_i \wedge \prod_{i=1}^{l} x_i = 1$, where $\bar{\mathbf{w}}_l$ is the output of SGD.*

*Proof.* We first show that learning $\mathcal{H}_l$ with respect to $l^{(l)}(\mathbf{w}, (\mathbf{x}, y)) = \ln\left(1 + e^{-y\langle \mathbf{w}, \phi_l(\mathbf{x})\rangle}\right) + \frac{\lambda_l}{2}\|\mathbf{w}\|^2$ corresponds to a Convex and Lipschitz learning problem. The loss is $\lambda_l$-strongly convex with respect to its first argument. For the Lipschitz constant, we have for all $\mathbf{x}, y$ and $\mathbf{w} \in \mathbb{R}^{2d+l-1}$:

$$\left\|\nabla_{\mathbf{w}} \ln\left(1 + e^{-y\langle \mathbf{w}, \phi_l(\mathbf{x})\rangle}\right)\right\| = \left\|-\frac{y\phi_l(\mathbf{x})}{1 + e^{y\langle \mathbf{w}, \phi_l(\mathbf{x})\rangle}}\right\| \leq \sqrt{2d + l - 1}. \tag{94}$$

Therefore, applying Theorem 3, we have that SGD after $T_l$ iterations returns a hypothesis $\bar{\mathbf{w}}_l$ such that for any $\delta \in (0,1)$ with probability at least $1 - \delta$ it holds:

$$L_{\mathcal{D}_l(p), \lambda_l}(\bar{\mathbf{w}}_l) \leq L_{\mathcal{D}_l(p), \lambda_l}(\hat{\mathbf{w}}_l) + \frac{2(2d + l - 1)}{\lambda_l \delta T_l}(1 + \ln T_l), \tag{95}$$

where $\hat{\mathbf{w}}_l = \arg\min_{\mathbf{w} \in \mathcal{H}_l} L_{\mathcal{D}_l(p), \lambda_l}(\mathbf{w})$. From the strong convexity of $L_{\mathcal{D}_l(p), \lambda_l}$, this implies:

$$\|\bar{\mathbf{w}}_l - \hat{\mathbf{w}}_l\|^2 \leq \frac{2}{\lambda_l}\left(L_{\mathcal{D}_l(p), \lambda_l}(\bar{\mathbf{w}}_l) - L_{\mathcal{D}_l(p), \lambda_l}(\hat{\mathbf{w}}_l)\right) \leq \frac{4(2d + l - 1)}{\lambda_l^2 \delta T_l}(1 + \ln T_l). \tag{96}$$

The previous bound on the parameter space can be translated to a guarantee on the calibration of the model. First, we find the optimal solution $\hat{\mathbf{w}}_l = \arg\min_{\mathbf{w} \in \mathcal{H}_l} L_{\mathcal{D}_l(p), \lambda_l}(\mathbf{w})$. We set the gradient of $L_{\mathcal{D}_l(p), \lambda_l}(\mathbf{w})$ to zero:

$$\mathbb{E}_{(\mathbf{x}, y) \sim \mathcal{D}_l(p)} \left[ \frac{-y \phi_l(\mathbf{x})}{1 + e^{y \langle \hat{\mathbf{w}}_l, \phi_l(\mathbf{x}) \rangle}} \right] + \lambda_l \hat{\mathbf{w}}_l = 0. \tag{97}$$

The objective $L_{\mathcal{D}_l(p), \lambda_l}(\mathbf{w})$ is strongly convex, so it admits a unique solution. We observe that this solution is of the form $\hat{\mathbf{w}}_l = \alpha \mathbf{e}_{d+2l-1}$, $\alpha \in \mathbb{R}$, where $\mathbf{e}_{d+2l-1} = [0, \ldots, 0, 1, 0, \ldots, 0]^T$ [10]. Indeed, the optimality conditions become:

$$\begin{cases} \mathbb{E}_{x_1, \ldots, x_d \sim \text{Rad}(1/2)} \left[ \frac{-\prod_{i=1}^{l} x_i \prod_{i=1}^{l} x_i}{1 + e^{\alpha \prod_{i=1}^{l} x_i \prod_{i=1}^{l} x_i}} \right] + \lambda_l \alpha = 0, \\ \mathbb{E}_{x_1, \ldots, x_d \sim \text{Rad}(1/2)} \left[ \frac{-\phi_l(\mathbf{x})_j \prod_{i=1}^{l} x_i}{1 + e^{\alpha \prod_{i=1}^{l} x_i \prod_{i=1}^{l} x_i}} \right] = 0, \ j \neq d + 2l - 1. \end{cases} \tag{98}$$

All but the $d + 2l - 1$'th equation are satisfied as $\mathbb{E}_{x_1, \ldots, x_d \sim \text{Rad}(1/2)} \left[ \prod_{i=1}^{l} x_i \phi_l(\mathbf{x})_j \right] = 0$ (there is always at least one zero-mean bit that survives in the product). The $d + 2l - 1$'th equation can be simplified to:

$$g(\alpha) := \frac{1}{1 + e^{\alpha}} - \lambda_l \alpha = 0. \tag{99}$$

This equation has, indeed, a unique root as $g$ is continuous and $g'(u) = -\frac{e^{-\alpha}}{(1 + e^{-\alpha})^2} - \lambda_l < 0$ for all $\lambda_l > 0, u \in \mathbb{R}$. Furthermore, $g(0) = 1 > 0$, while $g\left(\sqrt{\frac{1}{\lambda_l}}\right) = \frac{1}{1 + e^{\sqrt{\frac{1}{\lambda_l}}}} - \sqrt{\lambda_l} < 0$, as for any $\lambda_l > 0$ it holds: $1 + e^{\sqrt{\frac{1}{\lambda_l}}} \geq \sqrt{\frac{1}{\lambda_l}} + 2 > \sqrt{\frac{1}{\lambda_l}}$. From the intermediate value theorem, we get that $\alpha \in \left(0, \sqrt{\frac{1}{\lambda}}\right)$. Hence, for the output probabilities of the optimal model $\hat{\mathbf{w}}_l$, we have:

$$p_1(\hat{\mathbf{w}}_l \mid \mathbf{x}) = \sigma(\langle \hat{\mathbf{w}}_l, \phi_l(\mathbf{x}) \rangle) = \frac{1}{1 + e^{-\langle \hat{\mathbf{w}}_l, \phi_l(\mathbf{x}) \rangle}} = \frac{1}{1 + e^{-\alpha \prod_{i=1}^{l} x_i}},$$
$$p_{-1}(\hat{\mathbf{w}}_l \mid \mathbf{x}) = \sigma(\langle \hat{\mathbf{w}}_l, \phi_l(\mathbf{x}) \rangle) = \frac{1}{1 + e^{\langle \hat{\mathbf{w}}_l, \phi_l(\mathbf{x}) \rangle}} = \frac{1}{1 + e^{\alpha \prod_{i=1}^{l} x_i}}. \tag{100}$$

For any $\mathbf{x} \in \{\pm 1\}^{d+l-1}$ with $x_{d+l-1} = \prod_{i=1}^{l-1} x_i$, the above together with eq. equation 99 imply:

$$p_1\left(\hat{\mathbf{w}}_l \middle| \prod_{i=1}^{l} x_i = -1\right) = \frac{1}{1 + e^{\alpha}} = \lambda_l \alpha < \sqrt{\lambda_l},$$
$$p_{-1}\left(\hat{\mathbf{w}}_l \middle| \prod_{i=1}^{l} x_i = 1\right) = \frac{1}{1 + e^{\alpha}} = \lambda_l \alpha < \sqrt{\lambda_l} \tag{101}$$

Finally, combining with eq. equation 96, we have:

$$\begin{aligned} p_1\left(\bar{\mathbf{w}}_l \middle| \prod_{i=1}^{l} x_i = -1\right) &= \frac{1}{1 + e^{\langle \bar{\mathbf{w}}_l, \phi_l(\mathbf{x}) \rangle}} \\ &= \frac{1}{1 + e^{\langle \bar{\mathbf{w}}_l, \phi_l(\mathbf{x}) \rangle}} - \frac{1}{1 + e^{\langle \hat{\mathbf{w}}_l, \phi_l(\mathbf{x}) \rangle}} + \frac{1}{1 + e^{\langle \hat{\mathbf{w}}_l, \phi_l(\mathbf{x}) \rangle}} \\ &< \frac{\sqrt{2d + l - 1}}{4} \|\bar{\mathbf{w}}_l - \hat{\mathbf{w}}_l\| + \sqrt{\lambda_l} \\ &\leq \frac{2d + l - 1}{2\lambda_l} \sqrt{\frac{1 + \ln T_l}{\delta T_l}} + \sqrt{\lambda_l}, \end{aligned} \tag{102}$$

---

[10] One way to "guess" the optimal solution is the following. Suppose the solution had an additional non-zero coefficient: $\hat{\mathbf{w}}_l = \alpha \mathbf{e}_{d+2l-1} + \beta \mathbf{e}_j$, $j \neq d + 2l - 1$. Take, for concreteness, $j = 1$ (the argument is invariant to the choice of $j$). Then, the optimality conditions yield: $\mathbb{E}_{\mathbf{x}} \left[ \frac{-\prod_{i=1}^{l} x_i \prod_{i=1}^{l} x_i}{1 + e^{\prod_{i=1}^{l} x_i \left( \alpha \prod_{i=1}^{l} x_i + \beta x_1 \right)}} \right] + \lambda_l \alpha = 0$, and $\mathbb{E}_{\mathbf{x}} \left[ \frac{-x_1 \prod_{i=1}^{l} x_i}{1 + e^{\prod_{i=1}^{l} x_i \left( \alpha \prod_{i=1}^{l} x_i + \beta x_1 \right)}} \right] + \lambda_l \beta = 0$. By summing up and subtracting the equations, we get: $\frac{1}{1 + e^{\alpha + \beta}} - \lambda_l(\alpha + \beta) = 0$ and $\frac{1}{1 + e^{\alpha - \beta}} - \lambda_l(\alpha - \beta) = 0$. However, since the equation $\frac{1}{1 + e^u} - \lambda_l u = 0$ has a unique root for any $\lambda_l > 0$, it must be: $\alpha + \beta = \alpha - \beta$, or equivalently, $\beta = 0$.

and, similarly:

$$p_{-1}\left(\bar{\mathbf{w}}_l \middle| \prod_{i=1}^{l} x_i = 1\right) < \frac{2d+l-1}{2\lambda_l}\sqrt{\frac{1+\ln T_l}{\delta T_l}} + \sqrt{\lambda_l}. \tag{103}$$

$\square$

We obtain the following corollary on the calibration and "hard" prediction of the hypothesis returned by SGD for the $l$'th position of the output.

**Corollary 3.** *Consider running SGD (Algorithm 2) for minimizing $L_{\mathcal{D}_l(p),\lambda_l}(\mathbf{w}) = \mathbb{E}_{(\mathbf{x},y)\sim\mathcal{D}_l(p)}\left[\ln\left(1+e^{-y\langle\mathbf{w},\mathbf{x}\rangle}\right)\right] + \frac{\lambda_l}{2}\|\mathbf{w}\|^2$. Then, for any $\varepsilon > 0$, if $\lambda_l = \tilde{\Theta}\left(\frac{d^{2/3}}{\delta^{1/3}T_l^{1/3}}\right)$, and after $T_l = \tilde{O}\left(\frac{d^2}{\delta\varepsilon^6}\right)$ iterations, with probability at least $1-\delta$ over the sampled $\{(\mathbf{x}_i,y_i)\}_{i=1}^{T_l} \sim \mathcal{D}_l(p)$, we have:*

$$\frac{1}{1+e^{-\langle\bar{\mathbf{w}}_l,\mathbf{x}\rangle}} < \varepsilon, \forall\,\mathbf{x}\in\{\pm 1\}^{d+l-1} \text{ s.t. } x_{d+l-1} = \prod_{i=1}^{l-1}x_i \wedge \prod_{i=1}^{l}x_i = -1,$$
$$\frac{1}{1+e^{\langle\bar{\mathbf{w}}_l,\mathbf{x}\rangle}} < \varepsilon, \forall\,\mathbf{x}\in\{\pm 1\}^{d+l-1} \text{ s.t. } x_{d+l-1} = \prod_{i=1}^{l-1}x_i \wedge \prod_{i=1}^{l}x_i = 1. \tag{104}$$

*If additionally, $\varepsilon < 1/2$, then with probability at least $1-\delta$, we have:*

$$\text{sgn}\left(\langle\bar{\mathbf{w}}_l,\phi_l(\mathbf{x})\rangle\right) = \prod_{i=1}^{l}x_i, \tag{105}$$

*for all $\mathbf{x}\in\{\pm 1\}^{d+l-1}$ with $x_{d+l-1} = \prod_{i=1}^{l-1}x_i$.*

*Proof.* Recall that from equation 92, we have:

$$\frac{1}{1+e^{-\langle\bar{\mathbf{w}}_l,\mathbf{x}\rangle}} < \frac{2d+l-1}{2\lambda_l}\sqrt{\frac{1+\ln T_l}{\delta T_l}} + \sqrt{\lambda_l}, \forall\mathbf{x}\in\{\pm 1\}^d \text{ s.t. } \prod_{i=1}^{l}x_i = -1,$$
$$\frac{1}{1+e^{\langle\bar{\mathbf{w}}_l,\mathbf{x}\rangle}} < \frac{2d+l-1}{2\lambda_l}\sqrt{\frac{1+\ln T_l}{\delta T_l}} + \sqrt{\lambda_l}, \forall\mathbf{x}\in\{\pm 1\}^d \text{ s.t. } \prod_{i=1}^{l}x_i = 1, \tag{106}$$

Let $\lambda_l = \left(\frac{2d+l-1}{2}\right)^{2/3}\left(\frac{1+\ln T_l}{T_l}\right)^{1/3}$, then the upper bound becomes:

$$\frac{2d+l-1}{2\lambda_l}\sqrt{\frac{1+\ln T_l}{\delta T_l}} + \sqrt{\lambda_l} = 2\left(\frac{2d+l-1}{2}\right)^{1/3}\left(\frac{1+\ln T_l}{\delta T_l}\right)^{1/6}. \tag{107}$$

The first claim follows by solving the following inequality for $T_l$:

$$2\left(\frac{2d+l-1}{2}\right)^{1/3}\left(\frac{1+\ln T_l}{\delta T_l}\right)^{1/6} < \varepsilon. \tag{108}$$

For the second claim, observe that by a standard property of the logistic function, we have:

$$\text{sgn}\left(\langle\bar{\mathbf{w}}_l,\phi_l(\mathbf{x})\rangle\right) = \text{sgn}\left(\frac{1}{1+e^{-\langle\bar{\mathbf{w}}_l,\phi_l(\mathbf{x})\rangle}} - \frac{1}{2}\right). \tag{109}$$

Assume that $\varepsilon < 1/2$, then we consider the cases:

- If $\prod_{i=1}^{l}x_i = -1$, then we have:

$$\frac{1}{1+e^{-\langle\bar{\mathbf{w}}_l,\phi_l(\mathbf{x})\rangle}} < \varepsilon < 1/2, \tag{110}$$

which implies: $\text{sgn}\left(\frac{1}{1+e^{-\langle\bar{\mathbf{w}}_l,\phi_l(\mathbf{x})\rangle}} - \frac{1}{2}\right) = -1.$

- If $\prod_{i=1}^{l} x_i = 1$, then we have:

$$\frac{1}{1 + e^{\langle \bar{\mathbf{w}}_l, \phi_l(\mathbf{x}) \rangle}} < \varepsilon < 1/2, \tag{111}$$

which implies:

$$\begin{aligned}
\text{sgn}\left(\frac{1}{1 + e^{-\langle \bar{\mathbf{w}}_l, \phi_l(\mathbf{x}) \rangle}} - \frac{1}{2}\right) &= \text{sgn}\left(\frac{1}{1 + e^{-\langle \bar{\mathbf{w}}_l, \phi_l(\mathbf{x}) \rangle}} - 1 + 1 - \frac{1}{2}\right) \\
&= \text{sgn}\left(-\frac{1}{1 + e^{\langle \bar{\mathbf{w}}_l, \phi_l(\mathbf{x}) \rangle}} + \frac{1}{2}\right) \\
&= 1.
\end{aligned} \tag{112}$$

Therefore, for all $\mathbf{x} \in \{\pm 1\}^{d+l-1}$ with $x_{d+l-1} = \prod_{i=1}^{l-1} x_i$, it holds:

$$\text{sgn}\left(\langle \bar{\mathbf{w}}_l, \phi_l(\mathbf{x}) \rangle\right) = \prod_{i=1}^{l} x_i. \tag{113}$$

$\square$

### E.3.4 END-TO-END PRE-TRAINING RESULT

**Theorem 4.** *Let $d \geq 2$, $0 < \epsilon, \delta < 1$ and $0 < p < 3/4$. Let $\mathcal{D}(p)$ be a (parameterized) distribution over sequences defined as in equation 27. Consider SGD (Algorithm 3) with per-position iterations $T_1 = \tilde{O}\left(\frac{d^7}{\delta \varepsilon^4}\right)$, $T_{2a} = \tilde{O}\left(\max\left\{\frac{d^9}{\delta \varepsilon^6}, \frac{d^7}{p^2 \delta \varepsilon^4}\right\}\right)$, $T_2, \ldots, T_l = \tilde{O}\left(\frac{d^9}{\delta \varepsilon^6}\right)$, total iterations $T = \max\left\{T_1, T_{2a}, \frac{2 \max_{l \in \{2, \ldots, d\}} T_l}{p}, \frac{8}{p} \ln \frac{2}{\delta}\right\}$ and with regularization coefficients $\lambda_1 = \tilde{\Theta}\left(\frac{d^{3/4}}{\delta^{1/4} T_1^{1/4}}\right)$, $\lambda_{2a} = \tilde{\Theta}\left(\min\left\{\frac{d^{\frac{3}{4}} p^{\frac{1}{2}}}{\delta^{1/4} T_{2a}^{1/4}}, \frac{d}{\delta^{1/3} T_{2a}^{1/3}}\right\}\right)$, $\lambda_l = \tilde{\Theta}\left(\frac{d}{\delta^{1/3} T_l^{1/3}}\right)$ for all $2 \leq l \leq d$. Then, with probability at least $1 - \delta$, SGD returns hypotheses $\bar{\mathbf{w}}_1 \in \mathcal{H}_1, \bar{\boldsymbol{\theta}}_{2a} \in \mathcal{H}_{2a}, \bar{\mathbf{w}}_2 \in \mathcal{H}_2, \ldots, \bar{\mathbf{w}}_d \in \mathcal{H}_d$ that induce $h \in \mathcal{H}_{AR}^{Lin}$ for which for all $\mathbf{x} \in \{\pm 1\}^d$ it holds:*

$$\begin{aligned}
\left|\pi_h\left(\left(x_1, x_1 x_2, \ldots, \prod_{i=1}^{d} x_i, \text{<EOS>}\right) \middle| \mathbf{x}\right) - p\right| &\lesssim \varepsilon, \\
\left|\pi_h\left(\left(\prod_{i=1}^{d} x_i, \text{<EOS>}\right) \middle| \mathbf{x}\right) - \frac{1-p}{2}\right| &\lesssim \varepsilon.
\end{aligned} \tag{114}$$

*Furthermore, if $\varepsilon < (d+1) \min\left\{\frac{1}{2}, \ln\left(\frac{1+p}{1-p}\right), \left|\ln\left(\frac{1-p}{2p}\right)\right|\right\}$, then it holds:*

$$\hat{h}\left(\mathbf{x}; \{\bar{\mathbf{w}}_1, \bar{\boldsymbol{\theta}}_{2a}, \bar{\mathbf{w}}_2, \ldots, \bar{\mathbf{w}}_d\}\right) = \begin{cases} (x_1, \text{<EOS>}), & \text{if } p < 1/3, \\ \left(x_1, x_1 x_2, \ldots, \prod_{i=1}^{d} x_i, \text{<EOS>}\right), & \text{otherwise.} \end{cases} \tag{115}$$

*Proof.* The proof strategy is to bound the probability of either not sampling enough 'long' sequences or of SGD returning a 'problematic' hypothesis at some position. We invoke Corollaries 1 to 3 for $\frac{\varepsilon}{d+1}, \frac{2}{2(d+1)}$. **First position** First, we calculate the probability of outputting the parity $\prod_{i=1}^{d} x_i$ in the first position for the unregularized solution $\mathbf{w}_0$:

$$\begin{aligned}
\mathbb{P}\left[\tilde{h}_1(\mathbf{x}; \mathbf{w}_0) = \prod_{i=1}^{d} x_i\right] &= \frac{1 + p x_1}{2} \frac{\prod_{i=1}^{d} x_i + 1}{2} + \frac{1 - p x_1}{2} \frac{1 - \prod_{i=1}^{d} x_i}{2} \\
&= \frac{1 + p \prod_{i=2}^{d} x_i}{2}.
\end{aligned} \tag{116}$$

From Corollary 1, after $T \geq T_1\left(\frac{\varepsilon}{d+1}, \frac{\delta}{2(d+1)}\right)$ iterations for $\lambda_1 = \tilde{\Theta}\left(\frac{d^{3/4}}{\delta^{1/4} T_1^{1/4}}\right)$, SGD returns $\bar{\mathbf{w}}_1$ such that the following hold with probability less than $\frac{\delta}{2(d+1)}$:

$$\left|\mathbb{P}\left[\tilde{h}_1(\mathbf{x}; \bar{\mathbf{w}}_1) = x_1\right] - \frac{1+p}{2}\right| \geq \frac{\varepsilon}{d+1}, \tag{117}$$

and

$$\left| \mathbb{P}\left[\tilde{h}_1(\mathbf{x}; \bar{\mathbf{w}}_1) = \prod_{i=1}^{d} x_i\right] - \frac{1 + p\prod_{i=2}^{d} x_i}{2} \right|$$
$$= \left| \frac{\prod_{i=1}^{d} x_i + 1}{2}\left(\frac{1}{1 + e^{-\langle\bar{\mathbf{w}}_1, \mathbf{x}\rangle}} - \frac{1 + px_1}{2}\right) + \frac{1 - \prod_{i=1}^{d} x_i}{2}\left(\frac{1}{1 + e^{\langle\bar{\mathbf{w}}_1, \mathbf{x}\rangle}} - \frac{1 + px_1}{2}\right) \right|$$
$$\geq \frac{\varepsilon}{d+1}, \tag{118}$$

and, since $\frac{\varepsilon}{d+1} < \ln\left(\frac{1+p}{1-p}\right)$, it also holds:

$$\hat{h}_1(\mathbf{x}; \bar{\mathbf{w}}_1) = \mathrm{sgn}\left(\langle\bar{\mathbf{w}}_1, \mathbf{x}\rangle\right) = x_1. \tag{119}$$

**Second position (2a)** We calculate the probability of stopping the generation given that the first token has been equal to the parity $\prod_{i=1}^{d} x_i$ for the unregularized solution $\boldsymbol{\theta}_0$:

$$\mathbb{P}\left[\tilde{h}_{2a}(\mathbf{x}; \boldsymbol{\theta}_0) = \texttt{<EOS>}\middle| \tilde{h}_1(\mathbf{x}; \bar{\mathbf{w}}_1) = \prod_{i=1}^{d} x_i\right] = \frac{1-p}{1+p}\mathbb{1}\left\{\prod_{i=1}^{d} x_i = x_1\right\} + \mathbb{1}\left\{\prod_{i=1}^{d} x_i = -x_1\right\}$$
$$= \frac{1-p}{1+p}\frac{1 + \prod_{i=2}^{d} x_i}{2} + \frac{1 - \prod_{i=2}^{d} x_i}{2}$$
$$= \frac{1 - p\prod_{i=2}^{d} x_i}{1+p}. \tag{120}$$

From Corollary 2, after $T \geq T_{2a}\left(\frac{\varepsilon}{d+1}, \frac{\delta}{2(d+1)}\right)$ iterations for $\lambda_{2a} = \tilde{\Theta}\left(\min\left\{\frac{d^{\frac{3}{4}}p^{\frac{1}{2}}}{\delta^{1/4}T_{2a}^{1/4}}, \frac{d}{\delta^{1/3}T_{2a}^{1/3}}\right\}\right)$, SGD returns $\bar{\mathbf{w}}_{2a}, \bar{b}_{2a}$ such that, it holds with probability less than $\frac{\delta}{2(d+1)}$:

$$\left| \mathbb{P}\left[\tilde{h}_{2a}(\mathbf{x}; \bar{\boldsymbol{\theta}}_{2a}) = \epsilon\middle| \tilde{h}_1(\mathbf{x}; \bar{\mathbf{w}}_1) = x_1\right] - \frac{2p}{1+p} \right| \geq \frac{\varepsilon}{d+1}, \tag{121}$$

and

$$\left| \mathbb{P}\left[\tilde{h}_{2a}(\mathbf{x}; \bar{\boldsymbol{\theta}}_{2a}) = \texttt{<EOS>}\middle| \tilde{h}_1(\mathbf{x}; \bar{\mathbf{w}}_1) = \prod_{i=1}^{d} x_i\right] - \frac{1 - p\prod_{i=2}^{d} x_i}{1+p} \right|$$
$$= \left| \frac{\prod_{i=2}^{d} x_i + 1}{2}\left(\frac{1}{1 + e^{-\langle\bar{\mathbf{w}}_{2a}, \mathbf{x}\rangle - \bar{b}_{2a}}} - \frac{1-p}{1+p}\right) + \frac{1 - \prod_{i=2}^{d} x_i}{2}\left(\frac{1}{1 + e^{-\langle\bar{\mathbf{w}}_{2a}, \mathbf{x}\rangle - \bar{b}_{2a}}} - 1\right) \right|$$
$$\geq \frac{\varepsilon}{d+1}, \tag{122}$$

and, since $\frac{\varepsilon}{d+1} < \left|\ln\left(\frac{1-p}{2p}\right)\right|$, it holds:

$$\mathrm{sgn}\left(\langle\bar{\mathbf{w}}_{2a}, \phi_2(\mathbf{x})\rangle + \bar{b}_{2a}\right) = \mathrm{sgn}\left(\ln\left(\frac{1-p}{2p}\right)\right) = \begin{cases} -1, & \text{if } p \geq 1/3, \\ +1, & \text{if } p < 1/3 \end{cases}, \tag{123}$$

for all $\mathbf{x} \in \{\pm 1\}^{d+1}$ with $x_1 x_{d+1} = 1$.

**Count of long sequences** Let $z_1, \ldots, z_T \sim \mathcal{D}(p)$ and let $T_{\text{long}} = \sum_{i=1}^{T} \mathbb{1}\{|z_i| > d + 2\}$ be the count of "long" samples. By the definition of $\mathcal{D}(p)$, it is: $\mathbb{E}[T_{\text{long}}] = Tp$. By the multiplicative Chernoff bound, it holds:

$$\mathbb{P}\left[T_{\text{long}} \leq \frac{Tp}{2}\right] \leq e^{-\frac{Tp}{8}}, \tag{124}$$

hence with probability at least $1 - \frac{\delta}{2}$, we have:

$$T_{\text{long}} > \frac{Tp}{2}, \tag{125}$$

as long as $T > \frac{8}{p} \ln \frac{2}{\delta}$. If further, $T \geq \frac{2 \max_{l \in \{2,\ldots,d\}} T_l}{p}$, then with probability at least $1 - \frac{\delta}{2}$, we have: $T_{\text{long}} > T_l$ for all $l \in \{2, \ldots, d\}$.

**Positions 2 to $d$** From Corollary 3, after $T_{\text{long}} \geq T_l \left( \frac{\varepsilon}{d+1}, \frac{\delta}{2(d+1)} \right)$ iterations for $\lambda_l = \tilde{\Theta} \left( \frac{d}{\delta^{1/3} T_l^{1/3}} \right)$, SGD returns $\bar{\mathbf{w}}_l$ such that, with probability less than $\delta/2d$:

$$\left| \mathbb{P} \left[ \tilde{h}_l(\mathbf{x}; \bar{\mathbf{w}}_l) = \prod_{i=1}^{l} x_i \,\middle|\, \tilde{h}_1(\mathbf{x}; \bar{\mathbf{w}}_1) = x_1, \ldots, \tilde{h}_{l-1}(\mathbf{x}; \bar{\mathbf{w}}_{l-1}) = x_{l-1} \right] - 1 \right| \geq \frac{\varepsilon}{d+1}, \tag{126}$$

and, since $\frac{\varepsilon}{d+1} < \frac{1}{2}$:

$$\text{sgn} \left( \langle \bar{\mathbf{w}}_l, \phi_l(\mathbf{x}) \rangle \right) = \prod_{i=1}^{l} x_i \tag{127}$$

for all $\mathbf{x} \in \{\pm 1\}^{d+l-1}$ with $x_{d+1} = \prod_{i=1}^{l-1} x_i$.

**Union bound** Let $E_{\text{LONG}}$ denote the event that $T_{\text{long}} > T_l$ for all $2 \leq l \leq d$ and denote by $P_i$ the event that the above guarantees hold for position $i$. Then, by union bound, we have:

$$\mathbb{P} \left[ E_{\text{LONG}} \wedge P_1 \wedge P_{2a} \wedge P_2 \wedge \ldots P_d \right] = \mathbb{P} \left[ E_{\text{LONG}} \right] \mathbb{P} \left[ P_1 \wedge P_{2a} \wedge P_2 \wedge \ldots P_d | E_{\text{LONG}} \right]$$

$$\geq \left( 1 - \frac{\delta}{2} \right) \left( 1 - \mathbb{P} \left[ \exists i \in \{1, 2a, 2, \ldots, d\} : \neg P_i \right] \right)$$

$$\geq \left( 1 - \frac{\delta}{2} \right) \left( 1 - \sum_{i \in \{1, 2a, 2, \ldots, d\}} \frac{\delta}{2(d+1)} \right) \geq 1 - \delta. \tag{128}$$

Thus, with probability at least $1 - \delta$, SGD returns a hypothesis $h$ whose probability of correct, long and short, sequences is:

$$\pi_h \left( \left( x_1, x_1 x_2, \ldots, \prod_{i=1}^{d} x_i, \texttt{<EOS>} \right) \middle| \mathbf{x} \right) =$$

$$= \mathbb{P} \left[ \tilde{h}_1(\mathbf{x}; \mathbf{w}_1) = x_1 \right] \cdot \ldots \cdot \mathbb{P} \left[ \tilde{h}_d(\mathbf{x}; \mathbf{w}_d) = \prod_{i=1}^{d} x_i \,\middle|\, \tilde{h}_1(\mathbf{x}; \bar{\mathbf{w}}_1) = x_1, \ldots, \tilde{h}_{l-1}(\mathbf{x}; \bar{\mathbf{w}}_{l-1}) = x_{l-1} \right]$$

$$= \left( \frac{1+p}{2} + \xi_1(\mathbf{x}) \right) \left( \frac{2p}{1+p} + \xi_{2a}(\mathbf{x}) \right) (1 - \xi_2(\mathbf{x})) \ldots (1 - \xi_d(\mathbf{x})), \tag{129}$$

and:

$$\pi_h \left( \left( \prod_{i=1}^{d} x_i, \texttt{<EOS>} \right) \middle| \mathbf{x} \right)$$

$$= \mathbb{P} \left[ \tilde{h}_1(\mathbf{x}; \bar{\mathbf{w}}_1) = \prod_{i=1}^{d} x_i \right] \mathbb{P} \left[ \tilde{h}_{2a}(\mathbf{x}; \bar{\boldsymbol{\theta}}_{2a}) = \texttt{<EOS>} \,\middle|\, \tilde{h}_1(\mathbf{x}; \bar{\mathbf{w}}_1) = \prod_{i=1}^{d} x_i \right] \tag{130}$$

$$= \left( \frac{1 + p \prod_{i=2}^{d} x_i}{2} + \xi_1(\mathbf{x}) \right) \left( \frac{1 - p \prod_{i=2}^{d} x_i}{1+p} + \xi_{2a}(\mathbf{x}) \right),$$

where $|\xi_1(\mathbf{x})|, \ldots, |\xi_d(\mathbf{x})| \in O(\varepsilon/d)$ for all $\mathbf{x}$. In other words,

$$\left| \pi_h \left( \left( x_1, x_1 x_2, \ldots, \prod_{i=1}^{d} x_i, \texttt{<EOS>} \right) \middle| \mathbf{x} \right) - p \right| \lesssim \varepsilon. \tag{131}$$

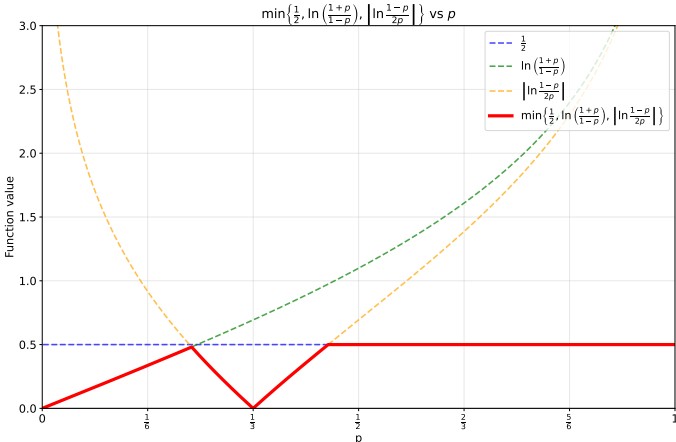

Figure 41: The functional form of the error term $\varepsilon$ vs $p \in (0, 1)$ in Theorem 4.

and

$$\left| \pi_h \left( \left( \prod_{i=1}^{d} x_i, \text{<EOS>} \right) \middle| \mathbf{x} \right) - \frac{1-p}{2} \right| \lesssim \varepsilon. \tag{132}$$

Furthermore, from eqs. equation 119, equation 123, equation 127, for all $\mathbf{x} \in \{\pm 1\}^d$, we have:

- $\hat{h}_1(\mathbf{x}; \bar{\mathbf{w}}_1) = \text{sgn}(\langle \bar{\mathbf{w}}_1, \mathbf{x} \rangle) = x_1,$

- 
$$\text{sgn}\left( \left\langle \bar{\mathbf{w}}_{2a}, \phi_2 \left( \mathbf{x}, \hat{h}^{(1)}(\mathbf{x}; \bar{\mathbf{w}}_1) \right) \right\rangle + \bar{b}_{2a} \right) = \text{sgn}\left( \ln \left( \frac{1-p}{2p} \right) \right)$$
$$= \begin{cases} -1, & \text{if } 1/2 > p \geq 1/3, \\ +1, & \text{if } p < 1/3 \end{cases}, \tag{133}$$

  hence
  $$\hat{h}_{2a}(\mathbf{x}; \bar{\boldsymbol{\theta}}_{2a}) = \begin{cases} \text{<EOS>}, & \text{if } 1/2 > p \geq 1/3, \\ \epsilon, & \text{if } p < 1/3, \end{cases}$$

  and

- $\hat{h}_l(\mathbf{x}; \bar{\mathbf{w}}_l) = \begin{cases} \prod_{i=1}^{l} x_i, & \text{if } \hat{h}_{2a}(\mathbf{x}; \bar{\boldsymbol{\theta}}_{2a}) \neq \text{<EOS>}, \\ \epsilon, & \text{if } \hat{h}_{2a}(\mathbf{x}; \bar{\boldsymbol{\theta}}_{2a}) = \text{<EOS>}, \end{cases}$ for $2 \leq l \leq d.$

This implies:

$$\hat{h}\left( \mathbf{x}; \{\bar{\mathbf{w}}_1, \bar{\boldsymbol{\theta}}_{2a}, \bar{\mathbf{w}}_2, \ldots, \bar{\mathbf{w}}_d\} \right) = \begin{cases} (x_1, \text{<EOS>}), & \text{if } p < 1/3, \\ \left( x_1, x_1 x_2, \ldots, \prod_{i=1}^{d} x_i, \text{<EOS>} \right), & \text{otherwise.} \end{cases} \tag{134}$$

$\square$

Theorem 1 in the main text follows as a corollary by Theorem 4 for $p = p_{\text{cot}}$. See Figure 41 for the functional form of the error term $\varepsilon(p)$. In particular, the functional form also explains the difficulties faced in training transformers for $p_{\text{cot}} \approx 1/3$.

### E.4 POST-TRAINING

We complement the analysis of the previous subsection to show now a positive learning result on the combined pre- & post-training procedure. Precisely, we show that, when $p_{\text{cot}} < 1/3$ which is the hard regime for pre-training, only $O\left(\log \frac{1-p_{\text{cot}}}{p_{\text{cot}}}\right)$ RL rounds suffice for the pre-trained model to reach perfect accuracy. Furthermore, we prove that the length of the answer grows in a predictable way.

**Theorem 5.** *Let* $d \geq 2, \delta \in (0,1)$ *and* $p_{\text{cot}} \in (0,1/3)$. *Let* $\mathcal{D}(p_{\text{cot}})$ *be a (parameterized) distribution over sequences defined as in equation 27. Let* $\varepsilon \leq c_0 \frac{p_{\text{cot}}}{1-p_{\text{cot}}}$ *for some constant* $c_0 > 0$. *Consider pre-training (Algorithm 3) per-position iterations* $T_1 = \tilde{O}\left(\frac{d^7}{\delta\varepsilon^4}\right), T_{2a} = \tilde{O}\left(\max\left\{\frac{d^9}{\delta\varepsilon^6}, \frac{d^7}{p^2\delta\varepsilon^4}\right\}\right), T_2, \ldots, T_l = \tilde{O}\left(\frac{d^9}{\delta\varepsilon^6}\right)$, *total iterations* $T = \max\left\{T_1, T_{2a}, \frac{2\max_{l\in\{2,\ldots,d\}}T_l}{p}, \frac{8}{p_{\text{cot}}}\ln\frac{2}{\delta}\right\}$ *and with regularization coefficients* $\lambda_1 = \tilde{\Theta}\left(\frac{d^{3/4}}{\delta^{1/4}T_1^{1/4}}\right), \lambda_{2a} = \tilde{\Theta}\left(\min\left\{\frac{d^{\frac{3}{4}}p_{\text{cot}}^{\frac{1}{2}}}{\delta^{1/4}T_{2a}^{1/4}}, \frac{d}{\delta^{1/3}T_{2a}^{1/3}}\right\}\right), \lambda_l = \tilde{\Theta}\left(\frac{d}{\delta^{1/3}T_l^{1/3}}\right)$ *for all* $2 \leq l \leq d$. *Furthermore, consider post-training with the STaR algorithm solving Problem LIN-RL at each round and denote by* $h^{(n)} \in \mathcal{H}_{\text{AR}}^{\text{Lin}}$ *the model returned at the end of the* $n$-th *STaR round. Then, there exists* $n^\star = O\left(\log\frac{1-p_{\text{cot}}}{p_{\text{cot}}}\right)$, *such that, if:*

1. *SGD steps per-position per-STaR round are as follows:* $T_1 = \tilde{O}\left(\frac{d^7}{\delta\varepsilon^4}\right)$, $T_{2a} = \tilde{O}\left(\max\left\{\frac{d^9}{\delta\varepsilon^6}, \frac{d^7}{p_{\text{cot}}^2\delta\varepsilon^4}\right\}\right), T_2, \ldots, T_l = \tilde{O}\left(\frac{d^9}{\delta\varepsilon^6}\right)$ *and total iterations* $T = \max\left\{T_1, T_{2a}, \frac{2\max_{l\in\{2,\ldots,d\}}T_l}{p_{\text{cot}}}, \frac{8}{p_{\text{cot}}}\ln\frac{2\log\left(\frac{1-p_{\text{cot}}}{p_{\text{cot}}}\right)}{\delta}\right\}$, *and*

2. *Regularization strengths per-position per-STaR round are as follows:* $\lambda_1 = \tilde{\Theta}\left(\frac{d^{3/4}}{\delta^{1/4}T_1^{1/4}}\right)$, $\lambda_{2a} = \tilde{\Theta}\left(\min\left\{\frac{d^{\frac{3}{4}}p_{\text{cot}}^{\frac{1}{2}}}{\delta^{1/4}T_{2a}^{1/4}}, \frac{d}{\delta^{1/3}T_{2a}^{1/3}}\right\}\right), \lambda_l = \tilde{\Theta}\left(\frac{d}{\delta^{1/3}T_l^{1/3}}\right)$ *for all* $2 \leq l \leq d$,

*with probability at least* $1 - \delta$ *over sampling from* $\mathcal{D}(p_{\text{cot}})$ *(during pre-training) and over sampling from models* $\tilde{h}_0, \ldots, \tilde{h}_{n^\star}$ *(during post-training), post-training returns* $\bar{\mathbf{w}}_1^{(1)}, \ldots, \bar{\mathbf{w}}_1^{(n^\star)} \in \mathcal{H}_1, \bar{\boldsymbol{\theta}}_{2a}^{(1)}, \ldots, \bar{\boldsymbol{\theta}}_{2a}^{(n^\star)} \in \mathcal{H}_{2a}, \bar{\mathbf{w}}_2^{(1)}, \ldots, \bar{\mathbf{w}}_2^{(n^\star)} \in \mathcal{H}_2, \ldots, \bar{\mathbf{w}}_d^{(1)}, \ldots, \bar{\mathbf{w}}_d^{(n^\star)} \in \mathcal{H}_d$ *that induce* $h^{(1)}, \ldots, h^{(n^\star)} \in \mathcal{H}_{\text{AR}}^{\text{Lin}} : \mathcal{X} \mapsto \mathcal{Y}$ *for which for all* $\mathbf{x} \in \mathcal{X}$ *it holds:*

$$\left|\pi_{h^{(n)}}\left(\left(x_1, x_1x_2, \ldots, \prod_{i=1}^{d}x_i, \texttt{<EOS>}\right)\middle|\mathbf{x}\right) - q_n\right| \lesssim \varepsilon, \tag{135}$$

*where* $|q_n - p_n| \lesssim 2^n\varepsilon$ *and* $p_{n+1} = \frac{2p_n}{1+p_n}$ *for all* $n \leq n^\star$ *with* $p_0 = p_{\text{cot}}$, *and also:*

$$\hat{h}^{(n^\star)}\left(\mathbf{x}; \left\{\bar{\mathbf{w}}_1^{(n^\star)}, \bar{\boldsymbol{\theta}}_{2a}^{(n^\star)}, \bar{\mathbf{w}}_2^{(n^\star)}, \ldots, \bar{\mathbf{w}}_d^{(n^\star)}\right\}\right) = \left(x_1, x_1x_2, \ldots, \prod_{i=1}^{d}x_i, \texttt{<EOS>}\right). \tag{136}$$

*Proof.* The proof proceeds in two main steps. We first observe that the STaR objective for the $n$'th round can be re-written as next-token prediction with respect to a shifted version $\mathcal{D}(q_n)$ of the original distribution $\mathcal{D}(p_{\text{cot}})$, where $0 < q_n < 1$. We then bound the deviation of sequence $q_n$ from a "noise-less" sequence $p_n$ that describes the evolution of the proportion of long data in the data mix, and whose closed form is available.

From Theorem 4, for $p = p_{\text{cot}}$ and $\delta = \delta/2$, with probability at least $1 - \delta/2$, we obtain model $h^{(0)} \in \mathcal{H}_{\text{AR}}^{\text{Lin}}$, such that for all $\mathbf{x} \in \{\pm 1\}$ it holds:

$$
\left| \pi_{h^{(0)}} \left( \left( x_1, x_1 x_2, \ldots, \prod_{i=1}^{d} x_i, \texttt{<EOS>} \right) \middle| \mathbf{x} \right) - p_{\text{cot}} \right| \lesssim \varepsilon,
$$
$$
\left| \pi_{h^{(0)}} \left( \left( \prod_{i=1}^{d} x_i, \texttt{<EOS>} \right) \middle| \mathbf{x} \right) - \frac{1 - p_{\text{cot}}}{2} \right| \lesssim \varepsilon,
$$
(137)

and

$$
\hat{h}^{(0)} \left( \mathbf{x}; \{ \bar{\mathbf{w}}_1, \bar{\boldsymbol{\theta}}_{2a}, \bar{\mathbf{w}}_2, \ldots, \bar{\mathbf{w}}_d \} \right) = (x_1, \texttt{<EOS>}) .
$$
(138)

The previous point-wise guarantee also implies:

$$
\left| \mathbb{E}_{\mathbf{x} \sim \text{Rad}(1/2)^{\otimes d}} \left[ \pi_{h^{(0)}} \left( \left( x_1, x_1 x_2, \ldots, \prod_{i=1}^{d} x_i, \texttt{<EOS>} \right) \middle| \mathbf{x} \right) \right] - p_{\text{cot}} \right| \lesssim \varepsilon,
$$
$$
\left| \mathbb{E}_{\mathbf{x} \sim \text{Rad}(1/2)^{\otimes d}} \left[ \pi_{h^{(0)}} \left( \left( \prod_{i=1}^{d} x_i, \texttt{<EOS>} \right) \middle| \mathbf{x} \right) \right] - \frac{1 - p_{\text{cot}}}{2} \right| \lesssim \varepsilon,
$$
(139)

Recall the form of the STaR problem for the first round:

$$
\min_{\mathbf{w}_1, \mathbf{w}_{2a}, b, \mathbf{w}_2, \ldots, \mathbf{w}_d} \mathcal{L} \equiv \mathbb{E}_{\substack{\mathbf{x} \sim \text{Rad}(1/2)^{\otimes d} \\ y \sim \pi_{h^{(0)}}(\cdot | \mathbf{x})}} [\mathbb{1}\{|y| = 2\} \mathcal{L}_{\text{short}} + \mathbb{1}\{|y| = d+1\} \mathcal{L}_{\text{long}} | r_{\text{cot}}(\mathbf{x}, y) = 1]
$$
(140)

where $\mathcal{L}_{\text{short}} = l^{(1)}(\mathbf{w}_1, (\mathbf{x}, y_1)) + l^{(2a)}((\mathbf{w}_{2a}, b), ([\mathbf{x}, y_1], \tilde{y}_2))$ and $\mathcal{L}_{\text{long}} = l^{(1)}(\mathbf{w}_1, (\mathbf{x}, y_1)) + l^{(2a)}(\{\mathbf{w}_{2a}, b\}, ((\mathbf{x}, y_1), \tilde{y}_2)) + \ldots + l^{(d)}(\mathbf{w}_d, ((\mathbf{x}, y_1, \ldots, y_{d-1}), y_d))$. Note that, with probability at least $1 - \delta/2$, the objective can be re-written as:

$$
\mathcal{L} = \mathbb{P}_{\mathbf{x}, y} \left[ y = \left( x_1, x_1 x_2, \ldots, \prod_{i=1}^{d} x_i, \texttt{<EOS>} \right) \middle| r_{\text{cot}}(\mathbf{x}, y) = 1 \right] \mathbb{E}_{\substack{\mathbf{x} \sim \text{Rad}(1/2)^{\otimes d}, \\ y = \left( x_1, x_1 x_2, \ldots, \prod_{i=1}^{d} x_i, \texttt{<EOS>} \right)}} [\mathcal{L}_{\text{long}}]
$$
$$
+ \mathbb{P}_{\mathbf{x}, y} \left[ y = \left( \prod_{i=1}^{d} x_i, \texttt{<EOS>} \right) \middle| r_{\text{cot}}(\mathbf{x}, y) = 1 \right] \mathbb{E}_{\substack{\mathbf{x} \sim \text{Rad}(1/2)^{\otimes d}, \\ y = \left( x_1, x_1 x_2, \ldots, \prod_{i=1}^{d} x_i, \texttt{<EOS>} \right)}} [\mathcal{L}_{\text{short}}]
$$
$$
= \frac{\mathbb{E}_{\mathbf{x}} \left[ \pi_{h^{(0)}} \left( \left( x_1, x_1 x_2, \ldots, \prod_{i=1}^{d} x_i, \texttt{<EOS>} \right) \middle| \mathbf{x} \right) \right]}{\mathbb{E}_{\mathbf{x}} \left[ \pi_{h^{(0)}} \left( \left( x_1, x_1 x_2, \ldots, \prod_{i=1}^{d} x_i, \texttt{<EOS>} \right) \middle| \mathbf{x} \right) \right] + \mathbb{E}_{\mathbf{x}} \left[ \pi_{h^{(0)}} \left( \left( \prod_{i=1}^{d} x_i, \texttt{<EOS>} \right) \middle| \mathbf{x} \right) \right]}
$$
$$
\cdot \mathbb{E}_{\substack{\mathbf{x} \sim \text{Rad}(1/2)^{\otimes d}, \\ y = \left( x_1, x_1 x_2, \ldots, \prod_{i=1}^{d} x_i, \texttt{<EOS>} \right)}} [\mathcal{L}_{\text{long}}]
$$
$$
+ \frac{\mathbb{E}_{\mathbf{x}} \left[ \pi_{h^{(0)}} \left( \left( \prod_{i=1}^{d} x_i, \texttt{<EOS>} \right) \middle| \mathbf{x} \right) \right]}{\mathbb{E}_{\mathbf{x}} \left[ \pi_{h^{(0)}} \left( \left( x_1, x_1 x_2, \ldots, \prod_{i=1}^{d} x_i, \texttt{<EOS>} \right) \middle| \mathbf{x} \right) \right] + \mathbb{E}_{\mathbf{x}} \left[ \pi_{h^{(0)}} \left( \left( \prod_{i=1}^{d} x_i, \texttt{<EOS>} \right) \middle| \mathbf{x} \right) \right]}
$$
$$
\cdot \mathbb{E}_{\substack{\mathbf{x} \sim \text{Rad}(1/2)^{\otimes d}, \\ y = \left( \prod_{i=1}^{d} x_i, \texttt{<EOS>} \right)}} [\mathcal{L}_{\text{short}}]
$$
$$
= \frac{p_0 + \zeta_0}{\frac{p_0 + 1}{2} + \zeta_0 + \xi_0} \mathbb{E}_{\substack{\mathbf{x} \sim \text{Rad}(1/2)^{\otimes d}, \\ y = \left( x_1, x_1 x_2, \ldots, \prod_{i=1}^{d} x_i, \texttt{<EOS>} \right)}} [\mathcal{L}_{\text{long}}]
$$
$$
+ \left( 1 - \frac{p_0 + \zeta_0}{\frac{p_0 + 1}{2} + \zeta_0 + \xi_0} \right) \mathbb{E}_{\substack{\mathbf{x} \sim \text{Rad}(1/2)^{\otimes d}, \\ y = \left( \prod_{i=1}^{d} x_i, \texttt{<EOS>} \right)}} [\mathcal{L}_{\text{short}}]
$$
$$
= \mathbb{E}_{(\mathbf{x}, y) \sim \mathcal{D}(q_1)} [\mathbb{1}\{|y| = 2\} \mathcal{L}_{\text{short}} + \mathbb{1}\{|y| = d+1\} \mathcal{L}_{\text{long}}],
$$

where $q_1 := \frac{p_0 + \zeta_0}{\frac{p_0 + 1}{2} + \zeta_0 + \xi_0}$ with $|\zeta_0|, |\xi_0| \lesssim \varepsilon$ (from eq. equation 139) and the first and second inequality holds from the law of total expectation and the definition of the conditional probability.

In other words, we showed that the reinforcement learning risk with reward $r_{\mathrm{cot}}$ during the first round of STaR is equal to a next-token prediction risk, where the distribution is the mixture of long and short strings but with shifted mixture weight $q_1$. Thus, from Theorem 4 for $p = q_1$ and $\delta = \delta_1 \in (0,1)$ with probability $(1 - \delta_1)(1 - \delta/2)$, SGD returns model $h^{(1)} \in \mathcal{H}_{\mathrm{AR}}^{\mathrm{Lin}}$, such that:

$$
\left| \pi_{h^{(1)}} \left( \left( x_1, x_1 x_2, \dots, \prod_{i=1}^{d} x_i, \texttt{<EOS>} \right) \middle| \mathbf{x} \right) - q_1 \right| \lesssim \varepsilon,
$$
$$
\left| \pi_{h^{(1)}} \left( \left( \prod_{i=1}^{d} x_i, \texttt{<EOS>} \right) \middle| \mathbf{x} \right) - \frac{1 - q_1}{2} \right| \lesssim \varepsilon,
$$
(141)

and

$$
\hat{h}^{(1)} \left( \mathbf{x}; \{\bar{\mathbf{w}}_1, \bar{\boldsymbol{\theta}}_{2a}, \bar{\mathbf{w}}_2, \dots, \bar{\mathbf{w}}_d\} \right) = \begin{cases} (x_1, \texttt{<EOS>}), & \text{if } q_1 < 1/3, \\ \left( x_1, x_1 x_2, \dots, \prod_{i=1}^{d} x_i, \texttt{<EOS>} \right), & \text{otherwise.} \end{cases}
$$
(142)

Therefore, by induction, as long as $q_n < 1/2$, invoking Theorem 1 $n$ times for $\delta = \delta_i \in (0,1)$ and $p = q_i$, with probability at least $\prod_{i=0}^{n}(1 - \delta_i) \geq 1 - \sum_{i=0}^{n} \delta_i$, post-training returns model $h^{(n)} \in \mathcal{H}_{\mathrm{AR}}^{\mathrm{Lin}}$ such that:

$$
\left| \pi_{h^{(n)}} \left( \left( x_1, x_1 x_2, \dots, \prod_{i=1}^{d} x_i, \texttt{<EOS>} \right) \middle| \mathbf{x} \right) - q_n \right| \lesssim \varepsilon,
$$
$$
\left| \pi_{h^{(n)}} \left( \left( \prod_{i=1}^{d} x_i, \texttt{<EOS>} \right) \middle| \mathbf{x} \right) - \frac{1 - q_n}{2} \right| \lesssim \varepsilon,
$$
(143)

and

$$
\hat{h}^{(n)} \left( \mathbf{x}; \{\bar{\mathbf{w}}_1, \bar{\boldsymbol{\theta}}_{2a}, \bar{\mathbf{w}}_2, \dots, \bar{\mathbf{w}}_d\} \right) = \begin{cases} (x_1, \texttt{<EOS>}), & \text{if } q_n < 1/3, \\ \left( x_1, x_1 x_2, \dots, \prod_{i=1}^{d} x_i, \texttt{<EOS>} \right), & \text{otherwise,} \end{cases}
$$
(144)

where $q_{n+1} = \frac{2(q_n + \zeta_n)}{q_n + 1 + 2(\zeta_n + \xi_n)}$ with $|\zeta_n|, |\xi_n| \lesssim \varepsilon$ with $q_0 = p_0$.

We now analyze the perturbed sequence $q_n$. Let $f(q_n, \zeta_n, \xi_n) := q_{n+1} = \frac{2(q_n + \zeta_n)}{q_n + 1 + 2(\zeta_n + \xi_n)}$. We Taylor expand $f$ around $(q_n, 0, 0)$:

$$
\begin{aligned}
f(q_n, \zeta_n, \xi_n) &= f(q_n, 0, 0) + \zeta_n \left. \frac{\partial f}{\partial \zeta_n} \right|_{(q_n, 0, 0)} + \xi_n \left. \frac{\partial f}{\partial \xi_n} \right|_{(q_n, 0, 0)} + R_f(\mathbf{u}, \zeta_n, \xi_n) \\
&= \frac{2q_n}{q_n + 1} + \zeta_n \frac{2 - 2q_n}{(q_n + 1)^2} + \xi_n \frac{-4q_n}{(q_n + 1)^2} + R_f(\mathbf{u}, \zeta_n, \xi_n) \\
&= \frac{2q_n}{q_n + 1} + \frac{2\zeta_n - 2\zeta_n q_n - 4\xi_n q_n}{(q_n + 1)^2} + R_f(\mathbf{u}, \zeta_n, \xi_n),
\end{aligned}
$$
(145)

where we used the Lagrange form of the remainder: $R_f(\mathbf{u}, \zeta_n, \xi_n) = \frac{1}{2} H(0, \mathbf{u})(\zeta_n^2 + \xi_n^2)$, where $\mathbf{u} \in \mathbb{R}^2$ such that: $|u_1| < |\zeta_n|$ and $|u_2| < |\xi_n|$ and $H(q, \zeta_n, \xi_n)$ is the Hessian of $f$ evaluated at $(q, \zeta_n, \xi_n)$. We now bound the deviation of the perturbed sequence $q_{n+1}$ from the un-perturbed one $p_{n+1}$:

$$
\begin{aligned}
|q_{n+1} - p_{n+1}| &= \left| f(q_n, \zeta_n, \xi_n) - \frac{2p_n}{p_n + 1} \right| \\
&= \left| \frac{2q_n}{q_n + 1} + \frac{2\zeta_n - 2\zeta_n q_n - 4\xi_n q_n}{(q_n + 1)^2} + R_f(\mathbf{u}, \zeta_n, \xi_n) - \frac{2p_n}{p_n + 1} \right| \\
&\leq \left| \frac{2q_n}{q_n + 1} - \frac{2p_n}{p_n + 1} \right| + \left| \frac{2\zeta_n - 2\zeta_n q_n - 4\xi_n q_n}{(q_n + 1)^2} \right| + |R_f(\mathbf{u}, \zeta_n, \xi_n)| \\
&\leq 2|q_n - p_n| + \left| \frac{2\zeta_n - 2\zeta_n q_n - 4\xi_n q_n}{(q_n + 1)^2} \right| + |R_f(\mathbf{u}, \zeta_n, \xi_n)|, \quad \text{provided } q_n > 0.
\end{aligned}
$$
(146)

Since $q_n < 1$ and $|\zeta_n|, |\xi_n| \in O(\varepsilon)$ and $|R_f(\mathbf{u}, \zeta_n, \xi_n)| \lesssim \varepsilon^2$, if we denote the error sequence by $e_n = |q_n - p_n|$, then:

$$e_{n+1} - 2e_n \lesssim \varepsilon. \tag{147}$$

Since $e_0 = 0$, this implies that the error at step $n$ is bounded as:

$$e_n \lesssim 2^n \varepsilon. \tag{148}$$

We want to find an integer $n^\star$ such that model $h^{(n^\star)}$ generates long answers. For this, it suffices to have $q_n > 1/3$, or equivalently:

$$p_{n^\star} - 2^{n^\star} C' \varepsilon > \frac{1}{3}. \tag{149}$$

We show that the unperturbed sequence $p_n = \frac{2p_{n-1}}{p_{n-1}+1}$ admits a closed form solution. Let $u_n = \frac{1}{p_n}$, then:

$$u_n = \frac{u_{n-1}+1}{2} = 1 + \frac{u_0 - 1}{2^n}, \tag{150}$$

or, for the original sequence $p_n$:

$$p_n = \left(1 + \frac{1}{2^n}\left(\frac{1}{p_0} - 1\right)\right)^{-1} = \frac{2^n p_0}{2^n p_0 + 1 - p_0} = \frac{p_0}{p_0 + (1 - p_0)\, 2^{-n}}. \tag{151}$$

Returning back to eq. equation 149, we seek $n^\star \geq 1$ such that:

$$\frac{p_0}{p_0 + (1 - p_0)\, 2^{-n^\star}} - 2^{n^\star} C' \varepsilon > \frac{1}{3}. \tag{152}$$

Assume that $2^{n^\star} C' \varepsilon < \frac{1}{15}$ for some $\varepsilon$ to be specified later. Then, it suffices to have:

$$\frac{p_0}{p_0 + (1 - p_0)\, 2^{-n^\star}} - \frac{1}{15} > \frac{1}{3}, \tag{153}$$

or

$$n^\star > \log\left(\frac{2(1 - p_0)}{3p_0}\right). \tag{154}$$

In that case, the error parameter $\varepsilon$ needs to be as small as:

$$\varepsilon < \frac{1}{C}\frac{p_0}{1 - p_0}, \tag{155}$$

for some $C > 0$. Finally, we pick the smallest $n^\star$ such that $q_{n^\star} > 1/3 + \mu$, where $\mu \geq \omega(e^{-d})$[11]. Note that $p_n < 1/3 \implies p_{n+1} < 1/2$, hence Theorem 4 can be invoked for $q_{n^\star} < 3/4$ if $C$ is sufficiently small. We set $\delta_0 = \delta/2$ and $\delta_1, \ldots, \delta_{n^\star} = \frac{\delta}{2n^\star}$[12], then with probability at least $1 - \delta$, we have:

$$\hat{h}^{(n^\star)}\left(\mathbf{x}; \left\{\bar{\mathbf{w}}_1^{(n^\star)}, \bar{\boldsymbol{\theta}}_{2a}^{(n^\star)}, \bar{\mathbf{w}}_2^{(n^\star)}, \ldots, \bar{\mathbf{w}}_d^{(n^\star)}\right\}\right) = \left(x_1, x_1 x_2, \ldots, \prod_{i=1}^d x_i, \texttt{<EOS>}\right). \tag{156}$$

$\square$

*Remark* 7. Note that the guarantees of Theorem 5 for the number of SGD iterations required per STaR round are very pessimistic: 1) they depend on the original mixture weight $p_{\mathrm{cot}}$ rather than the per-round updated weight $q_n$, and 2) they assume that the models are re-initialized at the origin prior to each STaR round, whereas of course in practice we continue fine-tuning the previously obtained model which speeds up convergence.

*Remark* 8. One can also analyze an (almost offline) version of REINFORCE algorithm with no reward normalization, instead of STaR, by noting that its loss also corresponds to the next-token prediction objective but scaled by a different per-round coefficient.

In particular, if the proportion of long data in the data mixture is not exponentially small, i.e., there exists constant $\kappa \in \mathbb{N}$ such that: $p_{\mathrm{cot}} \in \Omega(d^{-\kappa})$, then by following the previous pre- & post-training recipe on data coming from $\mathcal{D}(p_{cot})$, we can obtain a model that emits a long, correct sequence and predicts the parity of $d$ bits after $O(\mathrm{poly}(d))$ SGD iterations.

---

[11]We require $q_{n^\star}$ to not be exponentially close to 1/3, so that we do not pay $1/\varepsilon$ sample complexity for $\varepsilon < (d+1)\ln\frac{2q_{n^\star}}{1-q_{n^\star}} \asymp (d+1)(q_{n^\star} - 1/3)$

[12]Note that for $\delta_i = \frac{\delta}{2n^\star}$ the regularization terms and SGD iterations grow together with a factor $O\left(\ln\frac{1-p_{\mathrm{cot}}}{p_{\mathrm{cot}}}\right)$, but this gets suppressed by the $\tilde{O}(\cdot)$ notation.

