# OpenReview forum: "How reinforcement learning after next-token prediction facilitates learning"
_ICLR.cc/2026/Conference — ICLR 2026 Poster_

### Official Review · Reviewer_Hty6 · 2025-10-24

**Soundness:** 3
**Presentation:** 2
**Contribution:** 3
**Rating:** 6
**Confidence:** 3

**Summary:**

This paper explains why large language models improve dramatically when reinforcement learning (RL) follows next-token prediction training. The authors prove that next-token prediction alone cannot generalize on hard tasks if long reasoning chains are rare, whereas RL amplifies these rare sequences and enables efficient learning. Experiments on both synthetic and real-world reasoning tasks confirm that RL boosts accuracy and increases response length. The results highlight RL’s role in unlocking generalization from limited but valuable reasoning data.

**Strengths:**

* The paper is the first to provide a formal separation between next-token prediction and next-token prediction plus RL in autoregressive models, giving a novel theoretical account of why the widely-used “pre-train then RL” recipe succeeds.
* By showing that RL can turn a sample-hard, exponentially data-hungry problem into a polynomial-time solvable one whenever long demonstrations are merely polynomially rare, the work directly informs how scarce reasoning data should be leveraged in large-scale model development.

**Weaknesses:**

* All conclusions of the paper are drawn around the task of predicting the parity of d bits given access to a source of sequences, and its generalization to more common reasoning tasks (science or open-ended) is questionable.
* There is a lack of experiments on a broader range of and more recent LLMs, e.g., qwen3.

**Questions:**

* Theorem 1 gives $p_{cot} < 1/3$ as the exact point where greedy decoding stays short. Is this an artifact of the two-step linear decision model, or does it survive richer embeddings (e.g., transformers with non-linear MLPs)? An ablation that keeps the data distribution but increases model expressivity would clarify whether the threshold is distribution- or architecture-specific.

* The post-training bound hides the per-round sample size inside Õ(·) and assumes fresh data every round. How many total unique prompts does the algorithm really need?

* Parity has a single deterministic “long path”. Real reasoning data often contain many valid chains of varying length and quality. Does the two-component mixture still capture the dynamics, or does the presence of noisy/partial chains shift the critical $p_{cot}$ or require a different RL objective?

* The theory 2 requires $p_{cot} \in \Omega (d^{-\kappa})$. For $d\approx 1,000$ (typical for LLM prompts) this seems prohibitive. Is the polynomial dependence tight, or do transformers empirically succeed with much smaller constants?

---

> ### Author Response · Authors · 2025-11-24
>
> Thank you very much for your review. Before responding to your questions, let us briefly defend the contributions of the paper:
>
> > All conclusions of the paper are drawn around the task of predicting the parity of d bits [...].
> > There is a lack of experiments on a broader range of and more recent LLMs, e.g., qwen3.
>
> Indeed, the theoretical proofs are restricted to the parity target function. However, while we use the parity problem to initiate our study, our empirical results go far beyond it. We train randomly initialized GPT2 models on the task of number multiplication and observe exactly the same phenomena as in the parity setting. Moreover, in Section 5 we experiment with pre-trained models. Although we do not have direct control over the pre-training data of the Llama models, we conduct a similar study by first fine-tuning the model (SFT) on a mixed distribution of short and long mathematical reasoning data, followed by RL on the SFT model. In some cases, we observe an excellent alignment with our theoretical picture; for example, on GSM8k, the model during RL starts imitating the long examples present in the SFT distribution. The GSM8k experiments with Llama alone required more than 11,000 GPU hours (Appendix C.3). Altogether, we argue that the conclusions of the paper do not hinge solely on the parity problem, and our experiments include substantial evidence from "real" LLMs.
>
> Regarding your questions:
>
> > Theorem 1 gives $p_{cot} < 1/3$ [...] clarify whether the threshold is distribution- or architecture-specific.
>
> This threshold depends on both the distribution and the learner (architecture and loss function), but we argue that for any **reasonable** architecture and the log-loss, the threshold remains at 1/3. It is not an artifact of the two-step linear model used at the second position, since we have already shown empirically (e.g., Figure 2, right) that transformers exhibit the same threshold. At a high level, this occurs because any sufficiently trained model that does not have access to the parity feature $\prod_{i=1}^d x_i$ will produce similar conditional probabilities in the first and second output positions, which in turn determine the threshold. This was outlined in Remark 2, below Theorem 1. We have added a more comprehensive explanation in Appendix F.2. In particular, we added a new result (Lemma 1) that formalizes our previous assumption on "no access to a parity feature" and establishes the generality of the threshold; its proof relies on basic ideas from the Fourier analysis of Boolean functions.
>
> > The post-training bound hides the per-round sample size inside Õ(·) and assumes fresh data every round. How many total unique prompts does the algorithm really need?
>
> Indeed, all our parity results (both empirical and theoretical) are obtained in the online regime, where new data are sampled at each iteration. For this reason, the algorithm is not "concerned" with the "uniqueness" of the samples. Determining exactly how many unique samples are needed goes beyond the types of questions that learning theory typically addresses. However, we can estimate the number of unique samples the algorithm uses, which, as we show, is sufficient for generalization:
>
> Since the distribution is uniform over the Boolean hypercube in $d$ dimensions and the number of single-batch SGD updates $T$ is sub-exponential in $d$, the expected number of unique examples is roughly $T$. To see this, let $\mathbf{x}^{(1)}, \ldots, \mathbf{x}^{(T)} \sim \mathrm{Unif}( \{ \pm 1\}^d )$ be the inputs ("prompts") we sample at each SGD iteration and define the count of unique samples $S = \sum_{t = 1}^T 1 [ \{ \mathbf{x}^{(t)} \neq \mathbf{x}^{(j)}] \, \forall j < t \}$. We compute its expected value: $\mathbb{E} S = \sum_{t = 1}^T \mathbb{P}[ \mathbf{x}^{(t)} \neq \mathbf{x}^{(j)} \, \forall j < t ] = 1 + \sum_{t = 2}^T \prod_{j = 1}^{t-1} \mathbb{P} [ \mathbf{x}^{(t)} \neq \mathbf{x}^{(j)}] = 2^d ( 1 - ( 1 - \frac{1}{2^d} )^T).$ For $T \ll 2^d$ (which is the case in all of our guarantees), this simplifies to:
> $\mathbb{E} S \approx 2^d ( 1 - ( 1 - \frac{T}{2^d} ) \) = T$. Thus, the algorithm sees approximately $T$ unique "prompts" per RL round, where $T = \max ( T_1, T_{2a}, \frac{2 \max_{l \in \{2, \ldots, d\} T_l}}{p_{\mathrm{cot}}}, \frac{8}{p_{\mathrm{cot}}} \ln \frac{2 \log (\frac{1 - p_{\mathrm{cot}}}{p_{\mathrm{cot}}})}{\delta} )$ with the per-position $T_j$'s defined as in Theorem 5. The total number over the whole execution of the RL algorithm is $O( \log ( \frac{1-p_{\mathrm{cot}}}{p_{\mathrm{cot}}}) T ),$ since we showed that $O ( \log \frac{1 - p_{\mathrm{cot}}}{p_{\mathrm{cot}}} )$ RL rounds suffice to obtain a generalizing model.
>
> edit: fixed latex

---

> > ### Author Response · Authors · 2025-11-24
> >
> > > Parity has a single deterministic “long path” [...] Does the two-component mixture still capture the dynamics, or does the presence of noisy/partial chains shift the critical $p_{\mathrm{cot}}$ or require a different RL objective?
> >
> > No, the analysis is no longer valid when there is noise or additional reasoning paths. Our original submission already contained an extension of our main setting where the data contain reasoning paths of long, medium and short lengths (Appendix D.2). In that setting, we observed similar difficulty to generalize when $p_{\mathrm{cot}}$ or $p_{\mathrm{odd}}$ are small. To answer your question, we added a short calculation showing how the critical thresholds change from our "vanilla" setting (pg. 27-28). We also included additional simulations in Figure 14, where we showed how "medium" length chain of thought data can make pre-training "easier". We also improved the clarity of Figure 13 that demonstrates our findings regarding length penalties in this setting.
> >
> > Regarding the presence of noise, we added an entirely new section (Appendix D.3) where we study two types of noise in the pre-training data: noise that either affects the whole chain or just the final prediction token. In many cases, we observe a similar learning behaviour as in the noise-less setting (models struggle to generalize during pre-training, yet they immediately generalize during post-training). As the amount of noise increases, our experiments reveal very interesting phenomena, where post-training is able to "erase" the pre-training noise and overcome an apparanent lack of overlap with the target. It will be very interesting to understand this behaviour better in future work.
> >
> > Please consider taking a look at Sections D.2, D.3, and let us know if you have any questions.
> >
> > > The theory 2 requires [...] Is the polynomial dependence tight, or do transformers empirically succeed with much smaller constants?
> >
> > Our theory suggests that even if $p_{\mathrm{cot}}$ is very small, as long as it not exponentially small in the dimension (i.e., there exists a constant $\kappa \in \mathbb{N}$ such that $p_{\mathrm{cot}}\geq cd^{-\kappa}$ for some $c > 0$), then the combination of pre-training and post-training will yield a generalizing model using a number of iterations polynomial in $d$. This is good news, as it indicates that even for difficult LLM tasks, the number of long demonstrations required for RL to succeed can be small. Might it be that you may have either missed the negative exponent in our statement or misinterpreted our use of the $\Omega$ notation? Please let us know if our explanation aboves resolves the confusion.
> >
> > We hope this addresses the questions raised in the review. If so, we kindly ask you to consider raising your score. Otherwise, we would be happy to address any remaining concerns.

---

### Official Review · Reviewer_pXVH · 2025-10-30

**Soundness:** 3
**Presentation:** 3
**Contribution:** 3
**Rating:** 4
**Confidence:** 2

**Summary:**

This paper proposes a framework to theoretically understand the success the popular LLM training paradigm that RL post-training after SFT. This paper also uses the parity check experiments to show that RL enables the model to generalize to difficult tasks while SFT cannot. The paper also demonstrated this phenomenon on math problems.

**Strengths:**

- the paper is well-motivated - trying to understand the reason behind the current successful LLM training paradigm.
- the paper proposed a relatively comprehensive theoretical analysis framework.
- the paper empirically validated its perspective through a cleverly designed and controlled parity check experiment.

**Weaknesses:**

1. The theoretical proof relies on a simplified linear autoregressive model.
2. Some of the ideas of this paper are already pointed out in papers like DeepSeek-R1.
2. This paer lacks of practical guidance for future LLM training and new algorithm.

**Questions:**

- Could similar phenomena be observed when in a worded version of a parity task? Such as giving the task description as language input of a LLM.
- The theoretical analysis relies relies on a simple linear framework, how is it extended to non-linear structures like the Transformers?

---

> ### Author Response · Authors · 2025-11-24
>
> Thank you very much for your review. We first rebut some of the concerns you raised:
>
> > Some of the ideas of this paper are already pointed out in papers like DeepSeek-R1.
>
> We argue that this not entirely correct. Indeed, the starting point of our research was the observation that response length increases during RL in autoregressive LLMs that was popularized by the DeepSeek paper. However, it is entirely unclear **why** this happens (as it is not part of the optimization objective). In our work, we address this question (among others) both empirically and **theoretically** in a simplified setting where the data consist only of long and short sequences encoding a single task.
>
> > This paper lacks of practical guidance for future LLM training and new algorithm.
>
> In general, we believe that theoretical insights and understanding can often lead to new algorithmic ideas. One immediate direction from our study is understanding how and when to apply length penalties during RL, a question explored in Appendix D.2. In particular, we show that length penalties applied at late (rather than early) pre-training checkpoints can reduce response length, without sacrifising performance gains during RL.
>
> Regarding your questions:
>
> > Could similar phenomena be observed when in a worded version of a parity task? Such as giving the task description as language input of a LLM.
>
> Yes, we believe this is entirely possible, and in fact this is what our extensive empirical studies in Section 5 demonstrate. There, we considered a similar setting in which we first fine-tuned the model (SFT) on a mixture distribution of short and long mathematical reasoning data, followed by RL on the SFT model. As in the parity setting, we observed that in many scenarios with small $p_{\mathrm{cot}}$, next-token prediction alone (SFT) does not improve model performance, yet a few iterations of RL "upsample" the presence of long chain-of-thought data, which in turn helps the model generalize (see for example Figure 4 - and our qualitative analysis in Figures 30, 31, 32, 33).
>
> > The theoretical analysis relies relies on a simple linear framework, how is it extended to non-linear structures like the Transformers?
>
> Unfortunately, analyzing the non-convex landscape of transformer training without any simplifications of the learning procedure remains elusive in the literature. Some of the existing analysis we are aware of (which also study learning the parity target function) simplify both the architecture (e.g., fixing the MLP weights) and the learning algorithm (e.g., using a one-step gradient descent update) [1]. We view our theoretical results as powerful because they show that even a relatively simple architecture can exhibit the post-training benefits we observe in practice with LLMs. In a sense, our work places the post-training paradigm at the center of the empirical success, independent of the specific details of the transformer architecture.
>
> 1. Juno Kim, Taiji Suzuki. Transformers Provably Solve Parity Efficiently with Chain of Thought. ICLR 2025
>
> Please let us know if there are any outstanding concerns. Otherwise, we would kindly ask you to revisit your rating of our work. Thank you!

---

### Official Review · Reviewer_StSb · 2025-10-30

**Soundness:** 4
**Presentation:** 3
**Contribution:** 4
**Rating:** 8
**Confidence:** 3

**Summary:**

This paper studies why RL after next-token prediction helps large language models learn better reasoning. The authors build a simple setting where training data mixes short and long “chain-of-thought” examples. They theoretically and experimentally prove that next-token prediction alone often fails when long samples are less than 1/3 in pretraining dataset. And RL can quickly improve learning by focusing on longer samples, which leads to longer and more correct responses.

**Strengths:**

1. Strong theoretical proof combined with solid experiments, showing when and why RL succeeds.
2. The paper gives a simple and convincing explanation of why RL helps reasoning. Providing clear insights.
3. The answer of  two core questions—"why RL works" and "why length increases" could offer a new design direction for LLM reasoning optimization.

**Weaknesses:**

1. The theory assumes fully correct CoT examples. However, the real pretrain data usually contains noisy or wrong data. And even positive RL trajectories could contain noise or false-positive ones. The paper lacks discussion of these factors.
2. The theoretical proof could be written in a more organized manner, including formulas. This would make that part easier to understand.

**Questions:**

1. How robust are the theoretical conclusions if long CoT samples contain noise?
2. Can you still observe “length-driven generalization” when RL rewards contain a certain proportion of length penalty?

---

> ### Author Response · Authors · 2025-11-24
>
> Thank you very much for your review and your appreciation of our insights, theory and the experimental underpinnings! We respond to your concerns:
>
> > The theory assumes fully correct CoT examples. [...]
> > How robust are the theoretical conclusions if long CoT samples contain noise?
>
> Thank you for this suggestion. We included an entirely new section in Appendix D.3 devoted exactly to the question of how pre-training noise affects learning behaviour. In particular, we considered two different types of noise: one that affects the final output token, and another that affects the whole reasoning chain. In many cases, we observe similar learning behaviour as in the noise-less setting (models struggle to generalize during pre-training, yet they immediately generalize during post-training). Please consider taking a look there to see how pre-training behaviour is affected by noise (for instance, we provide calculations on what the pre-training accuracy under sampling with temperature 1.0 will be). As the amount of noise increases, our experiments reveal very interesting phenomena, where post-training is often able to "erase" the pre-training noise and overcome an apparent lack of overlap with the target, with modest sample-complexity overheads (e.g. Figures 19 and 21). It will be very interesting to fully analyze this behaviour theoretically in future work.
>
> > The theoretical proof could be written in a more organized manner, including formulas. This would make that part easier to understand.
>
> Thank you for the feedback. In the revised version, we re-organized parts of the proofs and added additional commentary between formulas. We believe we have made the proofs of Theorems 4 and 5 much more explicit, and we also corrected a few typos in the statement of Theorem 5. For example, we added more details on the union bound in the proof of Theorem 4 (lines 3834-3844).
>
> > Can you still observe “length-driven generalization” when RL rewards contain a certain proportion of length penalty?
>
> Yes! This is explored in Section D.2 of the Appendix, where we considered a variation of our main parity setting with additional "medium"-length chain of thought data. There we show that length penalties are more effective at later post-training checkpoints, since there is a greater chance that the model has learned from "partial" chain-of-thought data. In particular, in Figure 13 (bottom right), we show that a linear length penalty in the reward can lead to length-driven generalization. We improved the clarity of this Figure in the revised version.
>
>
> We appreciate your review. If you feel that our new experiments on noisy data, together with our explanations, addressed your concerns, we would appreciate it if you recommended this work to be highlighted at the conference. Thank you!

---

### Author Response · Authors · 2025-11-24

We would like to thank all reviewers for their comments and the appreciation of our work, the strong theoretical proof combined with solid experiments [StSb], the comprehensive theoretical analysis framework and empirical validation [pXVH] and the first of its kind formal separation between next-token prediction and next-token prediction plus RL in autoregressive models [Hty6].

Following your suggestions and inspired by your questions, in the revised manuscript, we implemented the following changes:

- We added a new section (Appendix D.3, together with Figures 15, 16, 17, 18, 19, 20, 21) studying how pre-training noise affects the learning behaviour in the main parity setting.
- We added calculations on the critical threshold for the setting with partial chain of thought data (Appendix D.2) and new simulations in Figure 14.
- We added a more detailed explanation of why the critical pre-training threshold is expected to be 1/3 in the main parity setting, as well a theoretical result on the generality of this observation (new section Appendix F.2). In particular, Lemma 1 formalizes what it means for a model to not have access to a parity feature. The proof uses basic ideas from the Fourier analysis of Boolean functions.
- We re-organized some parts of the proofs, adding more details (in particular in the proof of Theorem 4) and corrected some typos.
- We fixed other typos in captions and figures.

All changes are highlighted in orange.

---

### Meta-Review · Area_Chair_mZ1c · 2026-01-05

**Summary:**

This paper presents theoretical and empirical results to explain why LLMs are improved dramatically with RL that follows SFT. The authors prove that next-token prediction alone cannot generalize on hard tasks with rare long reasoning chains, whereas RL can effectively amplify these rare sequences and enable efficient learning.

The reviewers generally found the results to be interesting and solid. Two out of three reviewers voted for acceptance, and the slightly negative reviewer (pXVH) raised a few concerns regarding the oversimplified theoretical framework (a linear autoregressive model) and novelty (findings already presented in past studies). The authors provided good responses to address these concerns, which in my view are rather minor and do not offset the contributions of this paper.

**Reviewer Concerns:**

The slightly negative reviewer's (pXVH) concerns have been well addressed by the authors' responses, and the other two positive reviewers' concerns are quite minor.

**Reviewer Scores:**

All reviewers would tend to feel positive about this work and the scores would have been slightly higher.

---

### Decision · Program_Chairs · 2026-01-26

Accept (Poster)